# On the contribution of grain boundary sliding type creep to firn densification – an assessment using an optimization approach

Timm Schultz[1], Ralf Müller[1], Dietmar Gross[2], and Angelika Humbert[3,4]

[1]Institute of Applied Mechanics, Technische Universität Kaiserslautern, Kaiserslautern, Germany
[2]Division of Solid Mechanics, Technische Universität Darmstadt, Darmstadt, Germany
[3]Alfred-Wegener-Institut Helmholtz-Zentrum für Polar- und Meeresforschung, Bremerhaven, Germany
[4]Faculty of Geosciences, University of Bremen, Bremen, Germany

**Correspondence:** Timm Schultz (tschultz@rhrk.uni-kl.de)

**Abstract.** Simulation approaches to firn densification often rely on the assumption that grain boundary sliding is the leading process driving the first stage of densification. Alley (1987) first developed a process-based material model of firn that describes this process. However, often so-called semi-empirical models are favored over the physical description of grain boundary sliding owing to their simplicity and the uncertainties regarding model parameters. In this study, we assessed the applicability of the grain boundary sliding model of Alley (1987) to firn using a numeric firn densification model and an optimization approach, for which we formulated variants of the constitutive relation of Alley (1987). An efficient model implementation based on an updated Lagrangian numerical scheme enabled us to perform a large number of simulations to test different model parameters and identify the simulation results that best reproduced 159 firn density profiles from Greenland and Antarctica. For most of the investigated locations, the simulated and measured firn density profiles were in good agreement. This result implies that the constitutive relation of Alley (1987) characterizes the first stage of firn densification well when suitable model parameters are used. An analysis of the parameters that result in the best agreement revealed a dependence on the mean surface mass balance. This finding may indicate that the load is insufficiently described, as the lateral components of the stress tensor are usually neglected in one-dimensional models of the firn column.

## 1 Introduction

Firn densification models fall into two basic categories. Models in the first category, which includes most existing models, follow the so-called semi-empirical approach of Herron and Langway (1980), which itself is based on Sorge's Law (Bader, 1954) and the Robin hypothesis (Robin, 1958). Examples are the models of Arthern et al. (2010), Ligtenberg et al. (2011), and Simonsen et al. (2013). The empirical parameters of these models are typically adjusted on the basis of certain datasets of depth density profiles. In the second category of firn densification models, an attempt is made to quantify the physical processes related to firn densification. These processes include various types of creep and diffusion. Micromechanical models are used for

small-scale investigations (Johnson and Hopkins, 2005; Theile et al., 2011; Fourtenau et al., 2020), whereas models based on continuum mechanics can be used for large-scale simulations. Examples of the latter are the models of Arthern and Wingham (1998), Arnaud et al. (2000), and Goujon et al. (2003).

Alley (1987) first applied the theory of grain boundary sliding adopted from Raj and Ashby (1971) to firn densification at densities below the critical density of $\rho_c = 550\,\mathrm{kg\,m^{-3}}$. The description of this process by Alley (1987) was subsequently used in other firn densification models (Arthern and Wingham, 1998; Arnaud et al., 2000; Goujon et al., 2003; Bréant et al., 2017). However, the assumption that grain boundary sliding is the dominant process in firn densification at densities below $\rho_c = 550\,\mathrm{kg\,m^{-3}}$ has been questioned numerous times (Ignat and Frost, 1987; Roscoat et al., 2010). For example, Theile et al.

(2011) conducted experiments on a small number of snow samples and suggested that densification is more likely driven by processes within the grain rather than by the intergranular process of grain boundary sliding.

In this study, we aim to evaluate (i) whether the description of grain boundary sliding given by Alley (1987) is suitable for the simulation of firn densification at low density, (ii) how a modification of the constitutive relation introduced by Bréant et al. (2017) affects simulation results, (iii) whether hidden or additional dependencies on climatic or other conditions can be

identified in the constitutive relation, and (iv) how the constitutive relation of Alley (1987) might be improved. Note that our study aims to assess the constitutive relation for grain boundary sliding proposed by Alley (1987). An evaluation that clarifies whether grain boundary sliding is the dominant process driving firn densification below the critical density of $\rho_c = 550\,\mathrm{kg\,m^{-3}}$ must be conducted using other methods. Experimental attempts to do so have been made by, for example, Kinosita (1967), Ignat and Frost (1987), and Theile et al. (2011). In contrast to these experimental investigations, a data-driven model approach

is used in our study. Since the original study of Alley (1987) was published, the amount of available data has increased greatly. The data include a large number of firn profiles and forcing data, and they allow us to simulate firn profiles at very high quality using additional modeling techniques.

## 2    Methods

To test the description of grain boundary sliding given by Alley (1987), we used a numerical model to simulate the evolution

of a one-dimensional firn column with time. The model uses variants of the constitutive relation of Alley (1987), all of which combine several model parameters in a single factor. We force the model with data provided by the regional climate model RACMO2.3 (Van Wessem et al., 2014; Noël et al., 2015), which represents the climate of recent decades at 159 locations where firn density measurements were made. These firn measurements are available through the Surface Mass Balance and Snow Depth on Sea Ice Working Group (SUMup) snow density subdataset (Koenig and Montgomery, 2019). By varying

the factor used in the variants of the constitutive equation, we produced a large number of simulation results, which were compared to the corresponding density measurements. The quality of the factors used in the simulations was evaluated in terms of the deviation of the computed density profiles from the measured profiles. The factor values that yielded the best results reveal possible improvements in the description of grain boundary sliding for firn densification at low density. In the following

sections, the constitutive equation for grain boundary sliding given by Alley (1987), the optimization scheme, and the used
density and forcing data are described. A detailed description of the model is presented in the appendix (Sect. A).

## 2.1 Constitutive relation for grain boundary sliding

We explain briefly the following components and characteristics of the constitutive law of Alley (1987) describing the process
of grain boundary sliding.

$$\dot{\varepsilon}_{zz} = -\frac{2}{15} \quad \delta_b \quad \frac{8 D_{\mathrm{BD}} \Omega}{k_b T h^2} \quad \frac{1}{r \mu^2} \quad \left( \frac{\rho_{\mathrm{ice}}}{\rho} \right)^3 \quad \left( 1 - \frac{5}{3} \frac{\rho}{\rho_{\mathrm{ice}}} \right) \quad \sigma_{zz} \quad , \qquad D_{\mathrm{BD}} = A_{\mathrm{BD}} \exp \left( -\frac{Q_{\mathrm{BD}}}{RT} \right) \tag{1}$$

The factor of $2/15$ results from the geometric deviation pointed out by Alley (1987). Another geometric parameter, $\delta_b$, describes
the width of the grain boundary.

The following part of the equation describes the reciprocal bond or boundary viscosity (Raj and Ashby, 1971). The optimiza-
tion approach of Alley (1987) was intended to identify the optimal values of the boundary viscosity. Alley (1987) compared
the results of this optimization to the description of the boundary viscosity given by Raj and Ashby (1971), which includes
the volume of the $H_2O$ molecule $\Omega$, the Boltzmann constant $k_b$, the temperature $T$, and the amplitude of grain boundary ob-
structions, $h$. The latter is a measure of the roughness of the grain boundary. $D_{\mathrm{BD}}$ is an Arrhenius factor describing the rate
of boundary diffusion. Values of the typical activation energy for this process, $Q_{\mathrm{BD}}$, and the corresponding prefactor, $A_{\mathrm{BD}}$,
can be found in the literature (e.g., Maeno and Ebinuma, 1983), and are further discussed in Sect. 2.2. $R$ is the universal gas
constant.

The strain rate resulting from grain boundary sliding also depends on the grain radius $r$. The ratio of the grain radius to the
neck radius $\mu$ was introduced by Arthern and Wingham (1998) and is assumed to be constant. Various methods can be used
to determine the size of grains in crystalline materials (e.g., Gow, 1969). The model of Alley (1987) was developed assuming
perfectly spherical grains. Although this assumption is not true of firn, it provides a reasonable basis for modeling. Therefore,
throughout this paper, the grain radius $r$ represents the radius of theoretical spherical grains.

The next factor in Eq. (1) describes the dependence on the inverse density relative to the ice density cubed. The factor
of $1 - (5 \rho / 3 \rho_{\mathrm{ice}})$ causes the strain rate due to grain boundary sliding to decrease with increasing density until it vanishes
at the critical density of $\rho_c = 550 \, \mathrm{kg} \, \mathrm{m}^{-3}$. When the critical density is reached, close random packing is established (Ander-
son and Benson, 1963), and grains can no longer slide against each other; thus, the process of grain boundary sliding ends.
Other deformation processes, in particular dislocation creep (Maeno and Ebinuma, 1983), result in further densification with
increasing stress.

Alley (1987) suggested that additional processes contribute to densification below the critical density. It is feasible that the
effect of grain boundary sliding on the strain rate decreases, whereas that of other processes increases. The studies of Arthern
and Wingham (1998) and Bréant et al. (2017) use this description, in which only grain boundary sliding drives densification in
the first stage of firn densification. In the study of Bréant et al. (2017), the constitutive relation of Alley (1987) is changed such
that the transition into the second stage of densification is modified. We evaluate this modification in this work.

Finally, the stress in the vertical direction, $\sigma_{zz}$, resulting from the overburden firn drives grain boundary sliding. Whereas Alley (1987) used the product of the accumulation rate, acceleration due to gravity, and time since the deposition of a specific firn sample to describe the overburden stress, we use a more general form here [see Sect. A2, Eq. (A5)]. The other physical properties affecting the process are density $\rho$, temperature $T$, and grain radius $r$.

## 2.2 Optimization

To test the material model developed by Alley (1987), we formulated variants of Eq. (1) and compared the model results to density measurements of various firn cores. These variants of the constitutive equation [Eqs. (2) to (5)] preserve its general form but group several material parameters into a single factor. As a result, the simulation result does not depend on those parameters, but on the single factor. The factor is then varied to find an optimal simulation result that best reproduces the measured firn profile. The factor that yields the optimal simulation result depends on the measured firn density profile and the corresponding climate conditions. It is therefore site-specific. This feature makes it possible to assess whether the description of grain boundary sliding given by Alley (1987) can be used to reproduce measured firn profiles, assuming an optimal set of parameters. It further allows us to analyze the site-specific factors yielding the best simulation results for possible hidden dependencies.

Arnaud et al. (2000), Goujon et al. (2003), and Bréant et al. (2017) also incorporated the material parameters of the model of Alley (1987) into a single parameter. Bréant et al. (2017) also modified the factor of $5/3$ to change the density at which the deformation due to grain boundary sliding becomes zero. In the following, four variants, indicated by the subscripts $(\cdot)_{\mathrm{v}_1}$ to $(\cdot)_{\mathrm{v}_4}$, are presented:

$$\dot{\varepsilon}_{zz\,\mathrm{v}_1} = -C_{\mathrm{v}_1} D_{\mathrm{BD}} \frac{1}{T} \frac{1}{r} \left( \frac{\rho_{\mathrm{ice}}}{\rho} \right)^3 \left( 1 - \frac{5}{3} \frac{\rho}{\rho_{\mathrm{ice}}} \right) \sigma_{zz} \quad , \qquad D_{\mathrm{BD}} = A_{\mathrm{BD}} \exp\left( -\frac{Q_{\mathrm{BD}}}{RT} \right) , \tag{2}$$

$$\dot{\varepsilon}_{zz\,\mathrm{v}_2} = -C_{\mathrm{v}_2} D_{\mathrm{BD}} \frac{1}{T} \frac{1}{r} \left( \frac{\rho_{\mathrm{ice}}}{\rho} \right)^3 \left( 1 + \frac{0.5}{6} - \frac{5}{3} \frac{\rho}{\rho_{\mathrm{ice}}} \right) \sigma_{zz} \quad , \qquad D_{\mathrm{BD}} = A_{\mathrm{BD}} \exp\left( -\frac{Q_{\mathrm{BD}}}{RT} \right) . \tag{3}$$

Variant 1 [Eq. (2)] and Variant 2 [Eq. (3)] of the constitutive equation combine all material constants using the factors $C_{\mathrm{v}_1}$ and $C_{\mathrm{v}_2}$, respectively. The Arrhenius factor for boundary diffusion, $D_{\mathrm{BD}}$, [see Eq. (1)] is retained in these variants. Following Maeno and Ebinuma (1983), we use a value of $Q_{\mathrm{BD}} = 44.1\,\mathrm{kJ\,mol}^{-1}$ for the boundary diffusion activation energy. This variable was defined by Maeno and Ebinuma (1983) as two-thirds of the activation energy for lattice diffusion measured by Itagaki (1964). The corresponding prefactor is $A_{\mathrm{BD}} = 3.0 \times 10^{-2}\,\mathrm{m}^2\,\mathrm{s}^{-1}$. Alley (1987) assumed a similar value for the boundary diffusion activation energy.

In addition to the temperature $T$, the vertical strain rate $\dot{\varepsilon}_{zz}$ depends only on the firn density $\rho$, grain radius $r$, and stress in the vertical direction $\sigma_{zz}$. Variant 2 differs from Variant 1 in that it uses the modification introduced by Bréant et al. (2017). This modification causes a theoretical end to the process of grain boundary sliding at a density of $\rho_c^* = 596\,\mathrm{kg\,m}^{-3}$. It was introduced to obtain a better transition to the second stage of firn densification. The strain rate due to grain boundary sliding is therefore higher at the critical density when this modification of Bréant et al. (2017) is used.

To determine the effect of the Arrhenius factor, it is disregarded in Variants 3 and 4, as shown in Eqs. (4) and (5):

$$\dot{\varepsilon}_{zz\,\mathrm{v}_3} = -C_{\mathrm{v}_3} \frac{1}{T}\frac{1}{r} \left( \frac{\rho_{\mathrm{ice}}}{\rho} \right)^3 \left( 1 - \frac{5}{3}\frac{\rho}{\rho_{\mathrm{ice}}} \right) \sigma_{zz}\,, \tag{4}$$

120

$$\dot{\varepsilon}_{zz\,\mathrm{v}_4} = -C_{\mathrm{v}_4} \frac{1}{T}\frac{1}{r} \left( \frac{\rho_{\mathrm{ice}}}{\rho} \right)^3 \left( 1 + \frac{0.5}{6} - \frac{5}{3}\frac{\rho}{\rho_{\mathrm{ice}}} \right) \sigma_{zz}\,. \tag{5}$$

In Variant 4, the modification of Bréant et al. (2017) is used, whereas Variant 3 uses the original formulation of Alley (1987). The goal of optimization is to find optimal values of the factors $C_{\mathrm{v}}$ for every variant of the constitutive relation [Eqs. (2) to (5)] to produce simulated density profiles that best reproduce the measured profiles. As an example, we explain the optimization process for one selected firn core in more detail. The upper part of ice core ngt03C93.2 (Wilhelms, 2000) is shown in Fig. 1a.

Every simulation begins with a spin-up period in which constant values are used for forcing. The model is forced with prescribed values of temperature, accumulation rate, firn density, and grain radius at the surface. We check for steady-state conditions by comparing the change in density between time steps. If the maximum density change is smaller than $|\Delta\rho_{\mathrm{max}}| < 0.1\,\mathrm{kg\,m}^{-3}$, we assume that the steady state is reached. In this case, a transient run using varying forcing data follows. We use a constant value of 48 time steps per year for spin-up and transient simulation runs (see Sect. A7). The forcing at the location of ngt03C93.2 is shown in Fig. 1c. The resulting firn profile is then compared to the measured profile. We used the root mean square deviation (RMSD) between the measured and modeled density for comparison, as it is simple and easy to compute. To calculate the deviation, the simulated density values are interpolated linearly to the measurement locations along the profile. To ensure the quality of the results, we limited the calculation of the deviation to the domain defined by the the location of the uppermost available measurement point and the oldest simulated horizon within the firn profile affected by forcing. For ngt03C93.2, this horizon is the 1958 surface at a depth of approximately $11\,\mathrm{m}$ below the surface, indicated by dashed horizontal lines in Fig. 1a. Only results located above the simulated 1958 surface are used to calculate the deviation. The reason is that the forcing data from RACMO2.3 (Van Wessem et al., 2014; Noël et al., 2015) begin in 1958 for Greenland. Consequently, the results are not affected by the spin-up period. For firn profiles retrieved in Antarctica, climate forcing from RACMO2.3 begins in 1979. Thus, only those results located above the simulated horizon of 1979 are considered for comparison with the Antarctic firn profiles. The examination of other firn cores revealed that the surface of the oldest available forcing may be located at greater depth when the density of $\rho_c = 550\,\mathrm{kg\,m}^{-3}$ has already been reached. In those cases, the computation of the RMSD was limited to the domain showing density values below $540\,\mathrm{kg\,m}^{-3}$. We decided to use a density threshold below the critical density because of the asymptotic nature of the resulting density profiles obtained using Variants 1 and 3 of the constitutive equation [Eqs. (2) and (4)]. The value of $540\,\mathrm{kg\,m}^{-3}$ was chosen to ensure that the results obtained using the variants of the constitutive relation are comparable, whereas unique values of the factor $C_{\mathrm{v}}$ were quickly determined throughout the optimization.

As the implementation of our model is efficient and the approach is simple and reliable, we decided to determine the best factor $C_{\mathrm{v}}$ for the four variants of the constitutive equation by simply testing 250 values within certain ranges. These ranges are shown in Eqs. (6) and (7), which include and exclude the Arrhenius factor, respectively. Optimal factors can be found within

these ranges for every analyzed firn profile. To ensure that this is the case, all simulations were performed multiple times using different ranges of the factors.

$$1.0 \times 10^{-9}\,\mathrm{K\,s^2\,kg^{-1}} \quad \leq C_{\mathrm{v_1,v_2}} \leq \quad 2.5 \times 10^{-4}\,\mathrm{K\,s^2\,kg^{-1}} \tag{6}$$

$$2.5 \times 10^{-21}\,\mathrm{K\,s\,m^2\,kg^{-1}} \quad \leq C_{\mathrm{v_3,v_4}} \leq \quad 5.0 \times 10^{-15}\,\mathrm{K\,s\,m^2\,kg^{-1}} \tag{7}$$

Figure 1b shows the RMSD plotted over the 250 tested values for the four different factors. The variants are color-coded, and the best results are marked. The smallest values of the deviation are shown in the figure. The corresponding density profiles are shown in Fig. 1a.

The firn profile of ngt03C93.2 starts at a depth of approximately $1.3\,\mathrm{m}$. Therefore, an appropriate surface density boundary
condition must be found. As firn density profiles differ greatly, especially near the surface, the derivation of an appropriate surface density is difficult. Although Alley (1987) simulated the density starting at a depth of 2 m below the surface, we included this domain in our simulation so that we could apply transient surface forcing to our model. To find suitable values of the surface density, we included this parameter in our optimization. For each of the four variants and 250 factors $C_{\mathrm{v}}$, we tested 21 values of the surface density between $\rho_0 = 250\,\mathrm{kg\,m^{-3}}$ and $\rho_0 = 450\,\mathrm{kg\,m^{-3}}$, using steps of $\Delta\rho_0 = 10\,\mathrm{kg\,m^{-3}}$. The
best result was chosen. We used the method of testing 21 surface density values for all the analyzed firn profiles. We included profiles including measurements of the density at small depths. In this way, we established that the results are comparable. Profiles including near-surface density values are, however, well-represented. A total of $4 \times 250 \times 21 = 21\,000$ simulations were performed for ice core ngt03C93.2 to find the optimal results shown in Fig. 1. The same procedure was applied to all 159 firn profiles analyzed in the study.

## 3 Data

### 3.1 Firn profiles

To evaluate the description of grain boundary sliding given by Alley (1987), we used 159 firn profiles, 80 of which were retrieved in Greenland. The remaining 79 measurements were taken in Antarctica. The profiles are included in the SUMup snow density subdataset (Koenig and Montgomery, 2019). Individual references for all 159 firn profiles are listed in the supplemen-
175 tary material. The dataset does not include the four profiles used in the study of Alley (1987), as the original data for these firn cores are unpublished. To obtain firn profiles suitable for this study from the dataset, we filtered it according to the following criteria:

1. Profiles must consist of at least 10 data points.

2. The overall length of each profile must exceed 3 m.

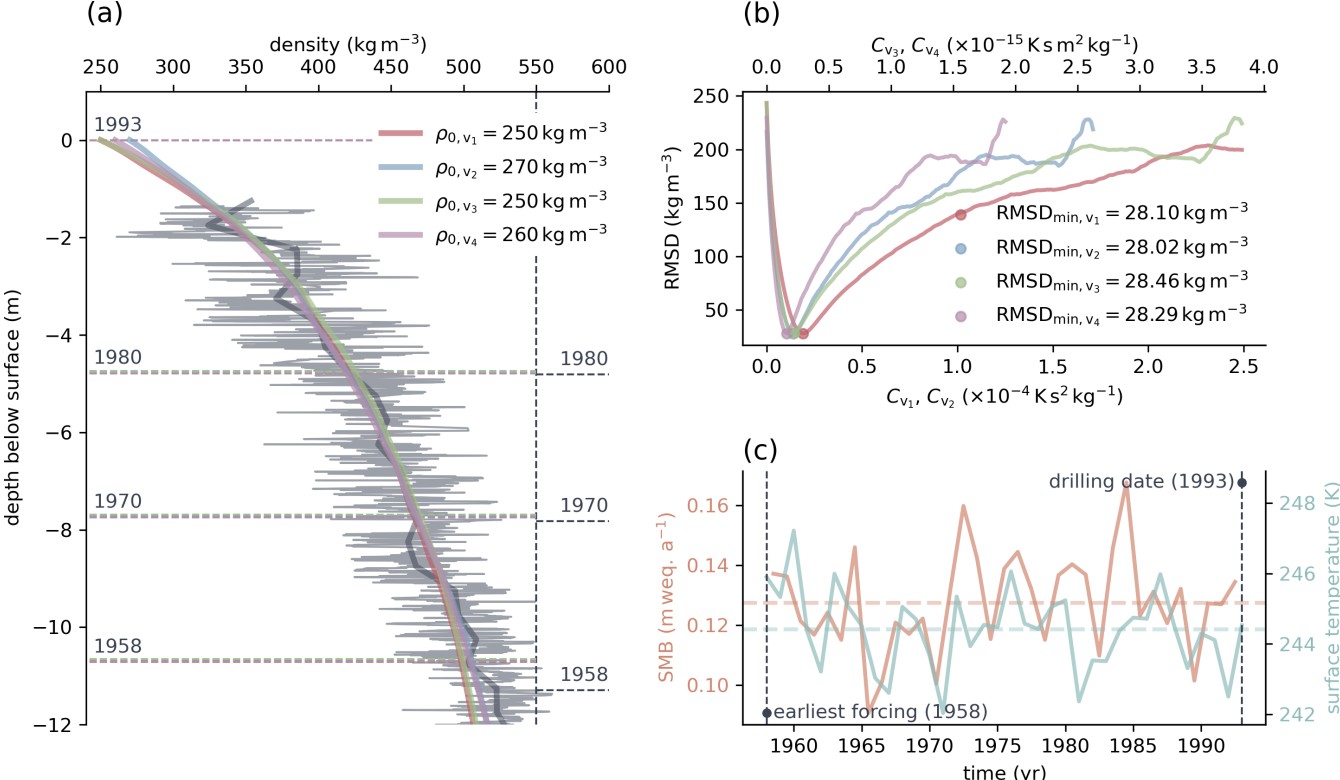

**Figure 1.** (a) Depth density profile of ice core ngt03C93.2 (Wilhelms, 2000) retrieved in Greenland (gray) and best corresponding model results obtained using four variants of constitutive law for grain boundary sliding of Alley (1987) (various colors). Dark gray line shows the mean density of the ice core calculated using a window of $0.5\,\mathrm{m}$ starting at the surface. Dashed horizontal lines represent horizons of firn deposited in the indicated years, where 1958 is the first year that forcing from RACMO2.3 (Van Wessem et al., 2014; Noël et al., 2015) is available. Colored dashed horizontal lines show horizons obtained in the simulations. Horizons plotted in gray (to the right of the vertical dashed line) represent the same surfaces as those determined by Miller and Schwager (2004) during analysis of the core. (b) RMSD between measured and modeled density plotted over the range of tested factor values. Note the different axes for different tested factors. (c) Representative forcing at the location of ice core ngt03C93.2, which was used in the simulation. Horizontal dashed lines show the mean values of the surface mass balance and surface temperature during the simulation.

3. Profiles must start at a depth of less than 3 m below the surface.

4. The initial density of the profiles must not exceed $\rho_c = 550\,\mathrm{kg\,m^{-3}}$.

5. The annual mean surface mass balance at the profile locations must be strictly positive.

6. Forcing data for at least five years must be available for the profile location.

Criteria 1 and 2 ensure the overall quality of the data, whereas criteria 3 and 4 ensure that the first stage of firn densification is included. As the model cannot handle melting, and the study focuses on dry firn densification, the surface mass balance should be positive, as stated in the fifth criterion. However, a positive mean annual surface mass balance does not ensure that no melting occurs over the course of a year. It is not possible to distinguish between sites affected by melting and sites where no melting occurs using the data available for this study. The number of these sites, however, is expected to be small. Because of the method used, their influence on the overall result is therefore small. The forcing data come from the regional climate model RACMO2.3 (Van Wessem et al., 2014; Noël et al., 2015), which provides the surface mass balance. The model delivers data for the periods 1958 to 2016 and 1979 to 2016 for Greenland and Antarctica, respectively (see also Sect. 3.2). The density measurements used for model comparison should thus be retrieved during these periods. We used only datasets for which at least five years of forcing data are available. Furthermore, a number of density profiles were manually excluded from the filtered data. These profiles include those with very low spatial resolution, atypical profiles showing decreasing density with depth, and measurements with a surface density very close to the critical density of $\rho_c = 550\,\mathrm{kg\,m^{-3}}$. As explained in Sect. 2.2 and illustrated in Fig. 1, we used only a certain domain for the comparison of the simulated and measured data. If this domain was found to be less than 2.5 m long for any of the tested variants of the constitutive equation, the firn profile in question was not analyzed further.

Figure 2 shows the locations from which the 159 density profiles were retrieved. The 80 measurements from Greenland are relatively uniformly distributed over the ice sheet. Coastal locations are not well covered owing to the requirement of a strictly positive surface mass balance. In the Antarctic, sites in East Antarctica are underrepresented. However, a wide variety of environments is covered, including the Filchner–Ronne Ice Shelf, the West Antarctic coast, and Dronning Maud Land.

## 3.2 Boundary conditions and forcing

To force the firn densification model, we need the surface values of density, temperature, accumulation rate, and grain radius at the locations of the 159 firn profiles. Although Alley (1987) used constant forcing, we followed the example of Arthern and Wingham (1998) and Goujon et al. (2003), who performed transient simulations.

As measured firn density profiles represent past climate conditions, the choice of forcing data in the proposed method is crucial. Uncertainties in the forcing will affect the simulation results and therefore the comparison with the measured firn profiles. Neither the model formulation nor the optimization scheme can compensate for these effects. We used data provided by the regional climate model RACMO2.3 (Van Wessem et al., 2014; Noël et al., 2015). Details on RACMO2.3, including the limitations of the model and the resulting data products, were presented by Van Wessem et al. (2014); Noël et al. (2015); van

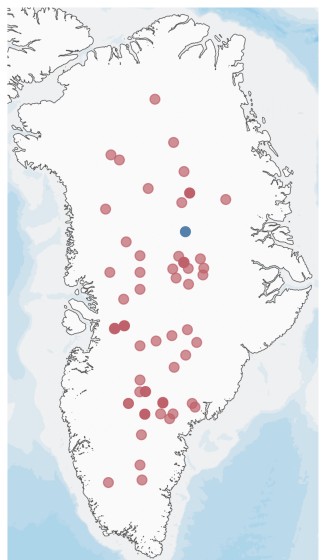
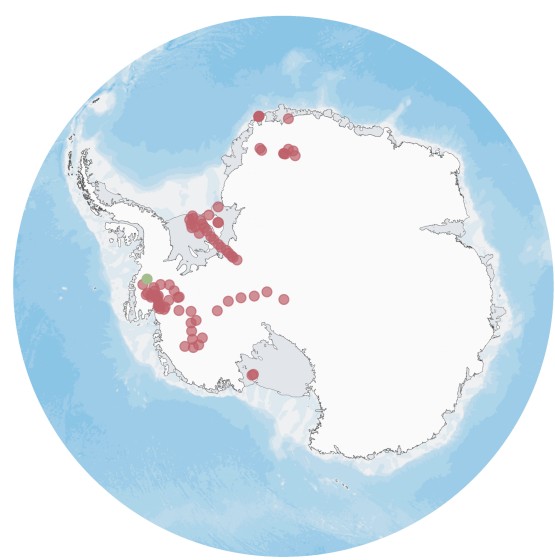

**Figure 2.** Locations of firn profiles used for model comparison. Eighty profiles were measured in Greenland, and 79 depth density datasets were retrieved in Antarctica. The blue marker shows the location of ice core ngt03C93.2 (Wilhelms, 2000, N $73.940°$, E $-37.630°$). The green marker shows the location of site 3 of the iSTAR traverse, from which the firn core shown in Fig. 3 was retrieved (Morris et al., 2017, N $-74.565°$, E $-86.913°$). Map data: Amante and Eakins (2009); Arndt et al. (2013), SCAR Antarctic Digital Database.

Wessem et al. (2018). RACMO2.3 provides forcing data for the Greenland ice sheet covering 1958 to 2016. For Antarctica, the time period is shorter, running from 1979 to 2016. Data for the mean annual skin temperature and surface mass balance of the Greenland ice sheet are available at mean spatial resolutions of $11.3\,\mathrm{km}$ and $1.0\,\mathrm{km}$, respectively, for this study. The mean spatial resolutions of the mean annual skin temperature and surface mass balance for Antarctica are $8.0\,\mathrm{km}$ and $28.5\,\mathrm{km}$, respectively.

The time period for the transient simulation runs, as described in Sect. 2.2, is specified by the earliest data available from RACMO2.3 and the drilling date of the firn core under consideration. For ice core ngt03C93.2 (Wilhelms, 2000), which was retrieved in central Greenland in 1993, the simulation time covers the period from 1958 to 1993 (see Fig. 1c). Constant values of the surface temperature and surface mass balance for the preceding spin-up period were calculated as mean values over this time range.

We present a second example to illustrate the effect of the temporal resolution of the forcing on the optimization results and why we used yearly averaged data provided by RACMO2.3. Figure 3 shows the depth density profile of the firn core retrieved at site 3 of the iSTAR Traverse in 2013 (Morris et al., 2017). The location of the site, at Pine Island Glacier in West Antarctica, is shown in Fig. 2. Instead of using forcing data from RACMO2.3, for this particular simulation we used ERA5-Land monthly averaged data from 1981 to present (Muñoz Sabater, 2019; Hersbach et al., 2020), as they are freely available at monthly resolution. From these data, we computed the annual average data for a second simulation run. The

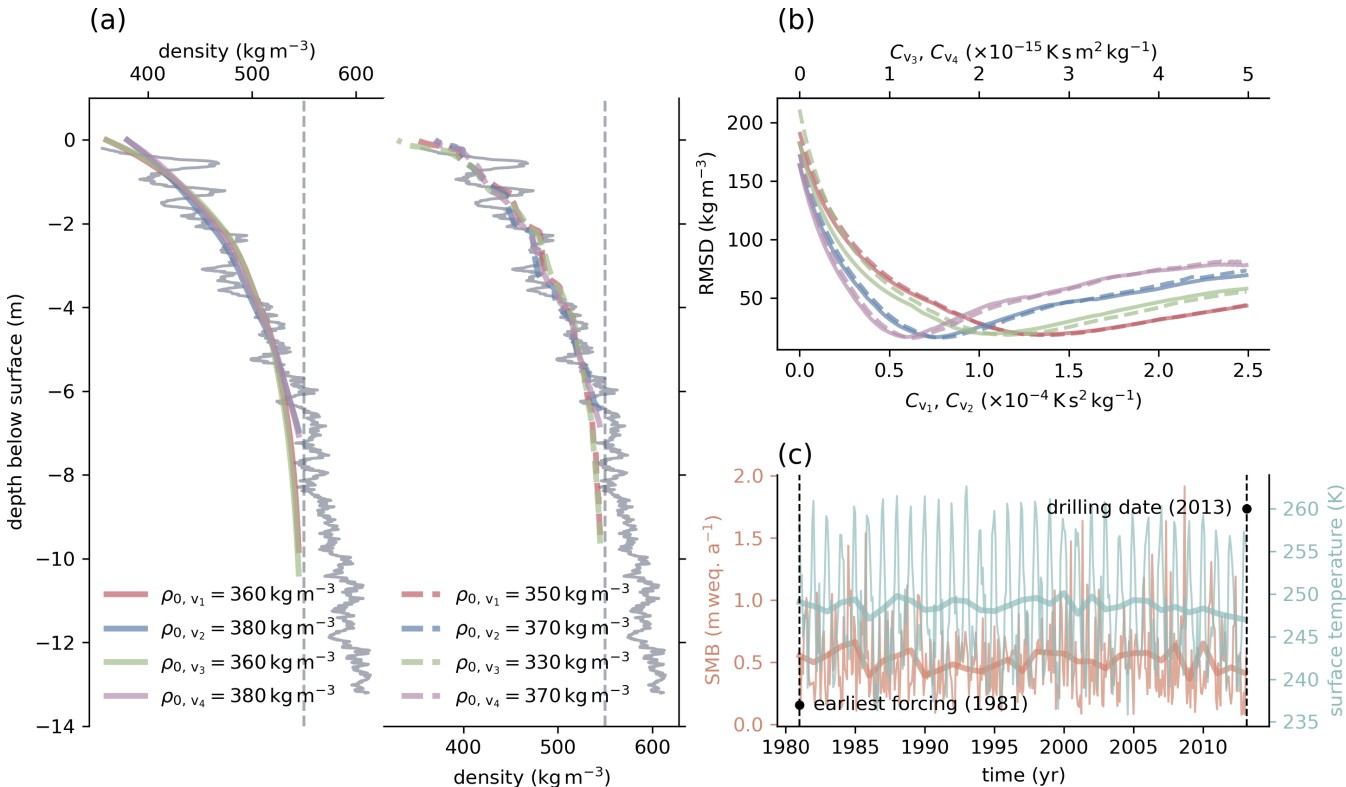

**Figure 3.** (a) Depth density profile (gray) of firn core retrieved at site 3 of the iSTAR traverse (Morris et al., 2017). Colored lines show the optimal simulation results for four tested variants of the constitutive relation. The simulated density profiles on the left were obtained using yearly averaged surface forcing, whereas those on the right, plotted using dashed lines, were obtained using monthly averaged forcing. (b) RMSD between best simulation result and measured firn profile over the range of tested optimization factors $C_v$. Colors indicate results for the different variants of the constitutive relation. Dashed lines indicate results computed with monthly averaged forcing, whereas solid lines indicate the use of yearly averaged surface forcing. (c) Forcing data from ERA5-Land monthly averaged data from 1981 to the present (Muñoz Sabater, 2019; Hersbach et al., 2020), from the earliest available forcing in 1981 to the date the firn core was drilled in 2013. Bold lines show yearly averaged data computed from monthly averaged data.

forcing data at both resolutions are shown in Fig. 3c. Figure 3a shows the simulated firn profiles that best reproduce the measured density profile, which were identified using the optimization approach. The results obtained using the annually averaged forcing data are shown on the left, whereas those obtained using monthly averaged data from ERA5 are shown on the right. The data with higher resolution data reveal much more detail within the simulated firn density profiles. However, the aim of this study is not primarily to reproduce the analyzed measured firn profiles with the highest possible detail, but to evaluate the constitutive relation of Alley (1987) using an optimization approach that identifies site-specific optimal constitutive factors $C_\mathrm{v}$ (see Sect. 2.2). Figure 3b shows the RMSD of the simulated profiles from the measured profile over the range of tested optimization factors. Dashed lines represent simulations performed using the high-resolution forcing data, whereas solid lines represent the annual averaged data. The difference between the optimization results is small. We therefore used the annual averaged data provided by RACMO2.3 suitable for this study, as the data cover a longer time period, especially for Greenland. As a result, we can analyze more firn profiles at greater detail. For ice core ngt03C93.2 (Wilhelms, 2000), the horizon of the year 1981, the earliest forcing available in ERA5, lies at a depth of approximately $5\,\mathrm{m}$ below the surface, as shown in Fig. 1a. The horizon of the earliest forcing available from RACMO2.3, the year 1958, is located at a depth of approximately $11\,\mathrm{m}$ below the surface. A much greater portion of the simulated firn profile is therefore affected by surface forcing. Furthermore, the use of yearly averaged data requires less overhead.

As noted in Sect. 2.2, we used 21 surface density values in the range $250\,\mathrm{kg\,m^{-3}} \le \rho_0 \le 450\,\mathrm{kg\,m^{-3}}$ for every tested firn profile. The value that yielded the best result was then used for further analysis. Owing to a lack of relevant data, and for simplicity and ease of comparison, the grain radius at the surface was assumed to be the same at all locations and to be constant over time. We used a grain radius of $r_0 = 0.5\,\mathrm{mm}$ on the basis of the measurements and empirical relation of Linow et al. (2017) and the assumption of Arthern and Wingham (1998). As climatic conditions, and therefore the surface grain size, vary among the investigated locations, this choice is a simplification. Owing to the use of the optimization approach, this parameter has a less significant effect, as it can be understood as a constant part of the grain radius, which is the same for every analyzed firn profile.

Figure 4 illustrates the range and distribution of surface boundary conditions at the investigated sites. The locations in Greenland show a higher mean surface temperature and surface mass balance than those in Antarctica. The surface density is higher at the Antarctic locations than at those in Greenland. Overall, a wide variety of typical climatic conditions for both ice sheets is covered.

## 3.3 Distribution and effects of input data

Ice core ngt03C93.2 (Fig. 1a) is an example of a high-resolution density measurement showing extensive small-scale layering. Only a few of the 159 firn profiles are of such high quality and include this type of layering. Although our proposed model works at high temporal and spatial resolution, it does not cover layering, as shown in Fig. 1a. The density profile retrieved at site 3 of the iSTAR traverse (Morris et al., 2017) (Fig. 3) illustrates that the model, if it is forced with data of higher temporal resolution, still does not cover the measured density variability. Small-scale layering of firn appears to be driven by a number of different processes (Hörhold et al., 2011). An extension of the model to cover such processes may be introduced in the

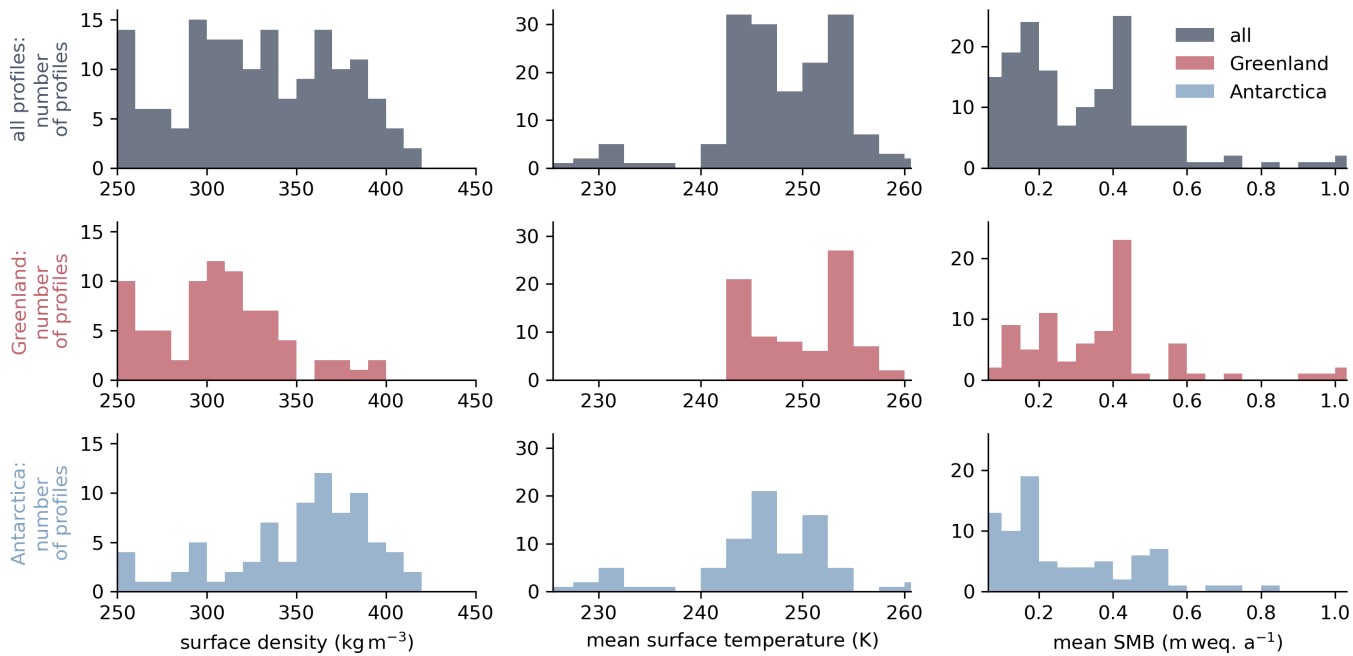

**Figure 4.** Distribution and comparison of boundary conditions at the 159 firn profile sites. All values are averaged over the simulation time for each location, beginning in 1958 and 1979 for Greenland and Antarctica, respectively, and ending at the date of measurement (see Sect. 3.2). The top row of plots (gray) shows the boundary conditions at all locations, whereas the middle and bottom rows, in red and blue, show only the locations in Greenland and Antarctica, respectively. Data for the surface temperature (center) and surface mass balance (right column) are provided by RACMO2.3 (Van Wessem et al., 2014; Noël et al., 2015).

future. We would prefer the approach of Freitag et al. (2013), who introduced the concept of impurity-controlled densification. Forcing data for this model are not globally available. However, the model in its current form does reproduce the mean density well, as demonstrated in Fig. 1a.

In this study, we focused on the initial stage of firn densification and the uppermost part of the firn column. We limited the model domain to a maximum depth of $-25\,\mathrm{m}$ below the surface. Furthermore, we did not develop the density further after the critical density of $\rho_c = 550\,\mathrm{kg\,m^{-3}}$ was reached. This choice raises the question of whether the use of a Neumann boundary condition set to zero at the profile base to solve for the temperature is justified for this particular model setup (Sect. A4). The critical density of $\rho_c = 550\,\mathrm{kg\,m^{-3}}$ is usually reached within the upper 10 m of the firn column. The temperature within this

domain is affected by surface conditions, which are covered by the forcing data. At greater depths, the temperature corresponds to the mean annual surface temperature and changes very little (e.g., Cuffey and Paterson, 2010, pp. 399 ff.). There have been few high-resolution temperature measurements of firn. Orsi et al. (2017) published a temperature profile of a 147 m length from borehole NEEM2009S1, which shows a temperature difference of little more than 1 K over the entire profile. Vandecrux et al. (2021) obtained the same result. This small temperature change at depths greater than approximately 10 m below the surface

has little effect on our model approach. To test the temperature dependence of the optimization approach, we performed the simulations again using the site-specific surface temperature forcing plus or minus $1\,\mathrm{K}$. The effect on the optimal factors $C_\mathrm{v}$ depends on the variant of the constitutive relation, but it is generally small. For example, the greatest difference between the optimal factors obtained using the correct and adjusted forcing for ice core ngt03C93.2 (Wilhelms, 2000) were found using Variant 1. This variant yields a value of $\Delta C_{\mathrm{v1,max}} = 0.02 \times 10^{-4}\,\mathrm{K\,s^2\,kg^{-1}}$, which corresponds to twice the sampling space

of the factors $C_{\mathrm{v_1}}$ used in the optimization. Variant 4 yields the same results. Because of this small effect overall, the stronger effect of the surface temperature in the investigated domain, and the limitation of the comparison to the domain actually affected by surface forcing, a Neumann boundary condition at the profile base is justified despite the low depth.

## 4   Results

Figure 5 shows the distribution of the RMSD calculated from the simulated firn profiles that best reproduce the measured

profiles. The four plots in the figure represent the four variants of the constitutive equation for grain boundary sliding [Eqs. (2) to (5)], as described in Sect. 2.2. Additionally, the median values of the data are shown. The distributions of the deviation for the four variants show little difference. The median values differ in a small range. Variants 2 and 3 show the smallest and largest values, respectively. The use of the modification introduced by Bréant et al. (2017) in the constitutive equation improves the agreement between the simulated and measured firn profiles by $6.2-6.8\,\%$. To put the values in perspective, the deviation

of the four best-fitting modeled firn profiles from ice core ngt03C93.2 (Wilhelms, 2000) displayed in Fig. 1 is approximately $28\,\mathrm{kg\,m^{-3}}$. That is, more than half of the simulations show even better agreement with the corresponding firn density profiles than this one. An overview of the deviation among all 159 measured firn profiles and the corresponding best simulation results can be found in the supplementary material.

    The range of factors contributing to the best-fitting firn profile is smaller for Variants 2 and 4 of the constitutive equation,

which use the modification of Bréant et al. (2017), compared to Variants 1 and 3, as shown in the box plots in Fig. 6. In contrast to factors $C_{\mathrm{v_1}}$ and $C_{\mathrm{v_2}}$, factors $C_{\mathrm{v_3}}$ and $C_{\mathrm{v_4}}$ incorporate the Arrhenius factor from the original description of grain boundary sliding given by Alley (1987). Therefore, direct comparison between the two groups of factors is not possible. The quartile coefficient of dispersion of the four variants, which is shown on the right side of Fig. 6, is a relative measure of the scatter of the values. The coefficient reveals that the factors obtained for Variants 1 and 2 are defined in a narrower range than those of

Variants 3 and 4. All four sets of factors show slightly nonuniform distributions that tend toward smaller values.

    To check for possible mean surface temperature dependence of the 159 factors found by optimization, these values are plotted against each other in Fig. 7. The values of the mean surface temperature were calculated from the forcing data for each firn profile site. The left plot illustrates the results for Variants 1 and 2 in red and blue, whereas the right plot shows the factors for Variants 3 and 4 using green and purple markers, respectively. The legend shows the Pearson correlation coefficient $r_{\mathrm{Pearson}}$,

a measure of the linear correlation of the two variables, and the distance correlation $dCor$ (Székely et al., 2007). The distance correlation was designed by Székely et al. (2007) to overcome problems with the Pearson correlation coefficient. It describes

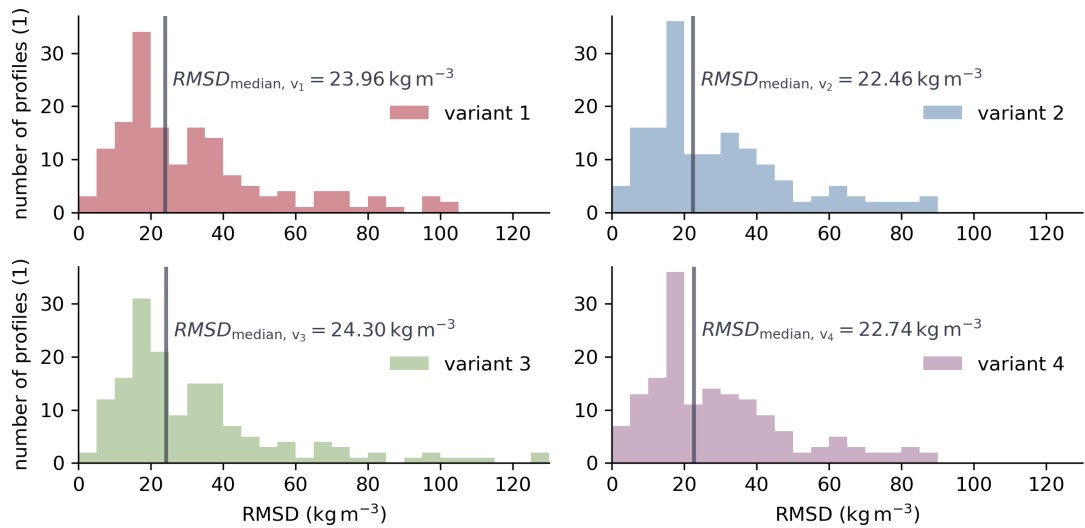

**Figure 5.** Distribution of smallest RMSD for every analyzed firn profile found using the optimization scheme outlined in Sect. 2.2. The four plots show the results for the four tested variants of the constitutive equation. Vertical lines indicate the median of the 159 values.

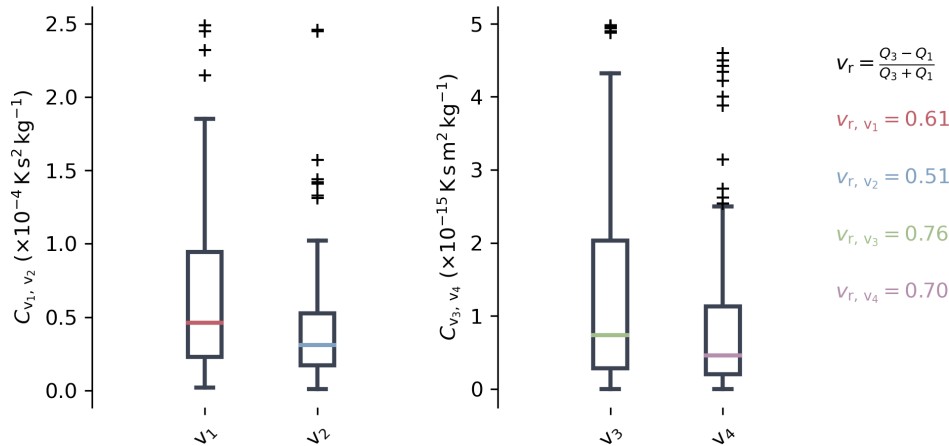

**Figure 6.** Box plots showing the distribution of the best factors, $C_{v_1}$ to $C_{v_4}$, for the four variants of the constitutive equation describing grain boundary sliding [Eqs. (2) to (5)] derived using the optimization scheme described in Sect. 2.2. Quartile coefficient of dispersion $v_r$ of each variant is shown on the right in the corresponding color. The quartile coefficient of dispersion was calculated using the first ($Q_1$) and third ($Q_3$) quartile values of the data sets, which are shown in black and represent a robust relative measure of dispersion.

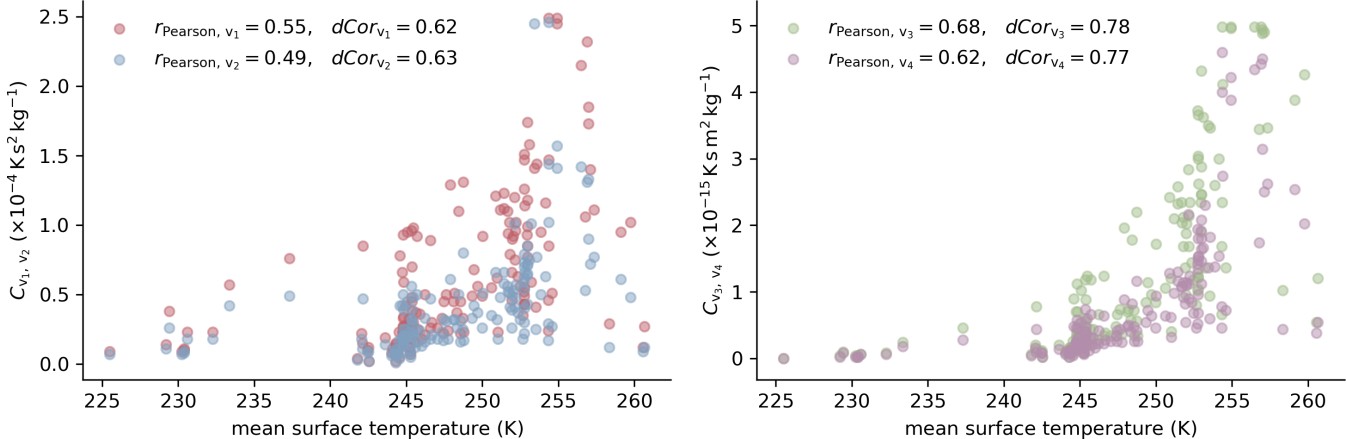

**Figure 7.** Best factors for every investigated firn profile determined using the optimization scheme and plotted against mean surface temperature during forcing period (see Sect. 4). Left: results for Variants 1 and 2 of the constitutive equation in blue and red, respectively. Right: results for Variants 3 and 4 in green and purple, respectively. The Pearson correlation coefficient $r_{\text{Pearson}}$, which represents the linear correlation between factors $C_{v_1}$ to $C_{v_4}$ and the mean surface temperature, as well as the distance correlation $dCor$, are given in the legend.

the correlation of two vectors and is not limited to linear dependence. It is defined between zero and one, where zero indicates the independence of the variables.

The correlation between the resulting factors and the mean surface temperature is higher compared to the correlation with other properties. This statement is especially true for factors $C_{v_3}$ and $C_{v_4}$ of Variants 3 and 4, respectively. However, the Pearson correlation coefficient indicates only a linear correlation. The scatter of factors $C_{v_3}$ and $C_{v_4}$ with respect to mean surface temperature might resemble a higher-order function, as indicated by the higher values of the distance correlation.

The values of the Pearson correlation coefficient and distance correlation shown in Fig. 8, which shows the factors with respect to mean surface mass balance, are higher than those in Fig. 7.

Like the mean surface temperature, the mean surface mass balance was calculated from the forcing data used during the simulations. The correlation coefficients for the results obtained using Variants 3 and 4 exceed those for the results obtained using the variants of the constitutive equation incorporating the Arrhenius factor explicitly. Variant 3 shows the best indication of a relationship between surface mass balance and the factor yielding the density profile that best reproduces the corresponding field measurement.

In Fig. 8, a striking feature appears around a mean surface mass balance of $0.4\,\mathrm{m\,weq.\,a^{-1}}$. The values of the factors yielding the simulated density profiles that best reproduce the measured profiles show a wide range for all four variants of the constitutive relation. These profiles are part of a study in western Greenland by Harper et al. (2012). The study region is relatively small, which explains the similar climatic conditions. Although the mean annual surface mass balance is positive, melting occurs throughout the year and affects the firn density.

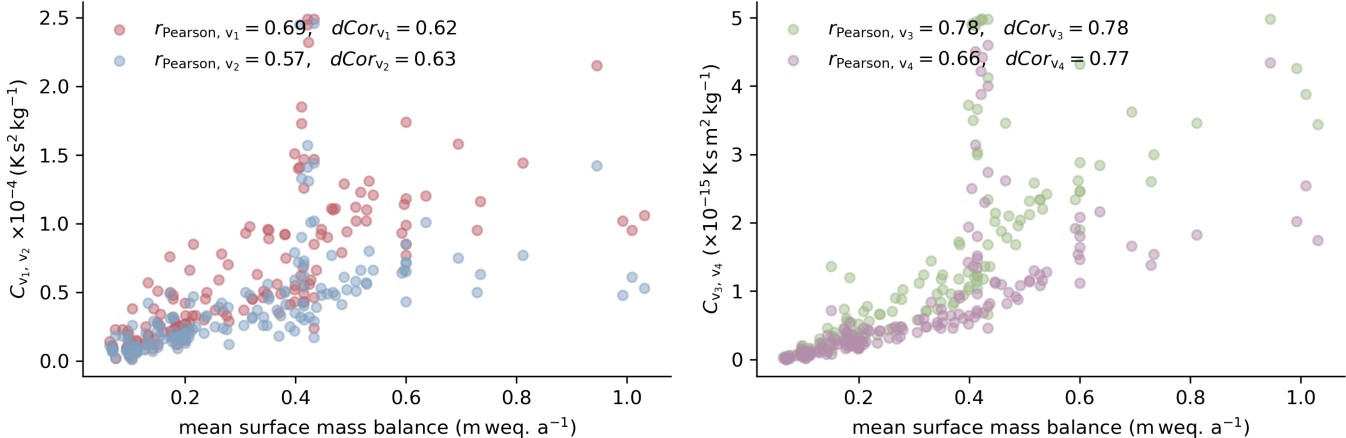

**Figure 8.** Factors obtained by optimization versus mean surface mass balance, calculated for 159 firn profiles from the forcing. Left: best-fitting values for Variants 1 and 2 (factors $C_{v_1}$ and $C_{v_2}$, respectively). Right: best-fitting values for Variants 3 and 4 (factors $C_{v_3}$ and $C_{v_4}$, respectively). Linear correlation between the factors and mean surface mass balance is indicated by the Pearson correlation coefficient $r_{Pearson}$, and the general correlation is indicated by the distance correlation $dCor$.

## 5 Discussion

Using the four variants of the constitutive relation of Alley (1987) [Eqs. (2) to (5)], we generated density profiles that were in good agreement with most of the 159 density measurements. Uncertainties may result from the forcing, limited knowledge of the initial grain radius, and the poor constraint on the density at the surface. As measured firn densities represent past climate conditions, unrealistic forcing always results in a mismatch between the simulated and measured firn properties, independent of the optimization approach and physical model. However, when the proposed optimization scheme was used, the differences between the simulated and measured density profiles show distinct minima, indicating that the forcing represents the climatic history of the firn profiles, in principle.

The optimization scheme produces site-specific values of the factors $C_{v_1}$ to $C_{v_4}$. As the optimization process is unique to each analyzed site and constitutive relation variant, the four simulated density profiles for each site are very similar, as illustrated in Fig. 5. This feature allows us to compare the factors obtained using the four variants. The differences between the factors do not arise primarily from differences in the simulated density profiles, but reflect the differences in the variants; consequently, the results are almost the same.

However, owing to the nature of the optimization scheme, possible errors in the forcing data or other parameters such as the activation energy used to calculate the grain radius (see Eq. A11) are also included in the site-specific optimal factors $C_{v_1}$ to $C_{v_4}$. If, for example, the surface temperature forcing at a site consistently deviates by 5 K during the simulation, this error can be compensated by adjusting the specific optimization factor $C_v$ (see Sect. 3.3). However, the large number of analyzed firn profiles compensates for such random errors. Systematic errors in the forcing data van Wessem et al. (2018) cannot be

identified. Therefore, improving the temporal resolution of the forcing and covering longer periods could result in better and more detailed simulation results, as shown in Fig. 3.

This study analyzed only dry firn densification. The current model cannot handle melting. We accommodate this feature by setting the annual mean surface mass balance at the investigated sites to be strictly positive (Sect. 3.1). However, this limitation means that we cannot ensure that no melting occurs over the course of a year. The results shown in Fig. 8 illustrate how this limitation affects the optimization results. The limitation is problematic, especially in recent years, when melting occurred over almost the entire Greenland Ice Sheet (e.g., Nghiem et al., 2012). The simulation of meltwater percolation through the

firn and its interaction with firn densification is important, especially in the upper part of the firn body (e.g., Vandecrux et al., 2020). The proposed method could be improved by the application of this model approach in future investigations. However, we identified some correlations between the optimization results and the surface mass balance.

        Note, again, that we have not investigated whether grain boundary sliding is indeed the dominant process during the first stage of firn densification. We assess whether a process with a functional dependence on density, firn overburden stress, temper-

ature, and grain radius represents the observed density profiles well. Any other deformation process with the same functional dependence would be equally well suited. Nevertheless, by maintaining the general structure of the constitutive relation of Alley (1987), we conclude that this description of grain boundary sliding is a good basis for a physically based model describing firn densification up to the critical density, $\rho_c = 550 \, \mathrm{kg \, m^{-3}}$.

        Comparing the results for Variants 1 and 2, as well as those for Variants 3 and 4, we found that an adjustment of $\left(1 - \frac{5}{3} \rho / \rho_{\mathrm{ice}}\right)$

to $\left(1 + \frac{0.5}{6} - \frac{5}{3} \rho / \rho_{\mathrm{ice}}\right)$ results in better reproduction of the measured density profiles. This result must be reviewed carefully in terms of the study design and the background of a physics-based model describing firn densification. As Alley (1987) pointed out, grain boundary sliding might be the dominant process driving firn densification at low densities, but it is presumably not the only one. The constitutive law of Alley (1987) is designed such that the densification due to grain boundary sliding becomes zero at a density of $\rho_c = 550 \, \mathrm{kg \, m^{-3}}$, owing to the densest packing of spheres obtained at that density and increasing

accommodation incompatibilities. The modification of Bréant et al. (2017) changes this behavior such that grain boundary sliding vanishes at a density of approximately $\rho_c^* = 596 \, \mathrm{kg \, m^{-3}}$, which could have advantages for the transition into the next stage of firn densification. We suggest that a simultaneous decrease in grain boundary sliding and increase in one or more other processes would provide a good characterization. Specifically, dislocation creep drives densification at higher density (Maeno and Ebinuma, 1983) because of increasing stresses. The onset of dislocation creep at densities below $\rho_c = 550 \, \mathrm{kg \, m^{-3}}$ would

not necessarily affect the entire bulk firn matrix, but increasing volume fractions of the porous matrix.

        Variants 1 and 2 of the constitutive relation of Alley (1987) incorporate the Arrhenius factor for boundary diffusion, $D_{\mathrm{BD}}$, from the description of the bond viscosity given by Raj and Ashby (1971). In the formulation of Variants 3 and 4, we neglected the Arrhenius factor. As shown in Fig. 5, the difference in the resulting RMSD is small regardless of whether $D_{\mathrm{BD}}$ is considered. This result is reasonable, as we determined individual factors $C_{\mathrm{v}}$ for each site. The similarity of the results allows us to compare

the factors $C_{\mathrm{v}_1}$ and $C_{\mathrm{v}_2}$, which were determined using the Arrhenius factor $D_{\mathrm{BD}}$, to factors $C_{\mathrm{v}_3}$ and $C_{\mathrm{v}_4}$, which were obtained using the variants of the constitutive relation without the Arrhenius factor. As $D_{\mathrm{BD}}$ is a function of temperature, it is reasonable to examine the dependence of the factors on the mean surface temperature, which is shown in Fig. 7.

The factors $C_{v_3}$ and $C_{v_4}$, which were determined using the variants without $D_{BD}$, show a stronger dependence on the mean surface temperature than the factors $C_{v_1}$ and $C_{v_2}$. In addition, the factors $C_{v_1}$ and $C_{v_2}$ show less dispersion than $C_{v3}$ and $C_{v_4}$, as shown in Fig. 6. The inclusion of $D_{BD}$ in the constitutive relation improves the determination of these factors. It is therefore a meaningful description within the constitutive relation. Although the inclusion of the Arrhenius factor improves the determination of $C_{v_1}$ and $C_{v_2}$, we still see some dependence on the mean surface temperature. A better determination of the parameters of the Arrhenius factor may resolve this dependence. If this is not the case, another dependence on the temperature may be introduced to improve the constitutive relation for grain boundary sliding.

We interpret the dependence on surface mass balance as indicating that the load is currently not represented well. The stress is represented by a second¬-order tensor. A firn column represented by a one-dimensional modeling approach would be surrounded by neighboring firn columns, resulting in lateral confinement that limits deformation in the horizontal direction. The horizontal components of the stress tensor are not zero. As firn is a compressible material, the determination of these horizontal stress components is not trivial. The frequently used term "overburden pressure" is misleading, as the mechanical pressure is defined as the spherical part of the Cauchy stress tensor (e.g., Haupt, 2002, p. 301) and is not, in general, identical to the normal stress in the vertical direction. With increasing depth, the magnitude of the horizontal stress components and their effects would increase. Modeling approaches that consider the full stress tensor were reported by Greve and Blatter (2009), Salamantin et al. (2009), and Meyer and Hewitt (2017). It might be worth using the constitutive relation for grain boundary sliding of Alley (1987) in such a modeling context. This approach will not necessarily result in a full three-dimensional model, as the problem can be formulated assuming axial symmetry, which would require adjustment of the constitutive relation. A more extensive interpretation of the factors that yielded the best agreement is therefore challenging. The determination of a single factor $C_v$ for one or all of the variants is not useful. It would not result in better simulation results compared to other published firn densification models. The site-specific values of the factors determined using the presented optimization approach simply show the differences in the variants of the constitutive relation.

## 6   Conclusions

Using variants of the constitutive relation for grain boundary sliding of Alley (1987) and an efficient optimization scheme, we reproduced 159 firn density profiles reasonably well. Thus, we conclude that the description of grain boundary sliding introduced by Alley (1987) is appropriate for the simulation of firn densification at low density.

The modification of the constitutive relation of Bréant et al. (2017) results in slightly better simulation results when only the first stage of firn densification is considered. Further factors, including the transition from the first to the second stage, must be considered to answer questions regarding in which domain and to what extent different processes drive firn densification.

In our optimization approach, we use a single factor representing various model parameters and search for the factor value resulting in the best agreement between the simulated and measured firn profiles. Consequently, the site-specific simulation results are independent of the possibly deficient model parameters, which are considered collectively. It is not possible to derive a distinct value for the factor representing the climatic conditions at all locations of the investigated firn profiles. We found a

linear dependence of the factors on the site-specific surface mass balance (Fig. 8). By contrast, we did not find a clear linear dependence on temperature (Fig. 7), but the results show that a site-specific parameter is required.

As the amount of surface accumulation affects the load conditions, we assume it is not represented well in the model. Unlike other firn densification models, the physical simulation of grain boundary sliding does not depend directly on the surface mass balance, but depends on the actual stress. Further interpretation of the obtained factors is difficult using the presented simulation setup. The description of grain boundary sliding given by Alley (1987) could be improved by using a higher-dimensional approach including the horizontal components of the stress tensor. Modeling approaches of this kind include those of Greve and Blatter (2009), Salamantin et al. (2009), and Meyer and Hewitt (2017).

We would like to emphasize that optimization of any type is possible only because of the enormous efforts of the SUMup team (Koenig and Montgomery, 2019), which has made a vast amount of firn core data available. These data are strategically crucial for advances in firn densification modeling, reinforcing the recommendations of FirnMICE (Lundin et al., 2017) for enhanced efforts toward physically based models.

*Code availability.* The code used to simulate firn profiles will become available via github.com and gitlab.com when this manuscript is published.

## Appendix A: Model description

### A1  Numerical treatment of densification

All model equations are solved on an adapting one-dimensional grid that is updated at every time step. The approach is based on an updated Lagrangian description, where the update velocity of the grid is the material flow velocity. This results in material-fixed coordinates. The Lagrangian-like description affords very high spatial and temporal resolution in the simulations. It can be shown that by integrating the local Eulerian form of the mass balance in one dimension over a material control volume with moving boundaries $z_1(t)$ and $z_2(t)$, we obtain (Ferziger and Perić, 2002, p. 374)

$$\frac{d}{dt} \int_{z_1(t)}^{z_2(t)} \rho \, dz + \int_{z_1(t)}^{z_2(t)} \frac{\partial}{\partial z} \left( \rho \left( v - v_b \right) \right) dz = 0 \,. \tag{A1}$$

Here $\rho$ represents the density, $z$ is the vertical coordinate, $t$ is the time, and $v$ is the material flow velocity, whereas $v_b$ represents the grid velocity or the velocity of the integration boundary. When the grid velocity equals the material flow velocity ($v_b = v$), the second term of Eq. (A1), which describes the advection, vanishes. The resulting equation is equal to the Lagrangian form of the mass balance (Ferziger and Perić, 2002, p. 374). On a one-dimensional grid consisting of a number of grid points, as illustrated in Fig. A1, we denote the grid point velocity as $v_b$, which is equal to the flow velocity $v_b \equiv v$. The locations of all grid points are updated at each time step by integrating the grid point velocity $v_b$ using a forward Euler scheme. Thus, advection is represented entirely by the adapting grid.

The grid point velocity $v_b$ is calculated using the constitutive equation for grain boundary sliding, as described in Sect. 2.1, and the definition of the strain rate in one dimension. The description of grain boundary sliding provides the strain rate in the vertical direction along the grid, $\dot{\varepsilon}_{zz}$, as a function of vertical stress $\sigma_{zz}$, density $\rho$, temperature $T$, and grain radius $r$

$$\dot{\varepsilon}_{zz} = f\left(\sigma_{zz}, \rho, T, r\right) = \frac{\partial v}{\partial z} = \frac{\partial v_b}{\partial z}\,. \tag{A2}$$

The strain rate of a material line element can be defined as the spatial derivative of the velocity, as shown in Eq. (A2) (Haupt,
2002, pp. 32–38). On a one-dimensional grid, which is defined as a number of grid points, the space between neighboring grid points can be considered as a material line element (see Fig. A1). Therefore, the grid point velocity $v_b$ is easily computed by integrating the strain rate $\dot{\varepsilon}_{zz}$ in the vertical direction along the length of the grid cell:

$$v_b = \int_{z_1}^{z_2} \dot{\varepsilon}_{zz}\, dz\,. \tag{A3}$$

To implement Eq. (A3), an integration constant determined by a suitable boundary condition is needed. It is reasonable to apply
a Dirichlet boundary condition that forces the grid point velocity $v_b$ to be zero at either the top or the base of the computational domain to represent a fixed reference point at the top or the base of the modeled firn profile, respectively. All other points defining the adapting grid are moving with respect to this anchor point. In this study, we placed the anchor point at the base of the simulated firn profiles ($z_0$ in Fig. A1). The depth coordinates of the profiles shown in the figures below were adjusted for better readability.

To represent accumulation at the top of a simulated firn profile, an inflow boundary condition must be implemented. To this end, at each time step an additional grid point is added at the top of the grid. Its coordinate within the grid $z_{n+1}(t + \Delta t)$ is calculated as

$$z_{n+1}(t + \Delta t) = z_n(t + \Delta t) + \Delta t\, a_0(t)\, \frac{\rho_{\text{water}}}{\rho_0}\,, \tag{A4}$$

where $a_0$ is the accumulation rate given as the $m\,s^{-1}$ water equivalent, $\Delta t$ is the length of the time step, $\rho_{\text{water}}$ is the density of
water, and $\rho_0$ is the density of the deposited snow. The position of a new grid point ($z_{n+1}(t + \Delta t)$ in Fig. A1) is the position of the firn surface at the last time step $z_n$ plus the thickness of the firn layer deposited during the last time step. This thickness is defined by the time step $\Delta t$ and the time-dependent accumulation rate $a_0(t)$. We use an accumulation rate given in the unit of meter water equivalent per second, which must be converted to the unit of meter firn equivalent taking the site-specific surface firn density $\rho_0$ and water density $\rho_{\text{water}}$ into account.

In this process, the number of grid points increases. Therefore, grid points are removed from the model domain at its base when a maximum number is reached.

## A2  Stress

To evaluate the stress in the vertical direction, $\sigma_{zz}$, we use the local form of the static linear momentum balance in its Eulerian description. We neglect the acceleration component, as the changes in velocity are assumed to be small; thus, we have

$$\frac{\partial \sigma_{zz}}{\partial z} + \rho g = 0\,. \tag{A5}$$

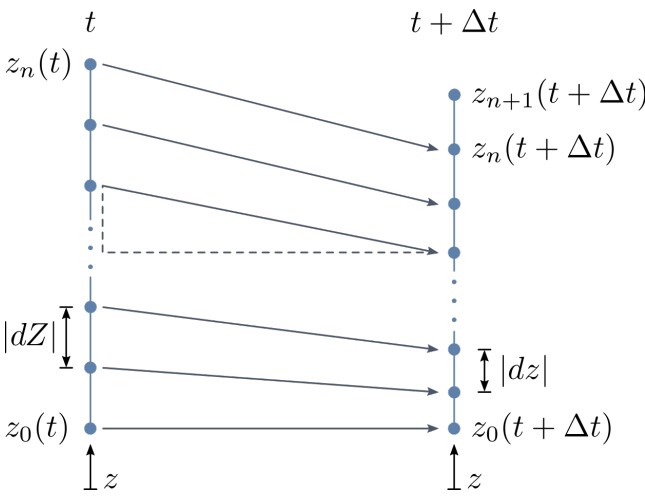

**Figure A1.** Principle of grid evolution. Left: grid at time $t$. Right: updated grid at time $t + \Delta t$. The grid points move at the grid point velocity $v_b$, which is equal to the material flow velocity $v$. At time $t + \Delta t$, an additional grid point $z_{n+1}$ representing accumulation is added. Distances between neighboring grid points can be understood as material line elements $|dZ|$ and $|dz|$ in the reference and in the current configuration, respectively.

The stress $\sigma_{zz}$ can be easily calculated by integrating the product of density $\rho$ and acceleration due to gravity $g$ along the simulated profile. We assumed that the surface of the profile is traction-free.

## A3  Density

As noted in Sect. A1 and illustrated by Eq. (A1), the change in density integrated over a control volume with respect to time must be zero. That is, the mass in a control volume cannot change. As the positions of the grid points, and thus the material control volume, do change, the density changes accordingly. The evaluation of the density integral over a control volume at two time steps yields

$$\rho(t)\left(z_2(t) - z_1(t)\right) = \rho(t + \Delta t)\left(z_2(t + \Delta t) - z_1(t + \Delta t)\right). \tag{A6}$$

As the space between two neighboring grid points can be understood as a material line element (Haupt, 2002, pp. 32–38), Eq. (A6) can be rewritten in the form of Eq. (A7), where $|dZ| = |z_2(t) - z_1(t)|$ and $|dz| = |z_2(t + \Delta t) - z_1(t + \Delta t)|$ are the lengths of a material line element in the reference configuration and its image in the current configuration, respectively:

$$\rho(t)|dZ| = \rho(t + \Delta t)|dz|. \tag{A7}$$

By rearranging Eq. (A7), we obtain the formulation of the density change with time depending on the definition of the strain $\varepsilon_{zz}$ in one dimension (Haupt, 2002, p. 34):

$$\rho(t + \Delta t) - \rho(t) = -\rho(t + \Delta t)\left(\frac{|dz| - |dZ|}{|dZ|}\right) = -\rho\,\varepsilon_{zz}. \tag{A8}$$

The evolution of the density can therefore be computed by integrating the strain rate $\dot{\varepsilon}_{zz}$ (Sect. 2.1) over time.

## A4 Temperature

As noted in Sect. A1, all advection in the model domain is represented by the moving grid. Therefore, the description of the temperature evolution reduces to simple heat diffusion:

$$\rho c_p \left(\frac{\partial T}{\partial t}\right) + \frac{\partial}{\partial z}\left(k(\rho)\frac{\partial T}{\partial z}\right) = 0.\tag{A9}$$

Following Paterson (1994), we assume a constant heat capacity of $c_p = 2009\,\mathrm{J\,kg^{-1}\,K^{-1}}$, and following the example of Zwinger et al. (2007), we assume the density-dependent thermal conductivity described by Sturm et al. (1997) as

$$k(\rho) = \left(0.138\,\mathrm{W\,m^{-1}\,K^{-1}}\right) - \left(1.010 \times 10^{-3}\,\mathrm{W\,m^3\,kg^{-1}\,K^{-1}}\right)\rho + \left(3.233 \times 10^{-6}\,\mathrm{W\,m^5\,kg^{-2}\,K^{-1}}\right)\rho^2.\tag{A10}$$

The temperature profile is initialized using a constant mean value computed from the site-specific surface forcing. To solve for temperature, a Neumann boundary condition is used at the lower boundary at the profile base. The vertical gradient of the temperature is set to be zero.

## A5 Grain radius

Alley (1987) used measured grain size data to fit his simulation results to four firn profiles. As information about grain size is sparse, we use a modeling approach to describe the grain radius. The evolution of the grain radius $r$ is simulated using the well-known description of Stephenson (1967) and Gow (1969), as given by Arthern et al. (2010). Stephenson (1967) and Gow (1969) described the grain size in terms of the mean cross-sectional area. By contrast, Arthern et al. (2010) assumes the mean cross-sectional area to be $A = (2/3)\pi r^2$ and formulates the grain growth rate as

$$\frac{\partial r^2}{\partial t} = k_0 \exp\left(-\frac{E_g}{RT}\right).\tag{A11}$$

This formulation enables simple calculation of the grain radius $r$. The values of the activation energy, $E_g = 42.4\,\mathrm{kJ\,mol^{-1}}$, and prefactor, $k_0 = 1.3 \times 10^{-7}\,\mathrm{m^2\,s^{-1}}$, of the Arrhenius factor are based on data reported by Paterson (1994) and were also adapted from Arthern et al. (2010). Unlike Arthern et al. (2010), we do not use the mean annual temperature but the actual temperature $T(z,t)$ along the simulated profile. To solve Eq. (A11), suitable boundary conditions must be provided. We used a constant surface grain radius. See Sect. 3.2 for further information on the boundary conditions.

## A6 Age

For comparison (Sect. 3.2) and general interest, the firn age $\chi$ was also simulated. Because advection is represented by an adapting grid, the description is very simple. It is calculated as

$$\frac{\partial \chi}{\partial t} = 1.\tag{A12}$$

Newly deposited snow has an age of zero, which is represented by a Dirichlet boundary condition. The age discretized at the grid points then increases according to the time step.

## A7    Time discretization

Time is discretized using constant time steps. For this study, a value of 48 time steps per year was found to be a good compromise between economical simulation and desirable resolution and was used in all simulations. The grid resolution depends on the time step, as a new grid point is generated at each time step to represent accumulation, as described in Sect. A1 and shown in Eq. (A4). Time-dependent properties such as density, temperature, grain radius, and age were developed using an explicit Euler scheme.

*Author contributions.* T.S. developed the numerical approach and code and conducted and analyzed all simulations. All authors jointly developed the concept of the modeling approach, discussed the results, and wrote the manuscript.

*Competing interests.* We declare that there are no competing interests.

*Acknowledgements.* This project was funded by the German Research Foundation in the Priority Program SPP1158 under project number 403642112. We acknowledge the fantastic community effort SUMup, which provides a database of firn core data. We thank Sepp Kipfstuhl (AWI) and Johannes Freitag (AWI) for discussions on firn densification.

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
