# Peer review of "On the Contribution of Grain Boundary Sliding to Firn Densification - an Assessment using an Optimisation Approach"

_The Cryosphere, 2021_

## Referee Comment (RC1)

**On the contribution of grain-boundary sliding to firn densification - an assessment using an optimisation approach. Schultz et al**

**1 General Comments**

This paper revisits a physics-based model for the densification of dry snow by grain-boundary sliding (Alley, 1987). The model contains a number of parameters which can, in principle, be determined *a priori* and requires a knowledge of temperature, accumulation rate and grain size in order to calculate the densification rate in the upper part of a firn layer. Alley tested his model on 4 profiles, starting at 2 m depth, and optimised one parameter, the grain-boundary viscosity, $\nu$, for each profile. He then found that the local best values for $\nu$ fitted an Arrhenius law with an activation energy of $41 \pm 2$ kJ mol$-1$. The 1987 paper does not show explicitly how well the model works if values of $\nu$ are calculated using this global equation, but the results given look promising. It is certainly very worthwhile for the authors to use the mass of new density profile data available from the SUMup data base to make a more complete assessment of the Alley model.

In general the presentation is clear, although the English needs some editing, and the results are potentially of interest to others working in the field. However, the paper could be much improved by a deeper discussion of the assumptions made and the significance of the results. Some suggestions follow:

(1) The SUMup data base is an on-going community effort which sets a very clear condition for use of the data. The guidelines state "When using this data set please cite both the individual researchers who provided the data as listed in the Citation column as well as the SUMup dataset". There is a citation for one profile in this paper (ngt03C93.2, shown in a figure) but there are no citations for the 158 other profiles used. This may seem like a tedious task, but it is essential that researchers are properly acknowledged. It may be that the editor will accept a table of profiles with citations as Supplementary Material, but whether this is adequate needs to be confirmed with the SUMup team.

(2) The authors have constructed a numerical model for the evolution of temperature, density and grain radius in firn using the Alley densification equation and forced by annual values of surface temperature and accumulation rate from the RACMO2.3 meteorological model. The advantages of using the RACMO data are (i) that profiles for which *in situ* climate data are not available can be simulated and (ii) that inter-annual climate variations can be taken into account. The disadvantages are that (i) local data have to been found by interpolation and (ii) systematic errors may be introduced. It would be helpful if the authors could provide an assessment of how accurate the forcing data are likely to be.

(3) Since the input temperature is only available at yearly intervals, the annual variation of temperature in the top layer of the firn cannot be simulated. Alley dealt with this problem by excluding the top 2 m of the firn, which experiences the greatest temperature fluctuations. In l.203 the authors state that they

optimise in a domain "bounded by the surface and the oldest horizon within the profile affected by the forcing". In fact some records, including ngt03C93.2, shown as an example in the paper, do not extend to the surface. I assume the authors actually use an upper bound as near to the surface as possible. In any case, it is quite important to discuss why they choose not to follow Alley's example. The reader will need some convincing that inclusion of data from a region with strongly-varying temperature does not affect the optimised parameter values.

(4) One of the factors in the Alley equation ($\alpha'$) describes the effect of spherical averaging of the sliding velocity along variously-oriented grain boundaries to obtain the mean vertical velocity. Based on geometric arguments Alley suggests $\alpha' = (1 - N/6)$ where $N$ is the coordination number and notes that, by observation, $N \approx 10\rho/\rho_i$. The authors make the interesting comment that $\alpha' \to 0$ for $\rho = 550$ kg m$^{-3}$; in other words the Alley equation implies that grain boundary sliding must stop at this density. An alternative expression for $\alpha'$ from Bréant implies that grain boundary must stop at a higher density of 596 kg m$^{-3}$. The authors choose to restrict their optimisation domain to densities below 540 kg m$^{-3}$ " due to the asymptotic characteristic of the resulting density profiles using the... constitutive equation" (l.209). But the reader might wonder why the cut-off should not come when densification by grain-boundary sliding is no longer much greater than densification by other processes (e.g. dislocation creep). There is a brief mention of the possibility of competing processes in the discussion section of this paper (l. 328) but this comes only as a suggestion for future consideration. I think the reader would appreciate a much deeper discussion of the implications of this choice of cut-off density, earlier in the paper.

(5) Another problem arises because the authors have not chosen to discuss the question of layering in firn density profiles, even though their example profile clearly shows large fluctuations in density superimposed on the general increase in density with depth. They use a microscale, physics-based model, which applies to a small element of snow, and a grid spacing of 48 points per annual layer, so it would be possible to investigate the effect of annual layering. The authors may choose not to do this, but it is important then to bring out their underlying assumption, which is that the Alley equation can be applied on the macroscale by simple substitution of the macroscale mean density. Given the work that has gone into writing the model and selecting profiles to simulate, it would a pity not to take the opportunity to include some discussion about the problem of up-scaling.

(6) In the paper 4 versions of the Alley model are tested by optimising one parameter $C_\nu$ per version. The reader will want to know how the values of $\nu$ implied by values of $C_{\nu 1}$ compare with the Alley values and also what the individual values of the cost function RSMD are. These could be included in a table of sites, input data and results in the Supplementary Material.

(7) The authors conclude that their results show that "the description of grain boundary sliding introduced by Alley (1987) is suitable for the simulation of firn densification at low density" (l. 362). In fact this would only be true if the value of parameter $C_\nu$ could be specified *a priori*. The authors do not

say that the linear relations with mean annual temperature and accumulation shown in Figures 7 and 8 could be used to determine global values of $C_\nu$ but, if this is what they mean, then we need to know what the cost functions are when profiles are simulated using these global values.

(8) On the whole the reader is left with the impression that the Alley model is inadequate, because the supposedly-fixed parameter $C_\nu$ varies with the input conditions. This may be somewhat unfair. The variation may arise because the model has been applied too near the surface, layering has been ignored or indeed, as the authors remark, the input meterological data are inaccurate. But if there is a real dependence on accumulation rate this can be traced back to the calculation of stress $t_{zz}$ (Equation (6)). Do the authors think that their results show that the vertical grain sliding velocity is not linearly related to the vertical force on the grain? Could this be because other non-linear processes are acting as well as grain-boundary sliding? Or is the Alley (1987) analysis too simple?

**2    Other Comments**

- l.2. "first introduced by Alley (1987)" Perhaps better to say that the theory of grain-boundary sliding in snow was developed by Alley in this paper? The concept that this process might occur was already in the literature.

- l.15. "a couple of research fields" implies two fields, whereas the authors list three

- l.20 "The first and greater (category)". This sounds like a value judgement; better to say that the majority of models fall in the first category

- l.22 "this approach neglects". Do these models really neglect overburden stress or is it implicit rather than explicit in the model equations?

- l.32 onwards "which applies for firn" . The paragraph goes on to say the theory does **not** apply to firn below the critical density. This is confusing. Does the theory not apply because the snow is not compacted enough? Or because the grain geometry is not regular?

- l.40. Theile et al (2011) "tried to point out" or "did point out"?

- l.43 Are these 4 profiles also simulated in this paper? Which are they?

- l.101 m w.e.a$^{-1}$ is used elsewhere; why does m w.e. s$^{-1}$ have to be used here?

- l.107. Why does 2/15 seem an arbitrary number? To say it results from the geometric deviation does not really explain to the reader where it comes from. Since the paper is based on the Alley model, expanding the derivation would be worthwhile. Then the reader would understand the physics behind the model without having to go back to the original paper

- l.109. Is the word "resembles" right here? Maybe "represents"?

- l.128. Is "test volume" right? Maybe "volume element"?

- l.150 Would it be worth pointing out that Alley used observed values of $r$?

- l.200 This sounds as if the authors do not think the Lundin and Verjans papers use an objective cost function. The sentence could be altered so it is clear that the authors are explaining that they have used a different objective cost function.

- Figure 2. Is 2(c) a true representation of the input to the model? Or is the accumulation constant for a given year?

- l.279. It is not clear here whether the lower boundary for the simulation is the same as the lower boundary for the domain used to calculate the deviation between observed and modelled densities. The lower boundary for the simulation needs to be deep enough for the temperature to be constant, whatever the temperature variation at the surface.

- l.278 The difference in surface density between Greenland and Antarctic profiles merits further discussion. To say merely it is "plausible" misses an opportunity to comment on different types of snow (see Salamatin for example) and whether the densification rates might be different for Greenland and Antarctica.

- l.320 This would be a good point to tell the reader whether using the Alley model, with one parameter optimised, gives a better fit than using a semi-empirical model (e.g. Herron and Langway) with the densification rate optimised. And " a reasonable good match" needs to be defined in terms of the RSMD

- l.330 This paragraph should come much earlier in the paper because it is key to understanding the physics of the densification problem. The ideas expressed are well-established in the literature, so should be reviewed before the authors proceed to describe their work. It would then be easier to discuss the choice of lower boundary condition, the role of layering and other factors in the light of current understanding.

- l.353 The question of whether horizontal divergence affects the creep densification rate was raised by Alley in the 1980s. It is not clear here whether the authors are suggesting that basal-sliding is also affected

- l.371. This paragraph should come earlier, when the input data are introduced, so the reader knows their limitations from the start

---

## Referee Comment (RC2)

Review of "On the Contribution of Grain Boundary Sliding to Firn Densification – an Assessment using an Optimisation Approach"
by Schultz et at.
* * *
**Summary and general comments:**

This paper presents work using the framework presented by Alley (1987) to investigate the common assumption that near-surface firn densification is driven by grain-boundary sliding. The authors use an optimization approach to find the best coefficients for four variants of their model at 159 different sites that have firn depth-density data in the SUMup dataset. The authors rightly identify the importance of improving our ability to model firn for numerous problems in glaciology, and I applaud them for creatively leveraging the extensive SUMup database in a way that has not previously been done.

I believe this work holds promise to provide a worthy contribution to *The Cryosphere*, but I recommend major revisions prior to its publication. The paper is very heavy on its description of the model development and methods used, but the analysis of the results and their implication on the broader firn modeling/glaciological community are spare. I found a number of their analyses to be unconvincing and assumptions to be not fully explained and/or justified. (e.g. their assumption of linear relationships between their parameters and temperature and surface mass balance). Additionally, the manuscript needs significant editing to improve its structure and clarity – it has many grammatical errors and heavily uses multi-clause sentences that muddle the authors' points. I have noted some (but not all) of these in my specific comments below.

A few general comments:
- The manuscript purports to be based on the grain boundary sliding model presented by Alley (1987), but the authors wrap all of the grain structure parameters from Alley into a single parameter C, which makes their model quite similar to other macroscale firn models for stage 1 (albeit with a different dependence on the current density). The authors do point this out eventually, but I would like more discussion about this assumption. Is it still a grain boundary sliding model, and why?

- Related: The authors use RMSD to quantify the misfit between their model variants and firn-depth-density profiles – but are any of these variants significantly better than previously published firn densification models for stage 1? I think it would be a relatively simple exercise to run, e.g. the Herron and Langway (1980) or another firn model, for these sites to show that the authors' model provides an improvement.

- A pesky problem for the firn modeling community is layering in the firn. The core the authors chose to highlight shows significant layering. Like other models, their model does not capture the layering; but, I suggest that a physically based model should be better at simulating layering than the macroscale models. As the authors are claiming that their model formulation is more physically based, a bit of discussion about this model deficiency would be welcomed.

- I would like to see a more detailed uncertainty analysis as a part of the discussion (or as its own section). Not until the end of the conclusion do the authors mention that the results will be affected by biases in firn model forcing data that comes from a regional climate model; this is true, and it should come sooner. Additionally, how do other assumptions contribute to uncertainty, e.g. the assumption of surface grain size and Arrhenius activation energy? To what extent is C just accommodating other errors – e.g. the authors assume that the model calculates grain size r accurately using the Gow formula – but errors are introduced by the boundary condition, and this formula is just an empirical relationship itself.

The authors propose a linear relationship to predict C from SMB. If this relationship exists, they should include the equation for the line, and use the value of C predicted by it to run their model. How do those results compare to the outputs from the optimized model? Are they acceptable? Are they better than predicted by a previously published model?
* * *
**Specific Comments**

**1. Introduction**
Line 15: the first paragraph should be rewritten – it is filled with grammatical errors and does not contain citations. I would argue that cold content (Vandecrux et al., 2020) and ice lenses (MacFerrin et al. 2019) are just as important for governing melt water retention. I do not disagree that firn densification is important for meltwater retention, but I do not think this paragraph adequately makes a strong case that it is. The last sentence implies that current models are not adequate but does not provide evidence that they are not.

L22: I would not say that the other models neglect the overburden stress exactly – for models operating at small-ish (sub annual) timesteps, they use the mean accumulation rate over the lifetime of a firn parcel (Li and Zwally 2011, 2015), which is effectively the stress, rather than the mean site accumulation rate.

L29: Arnaud and Goujon are actually the same firn-densification model, Goujon just added heat conduction to the Arnaud densification physics. I think it is worth noting at this point that Anaud/Goujon used Alley (1987) equations for stage 1 densification because that is what your paper is about. Their equations for stage 2 densification are based on Maeno and Ebinuma.

L26 – 35: Since your paper is about modeling grain boundary sliding in stage 1, I think it would be prudent for you to focus on how the models handle densification in stage 1. That is, you talk about sintering before you talk about grain boundary sliding (paragraph starting on line 32) – this is just distracting.

L36: change to 'first applied grain boundary sliding theory …'

L36: be specific of what you mean with 'low density' – less than the critical density, I think.

L39: numerous times: cite numerous studies

L40: 'tried to' – colloquial, vague language. Hypothesized? Showed?

L48: 'in the sense of Alley' → 'Following Alley,'

**Section 3.1**

My general feeling with Section 3.1 (and much of section 3) is that it is distracting from the what the authors state is the goal of their paper (at the end of the introduction), which is to evaluate the efficacy of a grain-boundary sliding densification model. I think that these details of the numerical treatment help the paper achieve that goal. My suggestion would be to move this section to an appendix at the end of the paper. Perhaps I feel this way because I got lost in some of the details, which I attribute to several instances of imprecise language (See below comments). I am admittedly not familiar with the updated Lagrangian formulation, but I don't think that understanding the numerics of that numerical scheme are

important for understanding the results (hence my suggestion that this section be moved to an appendix). The meat of the paper begins with section 4, so reduce section 3 to only what is necessary.

Equation 1: you say you let $v\_b = v$, but when you do that, the left term implies that the time derivative of the density (i.e. the densification rate) is zero – is that what you mean? I am not sure what you are getting at with this.

L74: 'build' → built

Line 71: I am confused a bit with $v$ and $v\_b$ – can you clarify the language? You use multiple terms ('flow velocity', 'boundary velocity' and 'material velocity', and later 'grid point velocity'), but only have the two variables. Please choose one term for each variable and stick with that. It is also not clear which velocity is which – e.g. with 'boundary velocity', are you talking about the boundary of one grid point/control volume, or the boundary of the entire model domain (i.e. downward advection due to underlying ice dynamics?).

Equation 2: please use a different variable for stress – you already use $t$ for time (eq. 1). This gets especially confusing because Alley (1987), equation 5, matches your equation 5; but your $t$ has a different meaning than the $t$ in the Alley paper. I don't think it is sufficient to differentiate between $t$ and $t\_zz$. Alley used $P$ for pressure; Morris and Wingham (2014) and Arthern et al. (2010) used $\sigma$.

Equation 4: With this formulation, it seems that in the accumulation zone the model domain will continue to thicken indefinitely?

**Section 3.2**
- write Equation 5 as a part of a sentence (as you have done with other equations)

- Stress term in Eq. 5 – Alley (1987) makes a steady-accumulation assumption. In this paper it appears that you do not – see comment regarding this at Section 4 comments.

- Line 107: 'seeming arbitrary at first' is a subjective comment (is it more arbitrary than the 5/3?); remove it.

- Do not use 'term' and 'factor' interchangeably. (e.g. you use 'term' to describe $8D_{BD}\Omega/k_b Th^2$ but this is a factor, not a term.)

-Line 117: 'kind of fade out behavior' – please use specific language and avoid colloquialisms – 'fade out' is used several more times, but I think you can be more precise, e.g. how does it fade out? e.g. "the influence of grain boundary sliding on the strain rate decreases asymptotically as the density approached the critical density".

L117: 'inverse relative density of rho_c = 550 kg m$^{-3}$' – that value is the actual density, not a relative density. I think you mean something else here; please clarify.

-Line 120: previously you said $T$ 'resembles the temperature' (what does that mean?), and here you explicitly say that $T$ is the temperature.

-Line 120: you previously said $r$ is grain radius, and here you redefine it slightly differently ('grain size' is a broader term than 'grain radius'). Which is it? Please be more specific of what you mean by grain

radius, because the grains are not perfect spheres and different people mean different things when they talk about grain radius.

**3.4**
L128: Again, I think you have just not explained your method clearly – what is a 'test volume'? This is the only time you use that phrase. Upon first glance, this sentence says that the densification rate is zero; the next sentence clarifies this a bit – but not until the third sentence do I figure out that your control volumes are changing. The upshot is that I don't doubt that your numerical scheme is correct; I think it is not explained clearly.

**3.5**
Line 142: This is an incomplete sentence. Discretized by what? Or maybe it is just written oddly?

Line 145: please cite page numbers with textbooks.

Equation 11: Is your equation for conductivity your formulation? I do not see it in Paterson 1994; please either cite source material or state that you came up with this parametrization (and how).

**3.6**
- Please provide what you use for a surface boundary condition for the grain size.

- "This is suitable as the grain radius is used in the constitutive equation by Alley (1987)" – please justify this assumption. Also, second use of word 'is' appears to be a typo.

**Section 4.**
Alley (1987) did not include heat diffusion (he assumed isothermal conditions), and he assumed steady accumulation. How does your inclusion of it affect your results vis-à-vis those of Alley? Please provide justification that it is appropriate to "test the concept of the material model developed by Alley.

Eq. 14 and 15: You are adding an Arrhenius term, which was not included in Equation 5 (but is of course included in many other firn models. Given that your paper is about optimizing the grain boundary sliding approach, can you justify this addition, or add text describing why you felt it was appropriate?

L172: How does lumping these parameters together affect the results? I would expect that they will vary by site/climate, and so what does it mean for your grain boundary sliding model that you remove the specific parameters that describe grain geometry?

L187: what does 'its' refer to? i.e. to test what's influence?

L191: It is not clear to me why a possible dependency (of the strain rate?) will reflect in the optimal values of Cv3 and Cv4? There is still a factor of 1/T in the equations.

L193: → 'The aim of the optimization…'

L195: Where is the Wilhelms core taken from? Please mark in on the map (figure 3) in a different color and provide coordinates. I suggest coming up with an easier-to-read name (i.e. state the full name once here when introducing and citing it, and give it an easy-to-recognize abbreviation to use for the remainder of the paper). I suggest this because I get concerned when I am reading if there are multiple cores with similar names that I need to be differentiating between.

L200: this implies that Lundin and Verjans did not use objective measures for their studies, which is not the case. Verjans did in fact use RMSE. I will also suggest that RMS deviation is not wholly objective in this scenario (which is not to say it is not useful) – imagine that you had a scenario where you were able to model the density perfectly at all depths except a single depth where it is off by 20 kg m$^{-3}$. If this misfit point is near the surface where the density is low, it is a much higher percent misfit than at greater depths where the density is greater.

L209: It occurs to me at this point that you have not described how you are modeling densification in stage 2 – ostensibly your model goes beyond stage 1 densification because this is important for getting the temperature profile/heat transfer correct. With the Arnaud/Goujon/Breant models, there is a factor introduced to allow the densification rates to match at the critical density – are you doing something similar? I do not think you need to provide a lot of detail but you should add a sentence describing what you do for density greater than 550.

L214: Please be more specific about what tests you did.

L215: A general note: Thank you for consistently including the units with your variables and equations.

L223: It would be helpful to explicitly say that there are 21 densities you tested, e.g. 'we tested 21 different values of the surface density…"

**Section 5**
L232: Be specific of how many sources, or leave that out. Probably you could just remove your short description of what SUMup is (the fact that is is available online and has a lot of measurements is not germane for the point of your paper)

Figure 2: Since you include the SMB and temperature in a plot, it might be useful to also include the mean of those as predicted by RACMO.

L241: Positive SMB and melt are not exclusive - quite a few sites experience significant melt in Greenland yet still have positive mass balance. Quite a few of your sites, as far as I can tell on your map, are in wet firn locations. How did you handle these sites with your simulation? Your model handles densification due to compaction but not densification due to meltwater percolation and refreezing. How does this assumption affect your results? Do you see different optimal values of C for sites where melt > 0? Or higher optimal surface densities?

L255: Except it appears that the cold, low-accumulation center of the East Antarctic ice sheet is not represented, and this is one place of high interest for firn studies because of its relevance to ice core studies. (The Arnaud model specifically used Vostok as a test site).

Section 5: I suggest moving section 5 to before section 4. At this point my feeling as a reader is that I have made it to the 12$^{th}$ page and am finally getting to the part of the paper where you tell me more specifics of the science you did (this is also related to my suggestion that you move a majority of section 3 to an appendix.)

Section 5.2: I suggest rewriting the first paragraph of this section for clarity.

L259 – 268: I am confused here. You said earlier that your model time resolution is 48 steps per year (roughly weekly), but you say here that you are using annual data from RACMO. Why do you not use the daily RACMO data downsampled to weekly resolution? Are you using the same value for the temperature

and accumulation for each of the time steps during a given year? On Line 260 you say you are interpolating, but what are you interpolating (spatially? temporally?), and how?

L269: In the cases where you have cores that have measurements from the near surface, why don't you just use the measured surface density from the cores? Are there cases where the surface density from the optimization scheme is significantly different than the observations from SUMup? It seems possible that your scheme could allow a surface density that is different than the observation in order to get a better overall RMSD fit. Do you use the same density at all time steps for a given site/simulation?

L270: What is the basis of your choice of 0.5mm for initial grain size? Is it a realistic assumption to assume it is the same everywhere? (I would think not – southern Greenland is quite different than e.g. the South Pole). How sensitive is your model to this parameter choice?

L274: But aren't the surface densities in good agreement because the range you chose for them is based on what the literature says in the first place? This seems circular.

L277: I think there is literature that you could find to cite regarding Antarctic vs. Greenland surface densities. Or you could query SUMup and find all measurements shallower than 50cm or 1m for Greenland and Antarctica.

L279 – 280: This seems out of place – should it be in the temperature section in section 3? Also you need to specify what your model domain depth is since you are using a Neumann condition.

**Section 6:**
L290: 'even better agreement' – this implies that that core has good agreement in the first place – but does it? You do not provide context for whether ~23 kg m$^{-3}$ is actually a good agreement. I suggest that RMSD is a good metric to intercompare the models' performance – v2 is doing better in general than v3. But it does get murkier when actually deciding if this is a "good" fit to the data – so this would be better if you could normalize the RMSD somehow, and provide a quantification of what you consider to be a good fit (any why).

Figure 5: please label which variant is which in a more obvious manner (the v1 is small and hard to spot).

L291: Isn't this more a function of the SMB (from RACMO) being accurate than it is the strain rate being correct? If you were to optimize your model to fit the depth-age profile best (rather than depth-density), would the same optimal parameters be the same?

L296: This is confusing. Alley (1987) as far as I can tell does not include an Arrhenius term. (Although he did derive an activation energy for grain boundary sliding). This makes me wonder: the activation energy you use is notably smaller than recent studies have implied it should be (Arthern et al. 2010; Morris and Wingham 2014) – how does your choice of activation energy affect your results?

Note on figures in general (especially Fig. 6): the colors you have chosen can be challenging to see (especially the yellow) when text is written in those colors.

L305/Figure 7: To me these data do not appear to be linear, and the lines do not appear to be good fits (especially at low and high temperatures). A Pearson correlation coefficient is based on the assumption that the data are linear, which I am not convinced they are, so I think it is deceiving to include that metric. To me the most obvious conclusion is that there is not a clear linear relationship between any of the c parameters and temperature. If this is not what I should take away from figure 7, you need to be more thorough convincing me.

Figure 7: what does a negative value of c indicate? (it is hard to tell if there are any; but figure 7's y axis extends to -2, and it looks like there could be? Make the axis the same as figure 8.)

310: Again, 'even higher' is implying that you had a good fit between temperature and c, but the a high Pearson is meaningless because the data in figure 7 are not linear.

L312-315/Figure 8: These do appear to be more linear than the C vs temperature plot, but I think it would be appropriate to do a statistical test to actually show that they data are linear (e.g. a statistical model with higher order terms that have coefficients not significantly different than zero, or some other test). I am concerned with the values at the high accumulation sites for variants 1 and 2 – there are clusters that are half of what would be predicted by your linear model. And, what is going around 0.4 m w.eq./a? (And 0.6?) These vertical series of dots would indicate to me that there is no correlation between SMB and C – i.e. you can have a number of sites with SMB = 0.4 and C can vary by a factor of 10.

- Your discussion should include discussion of why the optimal C values vary by site and how that influences our understanding of grain boundary sliding.

**Section 7:**
L323: Again, I suggest that getting depth-age correct is more dependent on getting the accumulation rate correct – and, since the accumulation rate is often determined by counting annual layers, the science can become a bit circular.

L326: I agree entirely with what you say in this paragraph – your model formulation is not too different than the other models that have been published in recent years, but up until this point you have been claiming that you are optimizing a grain boundary sliding model (including in the title). I think it would be appropriate for you to rework the text a little bit to be more forthcoming with this, rather than waiting until the very end to point it out.

L338: Morris (2018) did in fact create a 'transition model' to move from stage 1 to stage 2.

L345: I am confused here – are you suggesting that because you do not have a separate Arrhenius factor in equations 16 and 17 there is implicitly an Arrhenius factor wrapped into $C_{v3}$ and $C_{v4}$? If this is the case, say so clearly in Section 4. It seems that you have a baseline assumption in your study that firn strain has an Arrhenius temperature dependence. But, if Variants 3 and 4 give good answers, couldn't this just as easily indicate that firn does not have an Arrhenius dependence? Part of my point here is that you have coefficients, $C_{v3}$ and $C_{v4}$, that have all sorts of 'physics' wrapped into them, and I don't think you can just decide what those physics are.

L349: The first two sentences of this paragraph need to be rewritten to improve clarity. After multiple readings it is still not clear to me what you are trying to convey.

L349, 352: 'resulting factors' – do you mean the 4 values of C for your variants? C and the surface density? Please use more specific language, e.g. 'indicate a clear dependency of $C_{v3}$ and $C_{v4}$ on the mean surface mass balance'.

L350: This is not obvious to me – are you saying that the C factors are functions of temperature, and since temperature and SMB are correlated, then C must also be a function of SMB?

L353: What do you mean by "Dependency of the mismatch"?

L354: You can at least do a scale estimate of the effect that horizontal stresses would contribute – see Horlings et al., 2020. Your paper is about stage 1 densification/grain boundary sliding – on the timescales of firn densification on ice sheets, is horizontal stress going to make a significant difference? I am also skeptical that an 3D model incorporating horizontal stresses would significantly reduce your misfit – at least in the example core you highlighted, the layering of the density profile likely contributes to a significant portion of the misfit.

**Section 8:**

L367: not represented well in the model – are you effectively saying here that the stress should have an exponent other than 1?
L371: I agree with this paragraph; I think it should be moved into the discussion section along with a more complete uncertainty analysis.

**References:**
Arthern, Robert J., et al. "In situ measurements of Antarctic snow compaction compared with predictions of models." *Journal of Geophysical Research: Earth Surface* 115.F3 (2010).

Herron, Michael M., and Chester C. Langway. "Firn densification: an empirical model." *Journal of Glaciology* 25.93 (1980): 373-385.

Horlings, Annika N., et al. "Effect of horizontal divergence on estimates of firn-air content." *Journal of Glaciology* 67.262 (2021): 287-296.

Li, Jun, and H. Jay Zwally. "Modeling of firn compaction for estimating ice-sheet mass change from observed ice-sheet elevation change." *Annals of Glaciology* 52.59 (2011): 1-7.

Li, Jun, and H. Jay Zwally. "Response times of ice-sheet surface heights to changes in the rate of Antarctic firn compaction caused by accumulation and temperature variations." *Journal of Glaciology* 61.230 (2015): 1037-1047.

MacFerrin, Michael, et al. "Rapid expansion of Greenland's low-permeability ice slabs." *Nature* 573.7774 (2019): 403-407.

Morris, Elizabeth. "Modeling dry-snow densification without abrupt transition." *Geosciences* 8.12 (2018): 464.

Morris, E. M., and D. J. Wingham. "Densification of polar snow: Measurements, modeling, and implications for altimetry." *Journal of Geophysical Research: Earth Surface* 119.2 (2014): 349-365.

Vandecrux, Baptiste, et al. "Firn cold content evolution at nine sites on the Greenland ice sheet between 1998 and 2017." *Journal of Glaciology* 66.258 (2020): 591-602.

---

## Author Comment (AC1)

**Response to Anonymous Referee #1, 29 Mar 2021**

Dear colleague,

We thank you for the careful reading of the manuscript and the constructive remarks. We have discussed all points raised by you and accordingly changed the manuscript. In the following there is a point-by-point response and discussion of the issues addressed in the review.

There is one thing we want to mention. It seems to us that we mislead the reviewers in that respect that we give the impression that we are up to determining one perfect coefficient $C$, which can then be used for all future simulations of firn densification. This is not the case. We are using the optimisation approach to test if the constitutive relation by Alley (1987) in its particular form can be used to simulate firn density profiles fitting observed ones. We then analyse if the magnitude of factors $C$ leading to the best optimisation results exhibits a dependence on another system variable (e.g. surface mass balance), to see if in future, one could extend or improve the constitutive relation. Alternatively a better representation of the stress could lead to better results in the future. The study is not about finding the best factor $C$, but that a minimum in the optimisation process exists and how much off density profiles simulated using the constitutive relation by Alley (1987) are compared to measured firn density profiles.

Sincerely,

Timm Schultz on behalf of the co-authors

**1 General Comments**

(1) The SUMup data base is an on-going community effort which sets a very clear condition for use of the data. The guidelines state "When using this data set please cite both the individual researchers who provided the data as listed in the Citation column as well as the SUMup dataset". There is a citation for one profile in this paper (ngt03C93.2, shown in a figure) but there are no citations for the 158 other profiles used. This may seem like a tedious task, but it is essential that researchers are properly acknowledged. It may be that the editor will accept a table of profiles with citations as Supplementary Material, but whether this is adequate needs to be confirmed with the SUMup team.

> We agree with the referee that proper acknowledgement of the data sources is important. For the revised version of the manuscript we will provide a document in the supplementary material that lists not only all factors leading to the best simulation result together with the corresponding RMSD for all profiles and tested variants, but also the reference to the original data for each firn core.

(2) The authors have constructed a numerical model for the evolution of temperature, density and grain radius in firn using the Alley densification equation and forced by annual values of surface temperature and accumulation rate from the RACMO2.3 meteorological model. The advantages of using the RACMO data are (i) that profiles for which in situ climate data are not available can be simulated and (ii) that inter-annual climate variations can be taken into account. The disadvantages are that (i) local data have to been found by interpolation and (ii) systematic errors may be introduced. It would be helpful if the authors could provide an assessment of how accurate the forcing data are likely to be.

> The authors have no background in regional climate modelling. We use RACMO as a data product, which is well established in the community. Therefore we can not assess the accuracy of data provided by RACMO. Further information on RACMO can be found in corresponding publications, cited within the manuscript, or on the project's website hosted at Utrecht University:
>
> https://www.projects.science.uu.nl/iceclimate/models/index.php, 25.06.2021

(3) Since the input temperature is only available at yearly intervals, the annual variation of temperature in the top layer of the firn cannot be simulated. Alley dealt with this problem by excluding the top 2 m of the firn, which experiences the greatest temperature fluctuations. In l.203 the authors state that they optimise in a domain "bounded by the surface and the oldest horizon within the profile affected by the forcing". In fact some records, including ngt03C93.2, shown as an example in the paper, do not extend to the surface. I assume the authors actually use an upper bound as near to the surface as possible. In any case, it is quite important to discuss why they choose not to follow Alley's example. The reader will need some convincing that inclusion of data from a region with strongly-varying temperature does not affect the optimised parameter values.

The simulation of the very first, near surface layer of firn is a general problem. However, tests have shown that a seasonal temperature signal has very limited influence on the resulting firn density profile compared to the same model setup using the mean annual temperature. Variations of the density, especially on a small scale, are also driven by wind and changing impurity contents (e.g. Freitag et al., 2013). We will include such considerations in the revised version of the manuscript.

Furthermore we found that the constitutive relation for grain boundary sliding performs quite well in the upper two meters of the firn column. For example in the study by Arthern and Wingham (1998) a constant value for the strain rate was used in the upper two meters. We don't see any advantage in this. Most likely the optimisation for a site specific surface density also compensates for an insufficient model approach. We will also include these points in the revised version manuscript. However, are convinced that the general idea of the optimisation is not significantly affected by this.

The domain for the computation of the root mean square deviation between simulation results and measured firn density profiles is indeed bound by the upper most point of the measured profile. We will clarify this in the revised version of the manuscript.

(4) One of the factors in the Alley equation ($\alpha'$) describes the effect of spherical averaging of the sliding velocity along variously-oriented grain boundaries to obtain the mean vertical velocity. Based on geometric arguments Alley suggests $\alpha' = (1 - N/6)$ where $N$ is the coordination number and notes that, by observation, $N \approx 10\,\rho/\rho_i$. The authors make the interesting comment that $\alpha' \to 0$ for $\rho = 550\,\mathrm{kg\,m^{-3}}$; in other words the Alley equation implies that grain boundary sliding must stop at this density. An alternative expression for $\alpha'$ from Bréant implies that grain boundary must stop at a higher density of $596\,\mathrm{kg\,m^{-3}}$. The authors choose to restrict their optimisation domain to densities below $540\,\mathrm{kg\,m^{-3}}$ "due to the asymptotic characteristic of the resulting density profiles using the... constitutive equation" (l.209). But the reader might wonder why the cut-off should not come when densification by grain-boundary sliding is no longer much greater than densification by other processes (e.g. dislocation creep). There is a brief mention of the possibility of competing processes in the discussion section of this paper (l. 328) but this comes only as a suggestion for future consideration. I think the reader would appreciate a much deeper discussion of the implications of this choice of cut-off density, earlier in the paper.

> Our study focuses on grain boundary sliding, its description by Alley (1987), and the assumption that it is the leading process in the first stage of firn densification. We do not incorporate other processes at this point. The restriction of the optimisation domain to densities below $540\,\mathrm{kg\,m^{-3}}$, in certain cases, is a pragmatic choice. A criterion as suggested by the referee, considering the contribution of different processes, would be the better one. However, it would require the knowledge how these different processes actually act in the upper firn. This is not given at this point. We will address these points briefly in the revised version of the manuscript.

(5) Another problem arises because the authors have not chosen to discuss the question of layering in firn density profiles, even though their example profile clearly shows large fluctuations in density superimposed on the general increase in density with depth. They use a microscale, physics-based model, which applies to a small element of snow, and a grid spacing of 48 points per annual layer, so it would be possible to investigate the effect of annual layering. The authors may choose not to do this, but it is important then to bring out their underlying assumption, which is that the Alley equation can be applied on the macroscale by simple substitution of the macroscale mean density. Given the work that has gone into writing the model and selecting profiles to simulate, it would a pity not to take the opportunity to include some discussion about the problem of up-scaling.

Layering in firn is a topic highlighted by the growing amount of high resolution firn density data. However, it is not frequently addressed in firn density modelling. We follow the assumption of for example Freitag et al. (2013) that small scale layering in firn is driven by a changing concentration of impurities, leading to softening. This can be expressed in a dependency of the activation energy, used for the description of deformation mechanisms, on the impurity concentration. In the presented model this would be possible via the Arrhenius equation describing the description of boundary diffusion.

However, a suitable relation would be needed as well as data describing the impurity concentration (probably by a proxy). These data are not available on a global scale, so we (and the entire community) are lacking forcing data and can consequently not represent the short scale layering, but the average density over a larger spatial scale. By the use of the root mean square deviation as cost function, the optimisation result is not affected when using layered firn profiles as a basis.

We will address these points in the revised version of the manuscript and will add a running mean density profile for the example core ngt03C93.2 in Figure 2 a) to illustrate how the optimisation results relate to the mean density.

6) In the paper 4 versions of the Alley model are tested by optimising one parameter $C_v$ per version. The reader will want to know how the values of $v$ implied by values of $C_{v1}$ compare with the Alley values and also what the individual values of the cost function RSMD are. These could be included in a table of sites, input data and results in the Supplementary Material.

> In the revised version of the manuscript we will provide a list containing the best parameters resulting from the optimisation approach and the corresponding values of the root mean square deviation in the supplementary material. However, it is difficult to compare the resulting factors to the boundary viscosity used by Alley (1987). The boundary viscosity depends on the temperature which is neither constant with respect to time nor along the firn profiles. One would have to assume appropriate effective parameters. Furthermore the resulting factors $C$ may depend on other effects like the improper description of the stress, as pointed out in the conclusion section.

(7) The authors conclude that their results show that "the description of grain boundary sliding introduced by Alley (1987) is suitable for the simulation of firn densification at low density" (l. 362). In fact this would only be true if the value of parameter $C_v$ could be specified a priori. The authors do not say that the linear relations with mean annual temperature and accumulation shown in Figures 7 and 8 could be used to determine global values of $C_v$ but, if this is what they mean, then we need to know what the cost functions are when profiles are simulated using these global values.

> In the opinion of the authors, the determination of global values for the factors $C$ is not useful at this point. Such assumption of global values would not lead to better simulation results compared to other published firn densification models. However, we think the constitutive relation for grain boundary sliding by Alley (1987) is a good basis for the description firn densification at densities below the critical density. A determination of a global factor, or rather of better parameters incorporated in the factor for this study, may be possible if an improvement of the constitutive relation can be achieved. We see a possible improvement in a better description of the stress regime, also mentioned in the manuscript.
>
> We use the site specific values of $C$, found using the presented optimisation scheme, to identify differences in the variants of constitutive relation. Due to the fact that the general structure of the constitutive relation is maintained and the simulated density profiles are similar, because of the optimisation, we analyse the differences in the resulting factors, representing site specific ideal parameters, to identify possible additional dependencies and improvements.
>
> This was not formulated clearly in the first version of the manuscript. We will improve it to point out the idea of the method.

Comment (8) On the whole the reader is left with the impression that the Alley model is inadequate, because the supposedly-fixed parameter $C_v$ varies with the input conditions. This may be somewhat unfair. The variation may arise because the model has been applied too near the surface, layering has been ignored or indeed, as the authors remark, the input meterological data are inaccurate. But if there is a real dependence on accumulation rate this can be traced back to the calculation of stress $t_{zz}$ (Equation (6)). Do the authors think that their results show that the vertical grain sliding velocity is not linearly related to the vertical force on the grain? Could this be because other non-linear processes are acting as well as grain-boundary sliding? Or is the Alley (1987) analysis too simple?

> The study aims to determine a site specific ideal factor $C$ for every analysed firn profile. Even if for example layering is neglected, using the optimisation scheme, we are able to produce simulated firn density profiles matching measured ones well. As the amount of analysed firn profiles is large, random error should play a minor role when comparing the resulting values of the site specific factor $C$ to other environmental properties.
>
> Other processes than grain boundary sliding could act in the first stage of firn as well. We do not necessarily think the analysis by Alley (1987) is too simple, but the general assumption of a one dimensional model approach neglecting horizontal boundary conditions and lateral components of the stress tensor. We hereby do not mean so called "horizontal divergence" (e.g. Horlings et al., 2020), but stresses occurring due to lateral confinement. As these points were not explained clearly enough in the manuscript, we will revise it to highlight the aim of the study and its results.

**2 Other Comments**

l.2. "first introduced by Alley (1987)" Perhaps better to say that the theory of grain-boundary sliding in snow was developed by Alley in this paper? The concept that this process might occur was already in the literature.

> In the revised version of the manuscript the formulation will be altered as proposed by the referee.

l.15. "a couple of research fields" implies two fields, whereas the authors list three

> To avoid confusion, the wording will be changed in the updated version of the manuscript.

l.20 "The first and greater (category)". This sounds like a value judgement; better to say that the majority of models fall in the first category

> We will change the sentence following the suggestion of the referee.

l.22 "this approach neglects". Do these models really neglect overburden stress or is it implicit
rather than explicit in the model equations?

> The addressed models are based on the Robin hypothesis (Robin, 1958) which states that
> the density change is proportional to the change in stress resulting from accumulation.
> This hypothesis is basically an interpretation of the linear momentum balance in its quasi-
> static form. Hence the stress is implicitly integrated in the descriptions. To clarify this and
> avoid misunderstandings the sentence in question and the following one will be removed
> from the manuscript.

l.32 onwards "which applies for firn". The paragraph goes on to say the theory does not apply
to firn below the critical density. This is confusing. Does the theory not apply because the snow
is not compacted enough? Or because the grain geometry is not regular?

> The theory of hot isostatic pressing as described for example by Kang (2005) was developed
> based on very different conditions than those found in firn below the density of $550\,\mathrm{kg\,m^{-3}}$.
> Both the compaction as well as the grain geometry contribute to these differences. As
> pointed out, the study focuses on grain boundary sliding, therefore the theory of sintering
> and especially hot isostatic pressing is not relevant at this point. Therefore the addressed
> paragraph will be removed from the manuscript.

l.40. Theile et al (2011) "tried to point out" or "did point out"?

> We will formulate this sentence in a more appropriate way.

l.43 Are these 4 profiles also simulated in this paper? Which are they?

The four cores Alley used in 1987 are not part of the SUMup dataset. So they are not included in this study, but it is interesting to re-evaluate these data. The problem of many of the "old" firn cores is that the original data are not published. To our knowledge the 1987 paper by Alley is in fact the original reference for two of the cores.

The study by Spencer et al. (2001) features the four cores. There actually exists a dataset on the internet based on the 38 firn cores used in this study and some more cores. The authors are in fact in possession of this firn core data. However the origin of these data is not really traceable nor citable. Spencer et al. (2001) received many cores due to personal communication.

The authors strongly encourage everyone to contribute to the SUMup dataset to build a database as complete as possible. We will address the fact that the four profiles used by Alley (1987) are not featured in this study in Section 5.1 of the revised version of the manuscript.

l.101 $m\,w.e.\,a^{-1}$ is used elsewhere; why does $m\,w.e.\,s^{-1}$ have to be used here?

The unit of $\mathrm{m\,w.e.\,a^{-1}}$ is usually better comprehensible for readers than $\mathrm{m\,w.e.\,s^{-1}}$ and is therefore used throughout the manuscript, for example in Figure 2. It is absolutely possible to use the unit of $\mathrm{m\,w.e.\,a^{-1}}$ in Equation (4) and the following paragraph as well. Nevertheless it would require an additional conversation of units. The authors prefer to use SI-units in all equations and model implementations and would therefore hold on to the existing formulation.

l.107. Why does 2/15 seem an arbitrary number? To say it results from the geometric deviation does not really explain to the reader where it comes from. Since the paper is based on the Alley model, expanding the derivation would be worthwhile. Then the reader would understand the physics behind the model without having to go back to the original paper

The word "arbitrary" will be removed from the revised version of the manuscript. Although the derivation of the factor 2/15 is not difficult per se, it results from the combination of different surfaces describing intersections of grains, the coordination number of those grains and the correlation of the coordination number with the density derived by Alley (1987). The authors see no benefit to reproduce the geometrical considerations of Alley (1987). The interested reader can easily follow the derivation of the factor 2/15 by taking a look at Equations (1) to (4) in the original paper.

l.109. Is the word "resembles" right here? Maybe "represents"?

> The mistake will be corrected as suggested by the reviewer.

l.128. Is "test volume" right? Maybe "volume element"?

> In the revised version of the manuscript the term "control volume" is used consistently.

l.150 Would it be worth pointing out that Alley used observed values of $r$?

> Although Alley (1987) Alley used observed values for the grain radius, these data are not published nor are ranges given. The authors address this issue briefly in the revised version of the manuscript.

l.200 This sounds as if the authors do not think the Lundin and Verjans papers use an objective cost function. The sentence could be altered so it is clear that the authors are explaining that they have used a different objective cost function.

> In fact the authors see difficulties in the use of the depth integrated porosity as cost function. Nevertheless the revised version of the manuscript will no longer feature the studies by Lundin et al. (2017) and Verjans et al. (2020) at this point.

Figure 2. Is 2(c) a true representation of the input to the model? Or is the accumulation constant for a given year?

> Panel (c) of Figure 2 shows the true representation of the model input. In the revised version there is a detailed discussion of the model input and forcing terms.

l.278 The difference in surface density between Greenland and Antarctic profiles merits further discussion. To say merely it is "plausible" misses an opportunity to comment on different types of snow (see Salamatin for example) and whether the densification rates might be different for Greenland and Antarctica.

> We will neglect the discussion of different surface densities at this point, but not because it is not important, only because we do not have the means to contribute something on the surface density from our work. From the mechanical perspective it is an input parameter that is unfortunately poorly constrained and we would be delighted to see more studies upcoming on the issue of surface density. A database similar to SUMup would be of great value.

l.279. It is not clear here whether the lower boundary for the simulation is the same as the lower boundary for the domain used to calculate the deviation between observed and modelled densities. The lower boundary for the simulation needs to be deep enough for the temperature to be constant, whatever the temperature variation at the surface.

The lower boundary for the simulation does not match the domain used for the computation of the deviation. The lower boundary of the comparison domain is either limited by the length of the measured density profile, the oldest horizon influenced by the forcing or the density of $540 \, \mathrm{kg \, m^{-3}}$ (see also our response to your comment (4) regarding the optimisation domain).

However, we do not develop the density beyond the first stage and the critical density. Regarding the computation of the temperature a spin-up and consideration of the following stages of densification would be the best choice. We have not done that in this study, because not many of the analysed firn profiles extend below the depth of $10 \, \mathrm{m}$ to $15 \, \mathrm{m}$, which is the depth at which the mean annual surface temperature is established (e.g. Cuffey and Paterson, 2010). We are not investigating seasonality at this point.

In the revised version of the manuscript we cite the article by Vandecrux et al. (2021) which has been published after our submission. The study presents a firn temperature profile at "Camp Century", which shows that a Neumann boundary condition at the depth we are applying it is not an issue, as the temperature change below the depth of 5 m is insignificant for our approach. The study by Orsi et al. (2017) shows that the temperature change in the upper $150 \, \mathrm{m}$ of firn is only about $1 \, \mathrm{K}$, which is also small in context of our work.

In order to assess what influence this rather small temperature change has on the outcome of our study, the revised version of the manuscript contains a simulation in which we have reduced and raised the temperature at one site. We check whether the ability to find an optimal value throughout the optimisation process is reduced. As expected, we find that it has little influence on the assessment of the constitutive relation.

l.320 This would be a good point to tell the reader whether using the Alley model, with one parameter optimised, gives a better fit than using a semi-empirical model (e.g. Herron and Langway) with the densification rate optimised. And "a reasonable good match" needs to be defined in terms of the RSMD

Both referees gave us the feedback that an optimisation of the constitutive law by Alley (1987) using a single factor, should lead to the determination of a global factor $C$, which then could be applied to all analysed and probably more firn profiles. Such an approach would naturally lead to an evaluation of this factor $C$ by comparing the different model approaches with the new optimised one.

This is not what we intended to do. However, we admit that the reader could get this impression, simply because we failed in making it clear enough. With our approach we wanted to evaluate the ability of the constitutive relation by Alley (1987) to reproduce measured firn density profiles and to identify possible hidden dependencies and improvements. At this point we see no possibility to determine a global factor $C$, which would result in a firn densification model performing better than other existing ones. We propose that, although the constitutive relation for grain boundary sliding is a good basis, an improvement, especially by a proper incorporation of the stress state, is needed.

The revised version of the manuscript features this more clearly. Additionally we point out that the formulation of a single value for the factor $C$ makes no sense at this point in the discussion section.

l.330 This paragraph should come much earlier in the paper because it is key to understanding the physics of the densification problem. The ideas expressed are well-established in the literature, so should be reviewed before the authors proceed to describe their work. It would then be easier to discuss the choice of lower boundary condition, the role of layering and other factors in the light of current understanding.

In the revised version of the manuscript we establish the points addressed in this paragraph at an earlier point, when explaining the structure of the constitutive equation in Section 3.2 and the optimisation scheme in Section 4.

l.353 The question of whether horizontal divergence affects the creep densification rate was raised by Alley in the 1980s. It is not clear here whether the authors are suggesting that basal-sliding is also affected

> The point the authors intended to make here is not formulated clearly enough. Horizontal divergence due to movement of the underlying ice is not the process in question. We would rather suggest that when we consider a firn column, this firn column is confined in horizontal directions. Therefore a deformation in these directions is not possible. While the firn compresses in vertical direction, it can not elongate in horizontal directions. This means that the horizontal components of the stress tensor, even in a one dimensional approach, are not zero.
>
> Because firn is a compressible material, it is not trivial to compute these stresses. Simulation approaches considering the horizontal components of the stress tensor can be found for example in Zwinger et al. (2007), (Greve and Blatter, 2009, pp. 224–230), Salamantin et al. (2009) and Meyer and Hewitt (2017). The revised version of the manuscript will address these points in more detail.

l.371. This paragraph should come earlier, when the input data are introduced, so the reader knows their limitations from the start

> In the revised version of the manuscript we address the problem of uncertainties within the forcing data, as suggested by the reviewer, in Section 5.2, when introducing the boundary conditions and forcing data.

**References**

Alley, R. B.: Firn Densification by Grain-Boundary-Sliding: A First Model, Journal de Physique, 48, C1–249 – C1–256, https://doi.org/https://doi.org/10.1051/jphyscol:1987135, 1987.

Arthern, R. J. and Wingham, D. J.: The Natural Fluctuations of Firn Densification and their Effect on the Geodetic Determination of Ice Sheet Mass Balance, Climatic Change, 40, 605–624, https://doi.org/https://doi.org/10.1023/A:1005320713306, 1998.

Cuffey, K. M. and Paterson, W. S. B.: The Physics of Glaciers, Buttherworth-Heinemann, fourth edition edn., 2010.

Freitag, J., Kipfstuhl, S., Laepple, T., and Wilhelms, F.: Impurity-controlled densification: a new model for stratified polar firn, Journal of Glaciology, 59, 1163–1169, https://doi.org/https://doi.org/10.3189/2013JoG13J042, 2013.

Greve, R. and Blatter, H.: Dynamics of Ice Sheets and Glaciers, Advances in Geophyiscal and Environmental Mechanics and Mathematics, Springer-Verlag, Berlin Heidelberg, https://doi.org/10.1007/978-3-642-03415-2, 2009.

Horlings, A. N., Christianson, K., Holschuh, N., Stevens, C. M., and D., W. E.: Effect of horizontal divergence on estimates of firn-air content, Journal of Glaciology, 67, 287–296, https://doi.org/https://doi.org/10.1017/jog.2020.105, 2020.

Kang, S.-J. L.: Sintering, Densification, Grain Growth, and Microstructure, Butterworth-Heinemann, 2005.

Lundin, J. M. D., Stevens, C. M., Arthern, R., Buizert, C., Orsi, A., Ligtenberg, S. R. M., Simonsen, S. B., Cummings, E., Essery, R., Leahy, W., Harris, P., Helsen, M. M., and Waddington, E. D.: Firn Model Intercomparison Experiment (FirnMICE), Journal of Glaciology, 63, 401–422, https://doi.org/https://doi.org/10.1017/jog.2016.114, 2017.

Meyer, C. R. and Hewitt, I. J.: A continuum model for meltwater flow through compacting snow, The Cryosphere, 11, 2799–2813, https://doi.org/https://doi.org/10.5194/tc-11-2799-2017, 2017.

Orsi, A. J., Kawamura, K., Masson-Delmotte, V., Fettwis, X., Box, J. E. Dahl-Jensen, D., Clow, G. D., Landais, A., and Severinghaus, J. P.: The recent warming trend in North Greenland, Geophysical Research Letters, 44, 6235–6243, https://doi.org/https://doi.org/10.1002/2016GL072212, 2017.

Robin, G. d. Q.: Glaciology III: Seismic Shooting and Related Investigations, Norwegian-British-Swedish Antarctic Expedition, 1949-52, Scientific Results Vol. 5, Norsk Polarinstitutt, Oslo, 1958.

Salamantin, A. N., Lipenkov, V. Y., Barnola, J. M., Hori, A., Duval, P., and Hondoh, T.: Snow/Firn Densification in Polar Ice Sheets, in: Physics of Ice Core Records II: Papers collected after the 2nd International Workshop on Physics of Ice Core Records, held in Sapporo, Japan, 2-6 February 2007, edited by Hondoh, T., pp. 195–222, 2009.

Spencer, M. K., Alley, R. B., and Creyts, T. T.: Preliminary Firn-Densification Model with 38-Site Dataset, Journal of Glaciology, 47, 671–676, https://doi.org/https://doi.org/10.3189/172756501781831765, 2001.

Vandecrux, B., Colgan, W., Solgaard, A. M., Seffensen, J. P., and Karlsson, N. B.: Firn Evolution at Camp Century, Greenland: 1966-2100, Frontiers in Earth Science, 9, 1–16, https://doi.org/https://doi.org/10.3389/feart.2021.578978, 2021.

Verjans, V., Leeson, A. A., Nemeth, C., Stevens, C. M., Kuipers Munneke, P., Noël, B., and van Wessen, J. M.: Bayesian calibration of firn densification models, The Cryosphere, 14, 3017–3032, https://doi.org/https://doi.org/10.5194/tc-14-3017-2020, 2020.

Zwinger, T., Greve, R., Gagliardini, O., Shiraiwa, T., and Lyly, M.: A full Stokes-flow thermo-mechanical model for firn and ice applied to the Gorshkov crater glacier, Kamchatka, Annals of Glaciology, 45, 29–37, https://doi.org/https://doi.org/10.1029/2006JF000576, 2007.

---

## Author Comment (AC2)

**Response to Anonymous Referee #2, 24 May 2021**

Dear colleague,

We thank you for the careful reading of the manuscript and the constructive remarks. We have discussed all points raised by you and accordingly changed the manuscript. In the following there is a point-by-point response and discussion of the issues addressed in the review.

There is one thing we want to mention. It seems to us that we mislead the reviewers in that respect that we give the impression that we are up to determining one perfect coefficient $C$, which can then be used for all future simulations of firn densification. This is not the case. We are using the optimisation approach to test if the constitutive relation by Alley (1987) in its particular form can be used to simulate firn density profiles fitting observed ones. We then analyse if the magnitude of factors $C$ leading to the best optimisation results exhibits a dependence on another system variable (e.g. surface mass balance), to see if in future, one could extend or improve the constitutive relation. Alternatively a better representation of the stress could lead to better results in the future. The study is not about finding the best factor $C$, but that a minimum in the optimisation process exists and how much off density profiles simulated using the constitutive relation by Alley (1987) are compared to measured firn density profiles.

Sincerely,

Timm Schultz on behalf of the co-authors
* * *
**Summary and general comments:**

The manuscript purports to be based on the grain boundary sliding model presented by Alley (1987), but the authors wrap all of the grain structure parameters from Alley into a single parameter $C$, which makes their model quite similar to other macroscale firn models for stage 1 (albeit with a different dependence on the current density). The authors do point this out eventually, but I would like more discussion about this assumption. Is it still a grain boundary sliding model, and why?

> The aim of the study is not to develop a new firn densification model. This is not possible with the methods introduced and/or contradicts the idea of a physics based densification model. In this sense it is wrong to attribute the model used throughout the study to the authors. The aim of the study is to answer the following questions:
>
>   (i) Is a constitutive relation in the form of Alley's constitutive relation convenient to simulate firn densification below the critical density?
>
>  (ii) How does the modification of the constitutive relation introduced by Bréant et al. (2017), which alters the domain in which grain boundary sliding occurs, affect simulation results?
>
> (iii) Can we identify hidden or additional dependencies in the description of grain boundary sliding?
>
>  (iv) How could an improvement of the constitutive relation by Alley look like?
>
> In order to clarify this, we explicitly formulate these aims at the beginning of the revised version of the manuscript and discuss them at the end.

Related: The authors use RMSD to quantify the misfit between their model variants and firn-depth-density profiles – but are any of these variants significantly better than previously published firn densification models for stage 1? I think it would be a relatively simple exercise to run, e.g. the Herron and Langway (1980) or another firn model, for these sites to show that the authors' model provides an improvement.

> It makes no sense to compare our simulation results to results from another model. Our model results would most certainly show better agreement with the measured firn profiles than any other firn densification model, because the optimisation process provides an optimal factor for every firn profile. The aim of the study at this point is not primarily to find a global factor $C$ fitting all climatic conditions. We would be happy if we found one, but to evaluate if it would be possible using the constitutive relation by Alley (1987). We point this out more vividly in the revised version of the manuscript.

A pesky problem for the firn modeling community is layering in the firn. The core the authors chose to highlight shows significant layering. Like other models, their model does not capture the layering; but, I suggest that a physically based model should be better at simulating layering than the macroscale models. As the authors are claiming that their model formulation is more physically based, a bit of discussion about this model deficiency would be welcomed.

The modelling of firn layering is indeed a problem, highlighted by the growing amount of high resolution measurements, but often neglected in firn density modelling. We assume impurities influencing the deformation mechanisms in firn to drive layering on the small scale like pointed out by Freitag et al. (2013). This influence is reflected in the activation energy used to describe either the different deformation mechanisms driving the firn densification or in the activation energy used in empirical models, describing the average of all processes.

Therefore layering could be included in the presented model via the activation energy for the process of boundary diffusion. We actually conducted tests regarding this topic in the past. However, this requires information about the impurity concentration, which is not available on a global scale. The used constant value of the activation energy can therefore be understood as an temporally averaged value, leading to a mean density. By the use of the root mean square deviation as cost function, the optimisation result is however not affected when using layered firn profiles as a basis.

We address these points in the revised version of the manuscript. Additionally we include a running mean density for core ngt03C93.2 in Figure 2 a) to illustrate how the optimisation results relate to the mean density.

I would like to see a more detailed uncertainty analysis as a part of the discussion (or as its own section). Not until the end of the conclusion do the authors mention that the results will be affected by biases in firn model forcing data that comes from a regional climate model; this is true, and it should come sooner. Additionally, how do other assumptions contribute to uncertainty, e.g. the assumption of surface grain size and Arrhenius activation energy? To what extent is $C$ just accommodating other errors – e.g. the authors assume that the model calculates grain size $r$ accurately using the Gow formula – but errors are introduced by the boundary condition, and this formula is just an empirical relationship itself. The authors propose a linear relationship to predict $C$ from SMB. If this relationship exists, they should include the equation for the line, and use the value of C predicted by it to run their model. How do those results compare to the outputs from the optimized model? Are they acceptable? Are they better than predicted by a previously published model?

Possible errors are indeed accommodated for in the factor $C$ using the presented optimisation scheme. This includes random errors as well as systematic errors. The idea is to suppress random errors by using a great amount of firn density profiles and then searching for systematic errors to evaluate the constitutive relation by Alley (1987). However, systematic errors can also be introduced by other relations incorporated in the model, such as the Arrhenius equation describing the process of boundary diffusion or the formula by Gow (1969).

As the influence of the activation energy is rather great, we neglected the Arrhenius equation in two of the variants to determine its influence on the resulting factors $C$. We discussed its influence on the results briefly in the current version of the manuscript and will discuss it in more detail in the revised version. It has to be mentioned though, that the value of the activation energy is not determined by empirical measures. It is relatively well known as it describes the physical process of boundary diffusion. We further discuss the influence of the grain radius and its evolution in the revised version of the manuscript.

We don't want to give the impression that the factor $C$ can be predicted from the surface mass balance. Using the optimisation scheme we generated firn density profiles matching the measured firn profiles as good as possible using our particular model setup. We then evaluate the site specific values of the factors $C$ to assess the constitutive relation for grain boundary sliding and to find possible hidden or additional dependencies that may improve the relation. We found a possible additional dependency of the strain rate on the surface mass balance and interpret this as an insufficient description of the stress regime.

As these points were not formulated clearly enough in the original manuscript, we highlight them more clearly in the revised version.

**Specific Comments**

**Introduction**

Line 15: the first paragraph should be rewritten – it is filled with grammatical errors and does not contain citations. I would argue that cold content (Vandecrux et al., 2020) and ice lenses (MacFerrin et al. 2019) are just as important for governing melt water retention. I do not disagree that firn densification is important for meltwater retention, but I do not think this paragraph adequately makes a strong case that it is. The last sentence implies that current models are not adequate but does not provide evidence that they are not.

> We reworked the paragraph for grammatical errors. Indeed, we do not cite papers on melt water percolation and retention, but this is due to the fact that the entire work does not deal with melt water - the problem is even in cold conditions tricky to solve! Many thanks for raising to cite Vandecrux et al. (2020), we will be citing the newest work, from 2021 (Vandecrux et al., 2021), which was actually published a month after our submission, but contains very valuable data for our discussion.

L22: I would not say that the other models neglect the overburden stress exactly – for models operating at small-ish (sub annual) timesteps, they use the mean accumulation rate over the lifetime of a firn parcel (Li and Zwally 2011, 2015), which is effectively the stress, rather than the mean site accumulation rate.

> The addressed models are based on the Robin hypothesis (Robin, 1958) which states that the density change is proportional to the change in stress resulting from accumulation. This hypothesis is basically an interpretation of the linear momentum balance in the form of the one dimensional continuity equation. Hence the stress is implicitly integrated in these descriptions. Li and Zwally (2011) however point out very clearly that the static model approach by Herron and Langway (1980) will not lead to densification if there is no accumulation which is wrong and therefore needs further adjustment. Namely the integration of the accumulation rate over the firn column, which again is an interpretation of the momentum balance in a one dimensional case.
>
> In fact many of the models following the modelling approach by Herron and Langway (1980) use a description more directly linked to the stress. To clarify this and avoid misunderstandings the sentence under consideration and the following one are removed from the manuscript.

L26 – 35: Since your paper is about modeling grain boundary sliding in stage 1, I think it would be prudent for you to focus on how the models handle densification in stage 1. That is, you talk about sintering before you talk about grain boundary sliding (paragraph starting on line 32) – this is just distracting.

As we focus on grain boundary sliding the paragraph about sintering and hot isostatic pressing is removed in the revised version of the manuscript.

L29: Arnaud and Goujon are actually the same firn-densification model, Goujon just added heat conduction to the Arnaud densification physics. I think it is worth noting at this point that Anaud/Goujon used Alley (1987) equations for stage 1 densification because that is what your paper is about. Their equations for stage 2 densification are based on Maeno and Ebinuma.

Besides the models by Arnaud et al. (2000) and Goujon et al. (2003) also the models by Arthern and Wingham (1998) and Bréant et al. (2017) used the description of grain boundary sliding by Alley (1987). The authors intended to highlight this fact with the sentence beginning in line 37 of the manuscript.

"Since then the description of this process was used in various other firn densification models (Arthern and Wingham, 1998; Arnaud et al., 2000; Goujon et al., 2003; Breant et al., 2017)."

To make this even more distinct, the sentence is adjusted in the revised version of the manuscript.

L36: change to "first applied grain boundary sliding theory ..."

The formulation is changed accordingly in the revised version.

L36: be specific of what you mean with "low density" – less than the critical density, I think.

In fact density values below the critical density are meant here. We specify this in the revised version.

*Alley (1987) first applied the theory of grain boundary sliding, adopted from Raj and Ashby (1971), to firn densification at densities below the critical density of $\rho_c = 550 \, \mathrm{kg \, m^{-3}}$.*

L39: numerous times: cite numerous studies

The revised version of the manuscript includes additional references.

L40: "tried to" – colloquial, vague language. Hypothesized? Showed?

The sentence is changed accordingly.

*For example Theile et al. (2011), by conducting experiments on a small number of snow samples, suggested that the densification is more likely driven by processes within the grain than by the inter granular process of grain boundary sliding.*

L48: "in the sense of Alley" → "Following Alley,"

The sentence is changed accordingly.

**Section 3.1**

My general feeling with Section 3.1 (and much of section 3) is that it is distracting from the what the authors state is the goal of their paper (at the end of the introduction), which is to evaluate the efficacy of a grain-boundary sliding densification model. I think that these details of the numerical treatment help the paper achieve that goal. My suggestion would be to move this section to an appendix at the end of the paper. Perhaps I feel this way because I got lost in some of the details, which I attribute to several instances of imprecise language (See below comments). I am admittedly not familiar with the updated Lagrangian formulation, but I don't think that understanding the numerics of that numerical scheme are important for understanding the results (hence my suggestion that this section be moved to an appendix). The meat of the paper begins with section 4, so reduce section 3 to only what is necessary.

The authors would like to maintain the structure of the manuscript in its current form, but if recommended by the editor, we are flexible to adopt a different structure. Sections 3.1 and 3.2 are serving the understanding of the method as they describe the numerics and the constitutive relation for grain boundary sliding, while the remaining sections are rather short.

As the authors all have some kind of background in applied mechanics, we see an importance of a precise explanation of the numerical method and its basis in continuum mechanics. With regard to some of the results from the study an understanding of the treatment of stress and its link to the quasi static momentum balance within the model should be provided.

Furthermore we see that in other studies the description of large parts of the used models are not explained. This may become a problem in our opinion as model results are not reproducible. For example the presented method is most likely used with other model approaches but not specified as such properly. The term "Lagrangian" is often not used properly.

A precise and comprehensive model description also helps other researchers to get started in firn densification modelling in particular and in modelling in general. The cryospheric sciences attract researchers from all kinds of research fields, often not used to the established practices. This also includes for example mathematicians or computer scientists interested in efficient implementations and numerical schemes.

The description of the constitutive relation for grain boundary sliding by Alley (1987) is of great importance as it is the central part of the manuscript. However, we will revise Section 3 to establish a more precise language and improve its understandability.

Equation 1: you say you let $v_b = v$, but when you do that, the left term implies that the time derivative of the density (i.e. the densification rate) is zero – is that what you mean? I am not sure what you are getting at with this.

> "the left term implies that the time derivative of the density (i.e. the densification rate) is zero"
>
> This might be a misunderstanding. The integral of the mass density over a given control volume is the mass, not the density itself. Therefore the time derivative of the mass in just that control volume is zero. Conservation of mass, one of the principles of continuum mechanics, is given.
>
> In a Lagrangian or material based description the density can change by change of the control volume. Mass can not be created or disappear. In contrast to this in an Eulerian or location based description the control volume is fixed and mass inside the volume can change due to inflow and outflow through its surfaces. Hence the density changes accordingly. Conservation of mass is then established with respect to the overall system described.
>
> As these are fundamental principles of continuum mechanics we don't see any benefit in repeating them in the manuscript, especially as the referee notes the section is quite large already. The book by Haupt (2002) on continuum mechanics, which we refer to, allows the reader to get basic information about the addressed topics if desired. The described method however is not entirely a Lagrangian description as the physical properties are solved in an Eulerian reference frame for every time step. We will revise section 3 to improve the understandability for the reader.

Line 71: I am confused a bit with $v$ and $v_b$ – can you clarify the language? You use multiple terms ("flow velocity", "boundary velocity" and "material velocity", and later "grid point velocity"), but only have the two variables. Please choose one term for each variable and stick with that. It is also not clear which velocity is which – e.g. with "boundary velocity", are you talking about the boundary of one grid point/control volume, or the boundary of the entire model domain (i.e. downward advection due to underlying ice dynamics?).

> As this is definitely true, we will refer to the velocity $v_b$ as "grid point velocity" in the entire section in the revised version of the manuscript. Additionally we will consequently add the symbols $v$ and $v_b$ in the text, to make the distinction between the different velocities easier.

L74: "build" → built

> The mistake is corrected.

Equation 2: please use a different variable for stress – you already use $t$ for time (eq. 1). This gets especially confusing because Alley (1987), equation 5, matches your equation 5; but your $t$ has a different meaning than the $t$ in the Alley paper. I don't think it is sufficient to differentiate between $t$ and $t_{zz}$. Alley used $P$ for pressure; Morris and Wingham (2014) and Arthern et al. (2010) used $\sigma$.

We introduce the symbol $\sigma_{zz}$ for the stress in vertical direction in the revised version of the manuscript to avoid confusion with the time $t$.

Nevertheless we want to hold on to the differentiation between $\boldsymbol{\sigma}$ and $\sigma_{zz}$. In a Cartesian system stress can be represented by a tensor of rank two, commonly the Cauchy stress tensor $\boldsymbol{\sigma}$. Pressure is usually defined as one third of the trace of this tensor. Alley used the symbol $P$ for the "overburden pressure" defined as the product of the accumulation rate, acceleration due to gravity and time since deposition of the described firn parcel. This reveals a common inaccuracy as this definition does not describe some kind of pressure but the stress in vertical direction $\sigma_{zz}$. We want to emphasize that $\sigma_{zz}$ is the stress in vertical direction as our interpretation of the study results states other entries of the stress tensor $\boldsymbol{\sigma}$, describing stress in horizontal directions, may be considered in future approaches to firn density modelling.

Equation 4: With this formulation, it seems that in the accumulation zone the model domain will continue to thicken indefinitely?

This is correct. In this study we address this problem by removing grid points within the computational domain if a certain limit of grid points is reached. This limit however is chosen in a way that guarantees a high quality representation of the modelled domain. In the revised version of the manuscript we will add a short comment noticing this point and a description of how this problem is solved.

*This process results in a growing number of grid points. Therefore grid points are removed from the extending model domain at its base when a maximum number is reached.*

**Section 3.2**

write Equation 5 as a part of a sentence (as you have done with other equations)

Equation (5) will be integrated in the first paragraph of the section in the revised version of the manuscript.

*Stress term in Eq. 5 – Alley (1987) makes a steady-accumulation assumption. In this paper it appears that you do not – see comment regarding this at Section 4 comments.*

This is true, Alley (1987) uses a steady-accumulation assumption for his simulations. The derivation of the model however does not need this assumption. The densification rate depends on the average force per grain, linked to the so called overburden pressure, which itself is derived from the accumulation rate. A changing average force per grain due to a changing accumulation rate does not conflict with the model.

Just afterwards the derivation of the constitutive relation Alley makes the assumption that certain quantities, including the accumulation rate, "are constants at any site assuming isothermal conditions". It can be assumed that Alley used this formulation because high quality time series of accumulation data were sparse in 1987. Nowadays we have access to such data and can use it to improve firn densification models.

However the revised version of the manuscript will address the point that the formulation in the original paper is different. Please see also our response to your comments regarding this difference to the original work by Alley (1987) in Section 4.

*Line 107: "seeming arbitrary at first" is a subjective comment (is it more arbitrary than the 5/3?); remove it.*

Both 2/15 as well as 5/3, as pointed out later in the paragraph, are not arbitrary. The authors intended to highlight this fact by using this specific wording. However as it seems to be mistakable it will be removed in the future version of the manuscript.

*Do not use "term" and "factor" interchangeably. (e.g. you use "term" to describe $8D_{\mathrm{BD}}\Omega/k_bTh^2$ but this is a factor, not a term.)*

In the revised version of the manuscript the word is used consistently.

*Line 117: "kind of fade out behavior" – please use specific language and avoid colloquialisms – "fade out" is used several more times, but I think you can be more precise, e.g. how does it fade out? e.g. "the influence of grain boundary sliding on the strain rate decreases asymptotically as the density approached the critical density".*

We reformulated the parts under consideration in the revised version of the manuscript. The term "fade out" is avoided in the revised version of the manuscript.

*The vertical strain rate $\varepsilon_{zz}$ decreases with increasing density $\rho$, until it becomes zero at the critical density $\rho_c$.*

Line 120: previously you said $T$ "resembles the temperature" (what does that mean?), and here you explicitly say that $T$ is the temperature.

> The improper formulation "resembles" is removed from the revised version of the manuscript.

Line 120: you previously said $r$ is grain radius, and here you redefine it slightly differently ("grain size" is a broader term than "grain radius"). Which is it? Please be more specific of what you mean by grain radius, because the grains are not perfect spheres and different people mean different things when they talk about grain radius.

> The revised version of the manuscript solely contains the term "grain radius". Furthermore we added a definition of the grain radius. The model by Alley (1987) indeed assumes perfectly spherical grains. As though this is not true for firn, the assumption of such geometry seems suitable as a model approach. As this also affects the simulation of the grain radius, more details are presented in section 3.6.

**3.4**

L128: Again, I think you have just not explained your method clearly – what is a "test volume"? This is the only time you use that phrase. Upon first glance, this sentence says that the densification rate is zero; the next sentence clarifies this a bit – but not until the third sentence do I figure out that your control volumes are changing. The upshot is that I don't doubt that your numerical scheme is correct; I think it is not explained clearly.

> In the revised version of the manuscript we will take heed of consistent wording regarding control or test volumes.
>
> The integral over the density is the mass, not the density itself. Please see our answer to your comment on Equation (1) for a detailed explanation. The authors think that the terms "Langrangian description", "material fixed coordinates" as well as "control volume with moving boundaries $z_1(t)$ and $z_2(t)$" (see beginning of section 3.1) pretty distinctly introduce the concept of a changing control volume. We will revise section 3 to establish better understandability.

**3.5**

Line 142: This is an incomplete sentence. Discretized by what? Or maybe it is just written oddly?

We reformulated the sentence to make it better understandable.

*As pointed out in Section 3.1 all advection in the model domain is represented by the moving grid. Therefore the description of temperature evolution reduces to simple heat diffusion:*

Line 145: please cite page numbers with textbooks.

In the revised manuscript we added page numbers when citing monographs.

Equation 11: Is your equation for conductivity your formulation? I do not see it in Paterson 1994; please either cite source material or state that you came up with this parametrization (and how).

In fact a mistake happened here. During the development of the study we tried different descriptions of the thermal properties for firn and things got mixed up. The relation shown for the thermal conductivity is adapted from Arthern and Wingham (1998). However, this is not the description for the thermal conductivity we used throughout the study. We followed the example of Zwinger et al. (2007) and used the description by Sturm et al. (1997). The new version of the manuscript features the correct description and the corresponding citations.

**3.6**

Please provide what you use for a surface boundary condition for the grain size.

At this point of the manuscript we want to show the equations solved during the simulation process, mainly independent of the use case. We mention that a suitable boundary condition for the grain radius is needed in order to solve the equation and provide a reference to Section 5.2.
* * *
"This is suitable as the grain radius is used in the constitutive equation by Alley (1987)" – please
justify this assumption. Also, second use of word "is" appears to be a typo.

The mentioned formulation has been rewritten as part of the revision of section 3.6 to
clarify the computation of the grain radius. This includes the definition of the grain radius
and the relation of the grain radius to the mean grain cross-sectional area as defined by
Arthern et al. (2010).

*The evolution of the grain radius $r$ is simulated using the well known description of
Stephenson (1967) and Gow (1969) as given in Arthern et al. (2010). Stephenson (1967)
and Gow (1969) describe the grain size in means of the mean cross-sectional area. Arthern et al. (2010) however assume the mean cross-sectional area to be $A = (2/3)\pi r^2$ and
formulate the grain growth rate as ... This formulation allows for simple calculation of the
grain radius $r$.*

**Section 4**

Alley (1987) did not include heat diffusion (he assumed isothermal conditions), and he assumed
steady accumulation. How does your inclusion of it affect your results vis-à-vis those of Alley?
Please provide justification that it is appropriate to "test the concept of the material model
developed by Alley.

The authors see no reason why a more detailed description of the temperature and the
stress due to accumulation should not be justified. The model by Alley describes the
process of grain boundary sliding which depends on temperature and stress. It is in no
way adjusted to average values or steady state conditions. This is one of the advantages a
model describing the physical processes of firn densification would provide in contrast to
empirical models.

Furthermore the model was used in this way before (Arthern and Wingham, 1998;
Goujon et al., 2003). Arthern and Wingham (1998) tested how the model responds to
varying boundary conditions at the surface in contrast to steady state conditions. They
used random generated values as "information about the spatial scale of climate fluctuations is limited by the scarcity of meteorological stations and the large distance between
core drilling sites". Data products like RACMO now provide that data. Using constant
temperatures like Alley did in 1987 would provoke the question why available forcing data
was not being used. The same is true regarding the grain radius.

Nevertheless we address the fact that transient boundary conditions are applied to the
model before in Section "5.2 Boundary Conditions and Forcing".

L172: How does lumping these parameters together affect the results? I would expect that they will vary by site/climate, and so what does it mean for your grain boundary sliding model that you remove the specific parameters that describe grain geometry?

> The approach decouples the resulting depth density profiles from the parameters. Indeed the optimisation result in the form of the factors $C$ is site specific. This is the point of the study. We test if Alley's description of grain boundary sliding is convenient to model the density within the first stage of firn densification, if optimal parameters were used. And we analyse the resulting site specific factors afterwards regarding possible hidden dependencies.
>
> As this idea is not intelligible to the reader from the first paragraph in section 4, the authors point it out more clearly in the revised version of the manuscript. Please also see our answers to your comments on Section 6 regarding this topic.

Eq. 14 and 15: You are adding an Arrhenius term, which was not included in Equation 5 (but is of course included in many other firn models. Given that your paper is about optimizing the grain boundary sliding approach, can you justify this addition, or add text describing why you felt it was appropriate?

> In fact the same Arrhenius term is indeed included in Equation (5) ($D_{\mathrm{BD}}$). To highlight it the revised version of the manuscript includes a description of $D_{\mathrm{BD}}$ in Equation (5) in the same manner as in Equations (14) and (15).

L187: what does "its" refer to? i.e. to test what's influence?

> To clarify the sentence its structure is changed in a future version of the manuscript.
>
> *To test the influence of the Arrhenius law it is disregarded in Variants 3 and 4 as shown in Equations (16) and (17):*

L191: It is not clear to me why a possible dependency (of the strain rate?) will reflect in the optimal values of $C_{\mathrm{v}3}$ and $C_{\mathrm{v}4}$? There is still a factor of $1/T$ in the equations.

> As the addressed sentence is mistakable and provides no important information it is removed from the manuscript in the revised version.

L193: → "The aim of the optimization . . . "

> The missing "The" is added in the revised version of the manuscript.

L195: Where is the Wilhelms core taken from? Please mark in on the map (figure 3) in a different color and provide coordinates. I suggest coming up with an easier-to-read name (i.e. state the full name once here when introducing and citing it, and give it an easy-to-recognize abbreviation to use for the remainder of the paper). I suggest this because I get concerned when I am reading if there are multiple cores with similar names that I need to be differentiating between.

Figure 3 features the location of ice core "ngt03C93.2" in the revised version of the manuscript. Additionally the coordinates are provided.

Consistent and unambiguous naming of datasets, no matter what kind, is of great importance. Therefore, the authors would like to stick to the name "ngt03C93.2", although we note the name is somewhat cumbersome. The likelihood of confusion would be greater when using an abbreviation because the abbreviation could for example just as well refer to ice core "ngt06C92.2". Aside from that "ngt03C93.2" is the only core featured explicitly in the manuscript.

L200: this implies that Lundin and Verjans did not use objective measures for their studies, which is not the case. Verjans did in fact use RMSE. I will also suggest that RMS deviation is not wholly objective in this scenario (which is not to say it is not useful) – imagine that you had a scenario where you were able to model the density perfectly at all depths except a single depth where it is off by $20\,\mathrm{kg\,m^{-3}}$. If this misfit point is near the surface where the density is low, it is a much higher percent misfit than at greater depths where the density is greater.

Both studies, the one by Lundin et al. (2017) and the one by Verjans et al. (2020), used the depth integrated porosity (DIP) for comparison. Verjans et al. (2020) in fact used the RMSE, but to express differences between different values of the DIP.

There is one drawback of the DIP: Let's assume the DIP of a firn profile has the value $d$. When one flips the same firn profile vertically, the highest density now at the top, the lowest at the base, the DIP is still $d$. Actually, when one imagines the firn profile discretized at certain points in space, one can rearrange the density values at these points in every possible way, the DIP remains $d$. DIP only works, because the density profiles are always rather similar to each other, but is not unambiguous.

Nevertheless we agree with the reviewers point that the RMSD is also not entirely objective. In the revised version of the manuscript the references to the studies by Lundin et al. (2017) and Verjans et al. (2020) (great study by the way) will be removed. The root mean square deviation will not be called objective anymore.

L209: It occurs to me at this point that you have not described how you are modeling densification in stage 2 – ostensibly your model goes beyond stage 1 densification because this is important for getting the temperature profile/heat transfer correct. With the Arnaud/Goujon/Breant models, there is a factor introduced to allow the densification rates to match at the critical density – are you doing something similar? I do not think you need to provide a lot of detail but you should add a sentence describing what you do for density greater than 550.

In this study, we focus only on the first stage of densification, therefore, we do not compute the densification further than the end of this stage. Indeed, if we expand the simulation into further depth, the transition is more of an issue.

What is important from our perspective is that our focus is indeed on assessing the process of grain boundary sliding and not (yet) the best approach for the transition. From our experience with running simulations incorporating the following stages this is the most tricky point to solve. If the goal is to get the best estimate for delta age, or the best close-off density depth, this is by far more of an issue than only checking if the constitutive relation for the first stage is appropriate. Therefore, we think that we are basically ok, due to the low depth at which the first stage ends.

We do not develop the density beyond the first stage and the critical density. Regarding the computation of the temperature a spin-up and consideration of the following stages of densification would be the best choice. We have not done that in this study, because not many of the analysed firn profiles extend below the depth of 10 m to 15 m, which is the depth at which the mean annual surface temperature is established (e.g. Cuffey and Paterson, 2010). We are not investigating seasonality at this point.

In the revised version of the manuscript we cite the article by Vandecrux et al. (2021) which has been published after our submission. The study presents a firn temperature profile at "Camp Century", which shows that a Neumann boundary condition at the depth we are applying it is not an issue, as the temperature change below the depth of 5 m is insignificant for our approach. The study by Orsi et al. (2017) shows that the temperature change in the upper 150 m of firn is only about 1 K, which is also small in context of our work.

In order to assess what influence this rather small temperature change has on the outcome of our study, the revised version of the manuscript contains a simulation in which we have reduced and raised the temperature at one site. We check whether the ability to find an optimal value throughout the optimisation process is reduced. As expected, we find that it has little influence on the assessment of the constitutive relation.

L214: Please be more specific about what tests you did.

The sentence is altered in the revised version of the manuscript so that the process leading to the presented range boundaries is comprehensible.

*To ensure optimal factors are found within the presented range boundaries the all simulations were performed multiple times.*

L215: A general note: Thank you for consistently including the units with your variables and equations.

L223: It would be helpful to explicitly say that there are 21 densities you tested, e.g. "we tested 21 different values of the surface density . . . "

We include the number "21" in the revised manuscript.

**Section 5**

Section 5: I suggest moving section 5 to before section 4. At this point my feeling as a reader is that I have made it to the 12th page and am finally getting to the part of the paper where you tell me more specifics of the science you did (this is also related to my suggestion that you move a majority of section 3 to an appendix.)

The authors would like to maintain the current structure of the manuscript. In terms of Section 3, please see our response to your comment, regarding the suggestion to move it to an appendix.

In our opinion the order of Sections 4 and 5 should not be changed to first introduce the methods, and then present the data used within the study. Furthermore, the choice of data is based, at least to some degree, on the used method. The main scientific idea is related to the optimization scheme in conjunction with the physically based model of grain boundary sliding. The considered data was gathered by other researchers and is not in the focus of the present investigation.

Figure 2: Since you include the SMB and temperature in a plot, it might be useful to also include the mean of those as predicted by RACMO.

Panel c) of Figure 2 also features the mean values of the surface mass balance and surface temperature over the course of the simulation time in the revised version of the manuscript.

L232: Be specific of how many sources, or leave that out. Probably you could just remove your short description of what SUMup is (the fact that is is available online and has a lot of measurements is not germane for the point of your paper)

> The sentence is removed from the revised version of the manuscript as suggested by the reviewer.

L241: Positive SMB and melt are not exclusive - quite a few sites experience significant melt in Greenland yet still have positive mass balance. Quite a few of your sites, as far as I can tell on your map, are in wet firn locations. How did you handle these sites with your simulation? Your model handles densification due to compaction but not densification due to meltwater percolation and refreezing. How does this assumption affect your results? Do you see different optimal values of $C$ for sites where melt $> 0$? Or higher optimal surface densities?

> We are aware of the fact that a positive annual surface mass balance may include periods of melting. The data basis for this study is mean annual data from RACMO. A discrimination of those sites including melt and those which do not, is not possible using these data. We assume that the relatively low number of firn profiles influenced by melt compared to the overall number of sites does not affect our results, as we do not aim to find a specific value $C$ representing all firn profiles (see also our responses to your comments regarding the discussion section). In the revised version of the manuscript we comment on this problem.

L255: Except it appears that the cold, low-accumulation center of the East Antarctic ice sheet is not represented, and this is one place of high interest for firn studies because of its relevance to ice core studies. (The Arnaud model specifically used Vostok as a test site).

> This is most certainly a problem, we address it in the sentence just before. The Vostok core used by Arnaud et al. (2000) is not included in the SUMup dataset. Arnaud et al. (2000) cite Barkov (1973) as the original source. It seems, Koenig and Montgomery (2019) were not able to get hold of the original report and more relevant the data. Many cores from this era are not published in a usable way. The authors strongly encourage everyone in the possession of unpublished firn density data to contribute to the SUMup dataset.
>
> However the authors consider the SUMup dataset the most extensive dataset of firn density data available at this point and decided to focus on these data if they match the criterions presented in Section 5.1.

Section 5.2: I suggest rewriting the first paragraph of this section for clarity.

As suggested by the referee the paragraph has been rewritten.

*To force the firn densification model, surface values for density, temperature, accumulation rate and grain radius at the locations of the 159 firn profiles are needed. Although Alley (1987) used constant forcing we follow the example of Arthern and Wingham (1998) and Goujon et al. (2003) performing transient simulations.*

*As measured firn density profiles represent past climate conditions, the choice of forcing data is crucial for the presented method. Uncertainties in the forcing will reflect in the simulation results and therefore in the comparison with measured firn profiles. Neither the model formulation nor the optimisation scheme can compensate for that. We use data provided by the regional climate model RACMO2.3 (Van Wessem et al., 2014; Noël et al., 2015)5. RACMO2.3 provides forcing data for the Greenland ice sheet covering the period from 1958 to 2016. In case of Antarctica the time period is shorter, starting in 1979 and ending in 2016. Data for the mean annual skin temperature and surface mass balance for the Greenland ice sheet are available at mean spatial resolutions of 11.3 km and 1.0 km respectively for this study. Mean spatial resolutions for Antarctica are 8.0 km and 28.5 km for mean annual skin temperature and surface mass balance. Spatial interpolation of the fields leads to forcing data for the locations of the investigated firn profiles.*

L259 – 268: I am confused here. You said earlier that your model time resolution is 48 steps per year (roughly weekly), but you say here that you are using annual data from RACMO. Why do you not use the daily RACMO data downsampled to weekly resolution? Are you using the same value for the temperature and accumulation for each of the time steps during a given year? On Line 260 you say you are interpolating, but what are you interpolating (spatially? temporally?), and how?

We clarify that spatial interpolation is meant here in the revised version of the manuscript. We found that using forcing data with a higher temporal resolution affects the resulting firn density profile little compared to the results using mean annual data. However, we have not included these findings in the manuscript. We add them in the revised version. The time resolution of 48 time steps per year ensures for high spatial resolution of the simulation results, helping to compare results with measurements, and guarantees for numerical stability. We address this in the future version of the manuscript. Despite being used in many studies, data from RACMO are not freely available.

L269: In the cases where you have cores that have measurements from the near surface, why don't you just use the measured surface density from the cores? Are there cases where the surface density from the optimization scheme is significantly different than the observations from SUMup? It seems possible that your scheme could allow a surface density that is different than the observation in order to get a better overall RMSD fit. Do you use the same density at all time steps for a given site/simulation?

Even when near surface density values are available, it is not trivial to determine a surface value. Strategies could incorporate the computation of mean values over certain depths, fitting of more or less complex functions over certain ranges, combinations of both, and so on. Difficulties arise when applying one method for a great number of profiles because profiles start at different depths, the spatial resolution differs, the density varies. Firn profiles are very unique especially at low density.

We have worked out during our study that the approach presented in the manuscript works well. In case near surface values of the density are available the surface density resulting in the best match between simulation result and measured firn profile represents these values well due to the use of the root mean square deviation as cost function. For the reasons pointed out before it is difficult to quantify this. We do not distinguish between those profiles from which near surface values are available and those starting at greater depth, to maintain comparability. The surface density is constant with respect to time for a given site.

The revised version of the manuscript addresses these points.

L270: What is the basis of your choice of 0.5 mm for initial grain size? Is it a realistic assumption to assume it is the same everywhere? (I would think not – southern Greenland is quite different than e.g. the South Pole). How sensitive is your model to this parameter choice?

The choice of using the value 0.5 mm for the initial grain radius is mainly based on the study by Linow et al. (2017). The effective radius defined in this study can be understood as radii of spheres having the same specific surface area as measured throughout the study. Linow et al. (2017) derive an empiric relation for the surface grain radius from their measurements. It depends on the mean annual temperature and mean annual accumulation rate.

Applying this empiric relation to the RACMO data representing the locations of the analysed firn profiles leads to a mean surface grain radius slightly below 0.5 mm. 0.5 mm is also the grain radius used in the study by Arthern and Wingham (1998).

Nevertheless data for the grain size at the surface are sparse. Even the study by Linow et al. (2017) is based on only a few firn cores and features only one core from Greenland. The choice of a value constant in space and time, might not be realistic, but pragmatic. Tests throughout the study have shown the model is not very sensitive to this parameter. Furthermore because of the optimisation approach the initial grain size is less relevant. The change of the grain radius with depth is the important part here.

In the revised version of the manuscript we will address these points.

L274: But aren't the surface densities in good agreement because the range you chose for them is based on what the literature says in the first place? This seems circular.

This is in fact true. In the revised version of the manuscript we dismiss the comparison with other studies.

L277: I think there is literature that you could find to cite regarding Antarctic vs. Greenland surface densities. Or you could query SUMup and find all measurements shallower than 50 cm or 1 m for Greenland and Antarctica.

The number of firn profiles incorporating shallow density data would not be representative for either Greenland or Antarctica. The comparison of such data is difficult. We neglect the discussion of different surface densities at this point on purpose, as we believe it is out of place in this section.

L279 – 280: This seems out of place – should it be in the temperature section in section 3? Also you need to specify what your model domain depth is since you are using a Neumann condition.

Yes, we agree, and it moved to Section 3 in the revised version and we also specify the depth explicitly. Please see also our response to your comment on line 209.

**Section 6**

L290: "even better agreement" – this implies that that core has good agreement in the first place – but does it? You do not provide context for whether $\approx 23\,\mathrm{kg\,m^{-3}}$ is actually a good agreement. I suggest that RMSD is a good metric to intercompare the models' performance – v2 is doing better in general than v3. But it does get murkier when actually deciding if this is a "good" fit to the data – so this would be better if you could normalize the RMSD somehow, and provide a quantification of what you consider to be a good fit (any why).

The question of what is considered good and what is considered bad is difficult to answer. The authors think that the root mean square deviation actually is descriptive and understandable for the reader as it describes, as the name states, a mean deviation given in the unit of the density itself. A simulated density profile with a root mean square deviation of for example $20\,\mathrm{kg\,m^{-3}}$ is maybe off by $20\,\mathrm{kg\,m^{-3}}$ at every point. A value of the depth integrated porosity which is off by a certain value is much harder to grasp.

Along with the revised version of the manuscript we publish plots of the analysed firn density profiles and our optimisation results along with the root mean square deviation in the manner of Figure 2 (a). This allows the reader to get an overview of all results. Together with Figure 5 we think this will help to put the values of the root mean square in perspective.

Figure 5: please label which variant is which in a more obvious manner (the v1 is small and hard to spot).

In the revised version of the manuscript the four subplots contains an additional label.

L291: Isn't this more a function of the SMB (from RACMO) being accurate than it is the strain rate being correct? If you were to optimize your model to fit the depth-age profile best (rather than depth-density), would the same optimal parameters be the same?

> Optimisation for the age results in different optimal parameters. The quality of the depth-age profile does not only depend on the quality of the surface mass balance forcing but to some extent on the quality of the model. The model is driven by the surface mass balance forcing, so that things can not be examined separately. As age data is not available for the entire dataset, and for simplicity, we drop these considerations in the revised version of the manuscript.

L296: This is confusing. Alley (1987) as far as I can tell does not include an Arrhenius term. (Although he did derive an activation energy for grain boundary sliding). This makes me wonder: the activation energy you use is notably smaller than recent studies have implied it should be (Arthern et al. 2010; Morris and Wingham 2014) – how does your choice of activation energy affect your results?

In fact the model by Alley (1987) at first sight does not contain an Arrhenius term (see Alley, 1987, Equation (5)). It does however include the "boundary viscosity" $\nu$. As pointed out in the introduction of the manuscript, Alley (1987) has not simulated the four firn profiles described in his study. He fitted the curves resulting from his model to the firn profiles and evaluated the value of the boundary viscosity $\nu$ leading to his fitting results. He then proposed to use a description of the boundary viscosity $\nu$ provided by Raj and Ashby (1971) (see Alley, 1987, Equation (7)). This description includes an Arrhenius term describing the rate of boundary diffusion.

The activation energy of $44.1 \, \mathrm{kJ \, mol^{-1}}$ is the activation energy for the physical process of boundary diffusion in ice. This value results from measurements of the self diffusion of ice (for example Maeno and Ebinuma, 1983; Itagaki, 1964). Alley (1987) suggests the slightly different value of $42 \, \mathrm{kJ \, mol^{-1}}$ citing the text books by Hobbs ("Ice Physics", 1974) and Paterson ("The Physics of Glaciers", 1981), which should show this value was established in 1983.

The values of the activation energy resulting from the studies of Arthern et al. (2010) and Morris and Wingham (2014) (and many more) do not describe the process of boundary diffusion. These values result from the empirical approach of fitting a model of a certain form, often including an Arrhenius law, to a certain dataset. These activation energies might be understood as an average over all physical processes leading to firn densification. Including boundary diffusion, lattice diffusion, dislocation creep, vapour transport and maybe more (see Maeno and Ebinuma, 1983; Arthern and Wingham, 1998). The aim of this study however is to reassess the description of the physical process of grain boundary sliding by Alley enabled by the process of boundary diffusion.

To clarify these facts the authors mark the relevant activation energies and corresponding pre factors with the index "$(\cdot)_{\mathrm{BD}}$", to point out the process of boundary diffusion is meant. Furthermore the study design of Alley (1987) and the relevance of the boundary viscosity $\nu$ within the study will be repeated briefly in Section 3.2. The used values for activation energy $Q_{\mathrm{BD}}$ and pre-factor $A_{\mathrm{BD}}$ are derived in more detail by inclusion of more appropriate literature especially in section 4.

Note on figures in general (especially Fig. 6): the colors you have chosen can be challenging to see (especially the yellow) when text is written in those colors.

> In the revised version of the manuscript the yellow color #EBCB8B is changed to purple #B48EAD for better readability.

L305/Figure 7: To me these data do not appear to be linear, and the lines do not appear to be good fits (especially at low and high temperatures). A Pearson correlation coefficient is based on the assumption that the data are linear, which I am not convinced they are, so I think it is deceiving to include that metric. To me the most obvious conclusion is that there is not a clear linear relationship between any of the $C$ parameters and temperature. If this is not what I should take away from figure 7, you need to be more thorough convincing me.

> The intention in providing the Pearson coefficient and a linear fit was to give the reader an orientation to determine whether the factors $C$ show a dependency on the mean annual temperature or not. This dependency has not necessarily to be of linear kind as mentioned in the text. However, we acknowledge that this impression could arise.
>
> The main problem of the Pearson coefficient is, that even if it becomes zero, a dependency of higher order may be present. However, it can be used to determine to which extent two vectors are linear. The correlation with higher order functions is difficult to quantify if no concrete function predicting the observed values is known. Nevertheless we will introduce the distance correlation (Székely et al., 2007) in the revised version of the manuscript. It was designed in recent years to explicitly overcome the problems of the Pearson coefficient.
>
> Furthermore we removed the linear fits from Figures 7 and 8 and adjusted the text to clarify our thoughts presented to the reader.

Figure 7: what does a negative value of $C$ indicate? (it is hard to tell if there are any; but figure 7's y axis extends to $-2$, and it looks like there could be? Make the axis the same as figure 8.)

> The figure is altered as proposed by the referee.

310: Again, "even higher" is implying that you had a good fit between temperature and $C$, but the a high Pearson is meaningless because the data in figure 7 are not linear.

> The addressed term is changed in the revised manuscript with respect to the other changes in this section.

L312-315/Figure 8: These do appear to be more linear than the $C$ vs. temperature plot, but I think it would be appropriate to do a statistical test to actually show that they data are linear (e.g. a statistical model with higher order terms that have coefficients not significantly different than zero, or some other test). I am concerned with the values at the high accumulation sites for variants 1 and 2 – there are clusters that are half of what would be predicted by your linear model. And, what is going around $0.4 \mathrm{m\,w.eq.\,a^{-1}}$? (And 0.6?) These vertical series of dots would indicate to me that there is no correlation between SMB and $C$ – i.e. you can have a number of sites with SMB = 0.4 and $C$ can vary by a factor of 10.

> We want not necessarily to imply a linear dependency, but a general correlation. In the revised version of the manuscript the distance correlation provides a correlation measure independent of the assumption of a linear dependency (see our answer to your comment regarding line 305 of the manuscript). Sites with a mean surface mass balance around $0.4 \mathrm{m\,weq.\,a^{-1}}$ show indeed peculiar results. These sites are located in a cluster (hence the very similar conditions) in western Greenland. These sites are influenced by melt. We address this in the new manuscript.

Your discussion should include discussion of why the optimal $C$ values vary by site and how that influences our understanding of grain boundary sliding.

> The idea is to generate the best possible density profiles for each one of the four variants of the constitutive relation and each site. No matter which variant of the constitutive relation is used, the density profile always matches the measure density profile, due to the optimisation. As the resulting density profiles are similar, because the general structure of the constitutive relation is maintained, we can compare the factors $C$ leading to these density profiles.
>
> The differences in the factors $C$ do not primarily arise because of the differences in the resulting density profiles, as these differences are small, but from the differences in the variants of the constitutive relation used throughout the simulation. Nevertheless, also the small differences between the resulting density profiles matter.
>
> We will point out this idea at the beginning of the discussion section in the revised version of the manuscript.

**Section 7**

L323: Again, I suggest that getting depth-age correct is more dependent on getting the accumulation rate correct – and, since the accumulation rate is often determined by counting annual layers, the science can become a bit circular.

> As pointed out before, we neglect these considerations in the revised version of the manuscript.

L326: I agree entirely with what you say in this paragraph – your model formulation is not too different than the other models that have been published in recent years, but up until this point you have been claiming that you are optimizing a grain boundary sliding model (including in the title). I think it would be appropriate for you to rework the text a little bit to be more forthcoming with this, rather than waiting until the very end to point it out.

> We do not want to formulate a new model for the process of grain boundary sliding nor do we want to optimise the existing model by Alley (1987). Hence we are not claiming to optimize a grain boundary sliding model. We think the title states clearly that we do an assessment by using an optimisation scheme. We assess an existing model, the model by Alley (1987). We can not and we do not want to claim this model for ourselves. We assess if the constitutive equation for the process of grain boundary sliding by Alley from 1987 is convenient to model firn densification below the critical density using an optimisation scheme and data not available back in 1987. In the revised version of the manuscript we address the point that we assess the constitutive relation and it's format in the introduction more clearly.

L338: Morris (2018) did in fact create a "transition model" to move from stage 1 to stage 2.

> This is true. Morris (2018) addressed the problem that (semi)empricial models based on the approach by Herron and Langway (1980) a distinct kink in resulting density profiles, because of the different functions describing stage 1 and 2 of the densification by applying another empirical function for the transition.

> "Other functions that produce a smooth transition from one value to another exist and produce much of the same effect."

> This is most certainly an improvement for those kind of models. In the present investigation we are interested in the description of processes leading to firn densification, particularly in grain boundary sliding.

L345: I am confused here – are you suggesting that because you do not have a separate Arrhenius factor in equations 16 and 17 there is implicitly an Arrhenius factor wrapped into $C_{v_3}$ and $C_{v_4}$? If this is the case, say so clearly in Section 4. It seems that you have a baseline assumption in your study that firn strain has an Arrhenius temperature dependence. But, if Variants 3 and 4 give good answers, couldn't this just as easily indicate that firn does not have an Arrhenius dependence? Part of my point here is that you have coefficients, $C_{v_3}$ and $C_{v_4}$, that have all sorts of "physics" wrapped into them, and I don't think you can just decide what those physics are.

We do not have a baseline assumption that a firn densification model has to contain an Arrhenius equation, although it is likely meaningful. The description of grain boundary sliding by Alley (1987) in combination with the description of the bond viscosity by Raj and Ashby (1971) however does. We want to assess this constitutive relation.

The idea is to strip the constitutive relation down to its core. We neglect the Arrhenius law in two of the four variants. Due to the optimisation however, we produce very good results in terms of the density. These results, the depth density profiles, are very similar for all four variants. For the depth density profiles it does not really matter if we include the Arrhenius equation or not. But, because the results are so similar, we can afterwards compare the factors these results. This is the idea of the presented method.

By using one variant incorporating the Arrhenius equation another neglecting it, leading to the same depth density profiles but different factors, we can see the missing dependency on the temperature in the factors. However, as though the results improve when using the Arrhenius equation in terms of the determination of a global perfect factor (which does not exist), we still see a dependency on the temperature to some degree for Variants 1 and 2. This tells us that either the existing temperature dependency is flawed, for example by wrong parameters of the Arrhenius equation, or there's another dependency, so far not respected in the constitutive relation.

However, the paragraph is indeed confusing and does not really address these points. The revised version of the manuscript therefore features a much more detailed description of these thoughts.

L349, 352: "resulting factors" – do you mean the 4 values of $C$ for your variants? $C$ and the surface density? Please use more specific language, e.g. "indicate a clear dependency of $C_{v_3}$ and $C_{v_4}$ on the mean surface mass balance".

This expression is removed in the revised version of the manuscript.

*The determined factors $C_{v_3}$ and $C_{v_4}$, resulting from variants without the Arrhenius equation for boundary diffusion $D_{BD}$, show a stronger dependency on the mean surface temperature than factors $C_{v_1}$ and $C_{v_2}$. At the same time, factors $C_{v_1}$ and $C_{v_2}$ show less dispersion than factors $C_{v3}$ and $C_{v_4}$, as is shown in Figure 6. The inclusion of the Arrhenius equation $D_{BD}$ in the constitutive relation leads to better determination of these factors. It is therefore a meaningful description within the constitutive relation. Although the inclusion of the Arrhenius equation results in better determination of factors $C_{v_1}$ and $C_{v_2}$, we still see a dependency on the mean surface temperature to some degree. A better determination of the parameters of the Arrhenius equation may result in resolving this dependency. If this is not the case another dependency on the temperature may be introduced to improve the constitutive relation for grain boundary sliding.*

L349: The first two sentences of this paragraph need to be rewritten to improve clarity. After multiple readings it is still not clear to me what you are trying to convey.

These sentences are removed from the future version of the manuscript. For reasons see our response to your next comment, addressing the same paragraph.

L350: This is not obvious to me – are you saying that the $C$ factors are functions of temperature, and since temperature and SMB are correlated, then $C$ must also be a function of SMB?

The point the authors wanted to address here is that one could argue that the dependency on the surface mass balance is just a hidden dependency on the temperature because colder sites usually also show less accumulation. As the paragraph is a little abstruse and the referee seems to doubt this point, the paragraph is neglected in the revised version of the manuscript.

L353: What do you mean by "Dependency of the mismatch"?

This formulation is indeed not chosen well. The sentence is reformulated in the revised version of the manuscript.

*We interpret the dependency on the surface mass balance such, that the load situation is currently not represented well.*

L354: You can at least do a scale estimate of the effect that horizontal stresses would contribute – see Horlings et al., 2020. Your paper is about stage 1 densification/grain boundary sliding – on the timescales of firn densification on ice sheets, is horizontal stress going to make a significant difference? I am also skeptical that an 3D model incorporating horizontal stresses would significantly reduce your misfit – at least in the example core you highlighted, the layering of the density profile likely contributes to a significant portion of the misfit.

Horizontal divergence due to the movement of the underlying ice as for example investigated by Horlings et al. (2020) is not what is meant here. The mention of a three dimensional problem also does not necessarily imply a full three dimensional model approach.

As pointed out before, the stress tensor is a tensor of rank 2. Therefore the term "overburden pressure" is misleading as the pressure is a scalar property. An isolated firn column, would not only compress in vertical direction, but also elongate in horizontal directions. But there is no such thing as an isolated firn column, a firn column would be confined in horizontal directions by other firn. One also could imagine a firn column in a pipe. As this firn column is now not able to elongate in horizontal directions, the horizontal components of the stress tensor are not zero, as the firn kind of presses against the walls of the pipe. We therefore propose that in a purely one dimensional model approach the stress might not be represented well.

To solve such a case is not as trivial as it sounds, as the material in question, firn, is compressible. However, model approaches can be found at Zwinger et al. (2007), (Greve and Blatter, 2009, pp. 224–230), Salamantin et al. (2009) and Meyer and Hewitt (2017). The revised version of the manuscript will address these points.

**Section 8**

L367: not represented well in the model – are you effectively saying here that the stress should have an exponent other than 1?

The revised manuscript features a more detailed discussion of our assumptions on the representation of the stress in Section 7. This, together with additional changes in Section 8, clarifies that we don't consider the stress exponent to be wrong.

L371: I agree with this paragraph; I think it should be moved into the discussion section along with a more complete uncertainty analysis.

We moved the paragraph in question to the discussion section in the revised version of the manuscript. We discuss the uncertainties within the forcing and other error sources and their influence on the results in more detail.

---

## Author Response (AR1)

**Answer to Editor decision, 24 Jul 2021**

Dear Ms Keagan, dear Referees,

We again like to thank you for your very helpful contribution in improving our manuscript. Below you can find our point-to-point answers to your latest comments.

We have included another example firn profile in the manuscript to show and discuss the influence of seasonal variations on the optimisation. We also added another section to discuss how the optimisation approach is affected by assumptions we make regarding the boundary conditions of our model. As these additions increased the length of the manuscript further, we decided to move most of the former third section describing the model to an appendix, as suggested by the referees and the editor, to improve the readability of the manuscript. This led to an adjustment of the manuscript's structure. To cite all of the firn profiles used within the study correctly and to give an overview of the optimisation results, we added a table to the supplementary material which features all references and optimisation results.

Again, many thanks for your input and your effort to improve this manuscript.

Sincerely,

Timm Schultz on behalf of the co-authors

Referee #1's second comment: I appreciate that the authors do not have a background in regional climate modeling. Please include a statement in your revised manuscript addressing that local data is found by interpolation and introduces systematic errors into the RACMO data, which may introduce error into your results.

As proposed by the editor, we added a remark pointing out that the spatial interpolation of the forcing data from the different grids used by RACMO2.3 to the locations of the analysed firn profiles introduces errors to the data. It can be found within the Section 3.2 of the revised manuscript.

*Spatial interpolation of the fields obtained from RACMO2.3 output leads to forcing data for the locations of the investigated firn profiles. It has to be mentioned that such an interpolation may introduce systematic errors.*

Referee #1's third comment: Be sure to include information about how 'tests have shown that a seasonal temperature signal has very limited influence on the resulting firn density profile compared to the same model setup using the mean annual temperature', whether it be your own work included or appropriate citations to others' work.

In the revised manuscript we added a second example showing the influence of monthly averaged forcing data on our model and optimisation approach in Section 3.2. This example features a firn core retrieved at site 3 of the iSTAR Traverse conducted in 2013/14 on Pine Island Glacier in West Antarctica (Morris et al., 2017). To force the simulation we used "ERA5-Land monthly averaged data from 1981 to present" (Muñoz Sabater, 2019; Hersbach et al., 2020) as it is freely available. We compare this simulation result with another one using yearly averaged forcing data computed from the monthly averaged data to point out that our optimisation result is very little affected by the higher resolution forcing. Although the detail of the simulated firn density profile improves, we decide to use yearly averaged data from RACMO2.3 because a longer time period is covered and less overhead is produced.

Referee #2's general comment about uncertainty: as both referees highlighted a need for a discussion of the uncertainty analyses in your study, please make sure to thoroughly address this in your revised manuscript.

We added another section to the manuscript titled "Distribution and Influence of Input Data" in which we evaluate different concerns expressed by the referees regarding uncertainties. This new section features discussion on the layering of firn profiles and the use of a Neumann boundary condition.

Referee #2's general comment about Section 3: I respect your wishes to want to retain Section 3 within the paper, instead of moving it to an appendix. I agree with Referee #2's concern about the readability of this section, though, so I strongly encourage you to implement their suggestions to improve Section 3 in your revised manuscript.

Due to the additions we made in the revised version of the manuscript and its growing length, we finally decided to move most of the former Section 3 describing the model to an appendix, as suggested by Referee #2. Only the part on the constitutive relation for grain boundary sliding by Alley (1987) remains within the body of the manuscript because it is important for understanding the optimisation approach. It is now part of the "Methods" section. Nevertheless, we implemented all suggestions made by the referees and the editor.

Referee #2's general comment about Section 5: I respect your wishes to retain the order of your sections. Please consider editing the previous sections so that Section 5 begins earlier in the manuscript.

Due to the rearrangement of the manuscript including the addition of an appendix, the former Section 5 begins earlier in the manuscript now while maintaining the general structure of the sections.

**References**

Alley, R. B.: Firn Densification by Grain-Boundary-Sliding: A First Model, Journal de Physique, 48, C1–249 – C1–256, https://doi.org/https://doi.org/10.1051/jphyscol:1987135, 1987.

Hersbach, H., Bell, B., Berrisford, P., Hirahara, S., Horányi, A., Muñoz Sabater, J., Nicolas, J., Peubey, C., Radu, R., Schepers, D., Simmons, A., Soci, C., Abdalla, S., Abellan, X., Balsamo, G., Bechtold, P., Biavati, G., Bidlot, J., Bonavita, M., De Chiara, G., Dahlgren, P., Dee, D., Diamantakis, M., Dragani, R., Flemming, J., Forbes, R., Fuentes, M., Geer, A., Haimberger, L., Healy, S., Hogan, R. J., Hólm, E., Janisková, M., Keeley, S., Laloyaux, P., Lopez, P., Lupu, C., Radnoti, G., de Rosnay, P., Rozum, I., Vamborg, F., Villaume, S., and Thépaut, J.-N.: The ERA5 gloabal reanalysis, Quarterly Journal of the Royal Meterological Society, 146, 1999–2049, https://doi.org/https://doi.org/10.1002/qj.3803, 2020.

Morris, E. M., Muvaney, R., Arthern, R. J., Davies, D., Gurney, R. J., Lambert, P., De Rydt, J., Smith, A. M., Tuckwell, R. J., and Winstrup, M.: Snow Densification and Recent Accumulation Along the iSTAR Traverse, Pine Island Glacier, Antarctica, Journal of Geophysical Reasearch: Earth Surface, 122, 2284–2301, https://doi.org/https://doi.org/10.1002/2017JF004357, 2017.

Muñoz Sabater, J.: ERA5-Land monthly averaged data from 1981 to present, Tech. rep., Copernicus Climate Change Service (C3S) Climate Data Store (CDS), https://doi.org/10.24381/cds.68d2bb3, accessed on 30-08-2021, 2019.

---

## Referee Report (RR1)

Review of: *On the Contribution of Grain Boundary Sliding to Firn Densification - an Assessment using an Optimisation Approach*
by Schultz, Müller, Gross, and Humbert
* * *
**General Comments:**

This manuscript is much improved from the previous version – the scientific arguments are clearer and the manuscript is much easier to follow. I thank the authors for their careful consideration of the points I raised previously. I am pleased to recommend it for publication in the Cryosphere after a few minor points are addressed.

My biggest remaining concern regards melt: the authors mention that their model does not include melt and as such it is not considered, which I think is appropriate for this study. However, despite the authors' response, "We assume that the relatively low number of firn profiles influenced by melt compared to the overall number of sites does not affect our results", a quick glance at the map indicates that quite a number of the sites (in Greenland, at least) that are considered in this study do in fact experience melt each year. This study specifically considers densification near the surface, and it makes a strong case that grain boundary sliding is an adequate descriptor of the physical processes at play. However, in melt areas meltwater refreezing in the near surface firn is an additional (and potentially large) densification mechanism. I do not expect the authors to adapt their model to include melt, to try to determine if grain boundary sliding is still the correct physical descriptor in melt zones, or determine which of their sites do and not experience melt. However, I do think it would be appropriate in the discussion section to add a paragraph of how consideration of meltwater refreezing affects the authors' results – would their conclusions be the same? Would they expect their correlations to be better if only dry sites were considered? I think it is appropriate to include just a short bit of discussion, leaving questions open to be further investigated.

Aside from that point, I have a number of line-by-line comments that I expect will be easy fixes that will improve the readability of the paper.
* * *
**Line by line comments:**

Line 10: parameter → parameters

L53: Grammar on point (iv) is incorrect; do you mean 'how a modification of the constitutive relation by Alley (1987) could lead to an improvement of the description'?

L58: comma after investigations

L59: approach → approaches

L60: Change to 'Along with a large number...'

L69: Take out 'which incorporates the factor $D_{BD}$,' here because you describe later in the paragraph.

L73: I suggest changing to: "found in literature (e.g. Maeno and Ebinuma) and is further discussed in Section 2.2."

L86: Remove 'It has to be mentioned though, that' (colloquial phrase) – either just start with 'Alley (1987) …' or you could say, "We note that Alley…"

L87: Sentence is oddly worded; I suggest changing sentence structure to: "It is feasible that the strain rate due to grain boundary sliding decreases while it increases due to the influence of other physical processes"

Section 2.2, first paragraph: I appreciate the addition of this description of your study; it is succinct and clear.

L104: 'Further allows us' (word 'us' is missing)

L122-123: "The strain rate due to grain boundary sliding is therefore higher at the critical density when using the modification." → Because you have several variations yourself, please be specific describing this – I think you mean 'when using the Breant et al. (2017) modification'?

L134: Be specific of what you mean by constant values – constant values of temperature, accumulation rate, and surface density, I think? Are there others?

L134: How long is the spin up? What is the time step for the model runs?

I was initially concerned that your modeled profile would be affected by the steady-state spin up, but then I saw that you limit yourselves to 1958 and younger firn. Please expand on your method a bit here: I think that the 1958 surface is a modeled surface; is this correct? Or is the core dated and you know where the 1958 surface is in the observations? I am guessing that the 1958 date comes because that is when the RACMO RCM data begins; if so you should just state that clearly, e.g. something along the lines of: 'For all of our model-data comparisons, we limit our analyses to the firn shallower than the depth horizon of the modeled 1958 surface. This is because the climate data we use to force the model (RACMO2.3) begins in 1958. By imposing this limit, we ensure that the modeled firn profile used for comparisons is not affected by the spin up process." (and then continue your explanation of the alternate case of restricting to less than critical density). (and I do now see this in the Fig. 1 caption, but should be in the text)

L149: Remove 'as well'

L151: Change 'disregard' to 'exclusion'

L160: Change to: "This presents the problem of finding an appropriate surface-density boundary condition for the simulation"

L161: Offset "especially near the surface" with commas

L164: Remove: 'This method proofed to work well throughout the study'

I think you could be a bit more explicit stating that you tried all 21 surface density values with each of the 250 values of C; this is no small feat so I think it deserves to be highlighted a bit more. I suggest something like: remove 'following our approach', and: "For each of the 250 values of C for each variation, we tested 21 different…"

L165: The sentence starting: "Applying the method to all…" – I am not sure what you are getting at with this; either remove or rewrite/clarify.

L179 – point 4 – should be: 'must not exceed'

Figure 1 caption: specify that the colored dashed lines are model results, e.g. 'Colored dashed horizontal lines show modeled horizons of firn deposited in the indicated years'

L180: Specify that it is mean surface mass balance

L189-L190: Does this mean you should restrict your analyses to firn younger than 1979 for Antarctic cores (as described in Section 2.2)?

L212: I realize here that you are responding to the other referee's comment asking for a statement to this extent, but as it is written it sounds as if you are doing the spatial interpolation, where in reality is RACMO that is doing a spatial interpolation of ERA data – perhaps, "The spatially-interpolated RACMO fields have the potential to include systematic errors" or something like that – and you could cite (in which they specifically mention RACMO bias) e.g. *van Wessem, J. M., van de Berg, W. J., Noël, B. P. Y., van Meijgaard, E., Amory, C., Birnbaum, G., Jakobs, C. L., Krüger, K., Lenaerts, J. T. M., Lhermitte, S., Ligtenberg, S. R. M., Medley, B., Reijmer, C. H., van Tricht, K., Trusel, L. D., van Ulft, L. H., Wouters, B., Wuite, J., and van den Broeke, M. R.: Modelling the climate and surface mass balance of polar ice sheets using RACMO2 – Part 2: Antarctica (1979–2016), The Cryosphere, 12, 1479–1498, https://doi.org/10.5194/tc-12-1479-2018, 2018.*

L213: Can you be more specific about these systematic errors? I don't think for this paper you want to go down the road of investigating systematic errors in RCM outputs; perhaps it would be more appropriate for you to simple state something like, "any error in the RCM forcing data will manifest itself as error in the modeled firn profiles; these error analyses are outside the scope of this paper".

L220: change to: "We use second example to illustrate how…"

Figure 3 caption: The second sentence ('Colour coded…') does not make sense – reword/rewrite for clarity

L257: consider removing 'easily' – it would indeed be easy to add the Freitag impurity model, but I don't think getting the model to realistically simulate layering is easy.

L259: I don't follow what you are saying that the activation energy is temporally averaged? The Freitag activation energy, or the Arrhenius one? Didn't you just say that you are not including the Freitag equation?

L262: I don't follow here what you are trying to say about running mean – I agree that it neglects information, but you are simulating the firn at annual resolution, while the layered firn you are comparing to is much higher resolution than that. Are you assuming that deviations due to layering will be equally distributed positive and negative? I am not saying your method is wrong, but I think you need to explain your thinking more clearly. I would omit the part about the Freitag equations (there are some number of people in the firn community who don't agree that impurities are the source of layering), and stick to a simpler story: layering exists in real firn, your model does not simulate that (most firn models do not), you are still comparing to the raw data, and here is why.

L266: either state here that the neumann condition is set to zero, or reference section A4.

L302: Dependency on what?

---

## Author Response (AR2)

**Answer to Editor decision, 28 Oct 2021**

Dear Kaitlin,

Thank you again for editing our manuscript. As noted by you and the referees the grammar of our manuscript needed major revision. We have commissioned a professional language editing service to help us revise the manuscript. The new version is greatly improved with regard to the language. We also discussed all further remarks raised by the referees. Due to the comment of Referee #1 regarding the title of the manuscript and how it could result in confusion, we decided to change it.

*On the contribution of grain boundary sliding type creep to firn densification – an assessment using an optimization approach*

With the new title we want to highlight that the paper is not about an investigation of the process of grain boundary sliding itself, but about the functional relation describing this type of creep. We hope you approve of this.

Sincerely,

Timm Schultz on behalf of the co-authors

**Response to Anonymous Referee #1, 01 Oct 2021**

Dear colleague,

We thank you for reading our manuscript carefully again. We discussed all of your remarks. You can find point-by-point responses to them below. We contracted a professional language editing service to improve the grammar and readability of the manuscript. We thought a lot about the title of the paper and decided to change it to avoid confusion about the topic of the manuscript as you suggested.

*On the contribution of grain boundary sliding type creep to firn densification – an assessment using an optimization approach*

With this change we want to clarify that we don't investigate the process of grain boundary sliding itself, but its description. The micro-mechanical description of grain boundary sliding, like other forms of creep, for example Coble creep (Coble, 1970), is established by a functional relation of a certain type (see for example Goldsby and Kohlstedt, 2001). Alley (1987) adopted grain boundary sliding to firn. We assess the functional relation used for the description of this type of creep. We hope you approve of this new title.

Sincerely,

Timm Schultz on behalf of the co-authors

**General Comments**

This is a greatly improved version of the original paper. The authors have certainly put a lot of effort into answering the points made by both referees. There are still, however, places where further editing is needed so that the research is presented clearly. Part of the problem is that the English still needs a lot of improvement, but I think there is also an underlying confusion about what the paper is really about. For example, in the Abstract (lines 1 to 4) the authors draw a distinction between the Alley model for stage 1 densification, which they describe as a model for grain-boundary sliding, and models such as the Herron and Langway model, which they imply are not descriptions of grain-boundary sliding. But in fact the stage 1 expressions in Herron and Langway type models are generally considered to be models for grain-boundary sliding, they are just not developed by consideration of the detailed physics of the process. The distinction the authors are actually making is between an upscaled microphysics model (the Alley model) and macroscale Herron and Langway type models for the same process. The paper is not about the contribution of grain boundary sliding to firn densification - it is about whether the Alley model is a good representation of this process. I think the authors know this; it is just a question of editing the text and changing the title to avoid confusing the reader. I would suggest that the authors work through the paper, look at each mention of "grain boundary sliding" and consider whether they really mean "the Alley equation". For example at line 50 it would be better to write "In this study we aim to evaluate (i) whether the Alley equation is suitable for the simulation of firn densification at low density (ii) how the Breant et al. (2017) modification to this constitutive relation affects simulation results etc". In general the whole paper needs to be reviewed carefully to make sure that the text always says what the authors want it to say.

> As the referee suggested we worked again through the manuscript and reviewed in which way the term "grain boundary sliding" was used. Where it was necessary, we revised the manuscript to distinguish more clearly between the process of grain boundary sliding and Alley's description of this process. This includes the example in former line 50 mentioned by the referee. We further changed the title of the manuscript to avoid confusion.

**Specific Comments**

l. 45 the authors seem to imply that if the Alley equation does not reproduce stage 1 densification exactly this might indication that grain boundary sliding is not the only process acting in stage 1. But it could also indicate that the Alley equation is not an adequate description of grain boundary sliding.

> We are not aware of any firn densification model that reproduces stage 1 densification exactly nor are we aware of any model which reproduces nature exactly. As the addressed paragraph leads to confusion we decided to remove it from the manuscript.

l.144 This is a somewhat confusing sentence and needs to be rewritten. The authors have mixed up two things (i) that the factor 5/3 is derived from N/6 where N is the number of bonds and is approximately equal to 10 rho/rhoi and (ii) that the factor (1-5rho/3rhoi) implies that the densification rate goes to zero at a critical density rhoc = 550 kg m-3. Note also that relative density could mean rho/rhow so "density relative to ice" would be better.

> We note that the derivation of the factor 5/3 is somewhat different than described in this paragraph. Especially as theoretical densest packing, hexagonal or cubic, would result in a higher value of the critical density and a coordination number of $N = 12$. As established by Anderson & Benson (1963) the critical density corresponds to "close random packing". Our description was imprecise there.
>
> However the factor $1 - (5\rho/3\rho_{\text{ice}})$ leads to a strain rate of zero, when the critical density is reached. We narrowed our description to this point. The reader might look up the exact derivation of the factor in the original publication. We further established that density relative to ice density is meant here.
>
> *The factor of $1 - (5\,\rho/3\,\rho_{\text{ice}})$ causes the strain rate due to grain boundary sliding to decrease with increasing density until it vanishes at the critical density of $\rho_c = 550\,\text{kg}\,\text{m}^{-3}$. When the critical density is reached, close random packing is established (Anderson and Benson, 1963), and grains can no longer slide against each other; thus, the process of grain boundary sliding ends. Other deformation processes, in particular dislocation creep (Maeno and Ebinuma, 1983), result in further densification with increasing stress.*

l.285. It might be worth mentioning here that Alley specifies model density equals observed density at 2 m depth whereas the authors choose to treat surface density as another parameter to be optimised.

We established this in the revised version of the mansucript.

*Although Alley (1987) simulated the density starting at a depth of 2 m below the surface, we included this domain in our simulation so that we could apply transient surface forcing to our model. To find suitable values of the surface density, we included this parameter in our optimization.*

l. 400. Are the authors saying the geothermal heat flux is negligible for all sites, hence the Neumann lower boundary condition on temperature can be used?

While the geothermal heat flux is certainly not negligible for all sites in general, its influence on the uppermost 25 meters of firn is indeed negligible to the best of our judgement. The geothermal heat flux acts on the ice base several hundreds of meters up to thousands of meters beneath the analysed firn domain and is thus a kind of far field boundary conditions of the ice sheet, whereas we study here the firn column only. In order to support this assumption, we evaluate observed firn temperature profiles that are found in the literature.

[Figure]

Figure 1: Figure 1. Temperature profile of the West Antarctic Ice Sheet Divide Deep Borehole (Cuffey and Clow, 2011). The data were acquired in December 2011 at the WAIS–D (N -79.4676°, E -112.0865°).

Figure 1 shows the temperature profile for the Deep Borehole at the West Antarctic Ice Sheet Divide measured in 2011 (Cuffey and Clow, 2011). Starting at a depth of about 2000 m below the surface the influence of geothermal heat is obvious. The ice temperature rises from about 242 K to 264 K at the ice base. However, in the upper part of the ice column the influence of the geothermal heat flux declines fast. The temperature is determined by the mean surface temperature. We therefore conclude that the geothermal heat flux from the ice base is negligible in our study focussing on the upper 25 m of the firn column.

On the other hand the question can be raised to which extent the temperature differs at the depth of 25 m due to the influence of varying surface temperatures. The following figure shows temperature profiles of different locations. They all indicate, as well as the results shown in Orsi et al. (2017) and Vandecrux et al. (2021), which are already cited in the manuscript, that the change in temperature is small at a depth of 25 m below the surface. Note that even in Figure 3 (d), at the Styx location, where there is strong dry ablation predominant, the change in the temperature profile at a depth of 25 m below the surface is relatively small. Therefore we believe that the use of a temperature boundary condition as described in the manuscript is well justified. This is especially true as we restricted the domain of comparison to the firn influenced by the surface forcing.

[Figure]

Figure 2: Temperature profile measured at position NGT36, Greenland, N 77°29', W 47°28', (Schwager, 2000)

[Figure]

Figure 3: Observed borehole temperature and and reconstructed past surface temperatures at four antarctic sites, from Lyu et al. (2020).

Cuffey, K. and Clow, G. D. (2011). Temperature Profile of the West Antarctic Ice Sheet Divide Deep Borehole, Version 1. NSIDC, Data Set ID: NSIDC-0550, https://nsidc.org/data/NSIDC-0550/versions/1 [accessed: 12.11.2021].

Lyu, Z., Orsi, A. J. and Goosse, H. (2020). Comparison of observed borehole temperatures in Antarctica with simulations using a forward model driven by climate model outputs covering the past millennium. Climate Past, 16, 1411-1428, https://doi.org/10.5194/cp-16-1411-2020.

Orsi, A. J., Kawamura, K., Masson-Delmotte, V., Fettweis, X., Box, J. E., Dahl-Jensen, D., Clow, D., Landais, A. and Severinghaus, J. P. (2017). The recent warming trend in north Greenland. Geophysical Research Letters, 44, 6235-6243, https://doi.org/https://doi.org/10.1002/2016GL072212.

l. 560. The authors state that the use of a global value for the Alley parameter leads to worse simulation results than existing firn density models (by which they presumably mean the Herron and Langway type models for stage 1 densification). This is an important result, even if it is not what the authors wished to discover. The reader is left wishing that instead of only comparing results for the 4 variants of the Alley model the authors had included a benchmark macroscale model (for example Herron and Langway) and compared all 5 models.

Actually the presented study started with the idea of checking if we were able to find a global factor for the constitutive relation by Alley (1987), which led to good simulation results. This was motivated by the method used by Alley (1987) to determine the value of the amplitude of grain boundary obstructions. Instead we found that the optimal values for the individual sites scatter in a relatively wide range. So we asked ourselves why this is the case. We found indications that the optimal values depend on the mean surface temperature and the mean surface mass balance to some degree. In our opinion this is an interesting finding. This result can stand for itself.

We see no advantages in running the model with a global factor for the constitutive relation by Alley (1987). In this case the advantages of the micromechanical modeling approach would be lost. A global factor, for example the mean or median value of the location specific factors, would lack physical meaning.

**Response to Referee #2, 11 Oct 2021**

Dear Max,

Thank you for revising our manuscript carefully again. A special thank you for your comprehensive remarks regarding the grammar. They helped to improve the readability of the manuscript greatly. Additionally we have commissioned a language editing service to help us improve the manuscript You can find point-by-point answers to all your remarks below. Due to the review of the other referee we decided to change the title of the manuscript.

*On the contribution of grain boundary sliding type creep to firn densification – an assessment using an optimization approach*

With the new title, we want to clarify that we don't investigate the process of grain boundary sliding itself, but the functional relation describing this type of creep. We hope you approve of this.

Sincerely,

Timm Schultz on behalf of the co-authors
* * *
**General Comments**

This manuscript is much improved from the previous version – the scientific arguments are clearer and the manuscript is much easier to follow. I thank the authors for their careful consideration of the points I raised previously. I am pleased to recommend it for publication in the Cryosphere after a few minor points are addressed.

My biggest remaining concern regards melt: the authors mention that their model does not include melt and as such it is not considered, which I think is appropriate for this study. However, despite the authors' response, "We assume that the relatively low number of firn profiles influenced by melt compared to the overall number of sites does not affect our results", a quick glance at the map indicates that quite a number of the sites (in Greenland, at least) that are considered in this study do in fact experience melt each year. This study specifically considers densification near the surface, and it makes a strong case that grain boundary sliding is an adequate descriptor of the physical processes at play. However, in melt areas meltwater refreezing in the near surface firn is an additional (and potentially large) densification mechanism. I do not expect the authors to adapt their model to include melt, to try to determine if grain boundary sliding is still the correct physical descriptor in melt zones, or determine which of their sites do and not experience melt. However, I do think it would be appropriate in the discussion section to add a paragraph of how consideration of meltwater refreezing affects the authors' results – would their conclusions be the same? Would they expect their correlations to be better if only dry sites were considered? I think it is appropriate to include just a short bit of discussion, leaving questions open to be further investigated.

Aside from that point, I have a number of line-by-line comments that I expect will be easy fixes that will improve the readability of the paper.

> We agree with your point that surface melting, melt water percolation through the firn body, refreezing and the general interaction of melt water with the firn is an important part within the field of firn densification modelling. As suggested by you we added a paragraph on this subject in the discussion section.

> *This study analyzed only dry firn densification. The current model cannot handle melting. We accommodate this feature by setting the annual mean surface mass balance at the investigated sites to be strictly positive (Sect. 3.1). However, this limitation means that we cannot ensure that no melting occurs over the course of a year. The results shown in Fig. 8 illustrate how this limitation affects the optimization results. The limitation is problematic, especially in recent years, when melting occurred over almost the entire Greenland Ice Sheet (e.g. Nghiem et al., 2012). The simulation of meltwater percolation through the firn and its interaction with firn densification is important, especially in the upper part of the firn body (e.g. Vandecrux et al, 2020). The proposed method could be improved by the application of this model approach in future investigations. However, we identified some correlations between the optimization results and the surface mass balance.*

**Line by line comments**

Line 10: parameter → parameters

Corrected as suggested by the referee.

L53: Grammar on point (iv) is incorrect; do you mean "how a modification of the constitutive relation by Alley (1987) could lead to an improvement of the description"?

We reformulated the sentence (L39). It now reads:

*. . . and (iv) how the constitutive relation of Alley (1987) might be improved.*

L58: comma after investigations

We added the comma after "investigations" in line 44. We expect this line was meant.

L59: approach → approaches

It seems some mix up happened with the line numbers. We assume the comment refers to the sentence in line 44, which we reformulated. It now reads:

*In contrast to these experimental investigations, a data-driven model approach is used in our study.*

L60: Change to "Along with a large number..."

We decided to change and split the addressed sentence for better understandability and readability.

*Since the original study of Alley (1987) was published, the amount of available data has increased greatly. The data include a large number of firn profiles and forcing data, and they allow us to simulate firn profiles at very high quality using additional modeling techniques.*

L69: Take out "which incorporates the factor DBD," here because you describe later in the paragraph.

Corrected as suggested by the referee.

L73: I suggest changing to: "found in literature (e.g. Maeno and Ebinuma) and is further discussed in Section 2.2."

We altered the sentence as suggested by the referee.

L86: Remove "It has to be mentioned though, that" (colloquial phrase) – either just start with "Alley (1987) ..." or you could say, "We note that Alley..."

We adopted the suggestion by the referee.

*Alley (1987) suggested that additional processes contribute to densification below the critical density.*

L87: Sentence is oddly worded; I suggest changing sentence structure to: "It is feasible that the strain rate due to grain boundary sliding decreases while it increases due to the influence of other physical processes"

We reformulated the sentence. It now reads:

*Alley (1987) suggested that additional processes contribute to densification below the critical density. It is feasible that the effect of grain boundary sliding on the strain rate decreases, whereas that of other processes increases. The studies of Arthern and Wingham (1998) and Bréant et al. (2017) use ...*

Section 2.2, first paragraph: I appreciate the addition of this description of your study; it is succinct and clear.

L104: "Further allows us" (word "us" is missing)

Corrected as suggested by the referee.

L122-123: "The strain rate due to grain boundary sliding is therefore higher at the critical density when using the modification." → Because you have several variations yourself, please be specific describing this – I think you mean "when using the Breant et al. (2017) modification"?

We edited the sentence to clarify the modification introduced by Bréant et al. (2017) is meant here.

*It was introduced to obtain a better transition to the second stage of firn densification. The strain rate due to grain boundary sliding is therefore higher at the critical density when this modification of Bréant et al. (2017) is used.*

L134: Be specific of what you mean by constant values – constant values of temperature, accumulation rate, and surface density, I think? Are there others?

We added a sentence to briefly describe the forcing.

*Every simulation begins with a spin-up period in which constant values are used for forcing. The model is forced with prescribed values of temperature, accumulation rate, firn density, and grain radius at the surface.*

L134: How long is the spin up? What is the time step for the model runs?

I was initially concerned that your modeled profile would be affected by the steady-state spin up, but then I saw that you limit yourselves to 1958 and younger firn. Please expand on your method a bit here: I think that the 1958 surface is a modeled surface; is this correct? Or is the core dated and you know where the 1958 surface is in the observations? I am guessing that the 1958 date comes because that is when the RACMO RCM data begins; if so you should just state that clearly, e.g. something along the lines of: "For all of our model-data comparisons, we limit our analyses to the firn shallower than the depth horizon of the modeled 1958 surface. This is because the climate data we use to force the model (RACMO2.3) begins in 1958. By imposing this limit, we ensure that the modeled firn profile used for comparisons is not affected by the spin up process." (and then continue your explanation of the alternate case of restricting to less than critical density). (and I do now see this in the Fig. 1 caption, but should be in the text)

The length of the spin up is determined by the time it takes to reach steady state, as we use constant forcing during the spin up. We determine steady state conditions by evaluating the change of density between consecutive time steps. If the maximum change in density between two time steps is smaller than $0.1 \, \text{kg m}^{-3}$ we assume the profile will not change significantly anymore and steady state is reached.

For spin up and transient simulation runs we use a constant value of 48 time steps per year. This information was temporarily lost due to the fact the model description moved to the appendix.

You are right about your assumptions on the domain used for comparing simulated and measured firn densities. We wanted to include only those results for which we know about past climate conditions. In this way the spin up plays a subordinate role.

We edited the section based on your suggestions to include this information and to highlight more clearly what we wanted to express there.

L149: Remove "as well"

Corrected as suggested. The sentence now reads:

*As the implementation of our model is efficient and the approach is simple and reliable, we decided to determine the best factor $C_v$ for the four variants of the constitutive equation by simply testing 250 values within certain ranges.*

L151: Change "disregard" to "exclusion"

We reformulated the sentence it now reads:

*These ranges are shown in Eqs. (6) and (7), which include and exclude the Arrhenius factor, respectively.*

L160: Change to: "This presents the problem of finding an appropriate surface-density boundary condition for the simulation"

The sentence was changed as suggested.

L161: Offset "especially near the surface" with commas

Corrected as suggested.

L164: Remove: "This method proofed to work well throughout the study"

I think you could be a bit more explicit stating that you tried all 21 surface density values with each of the 250 values of C; this is no small feat so I think it deserves to be highlighted a bit more. I suggest something like: remove "following our approach", and: "For each of the 250 values of C for each variation, we tested 21 different..."

We removed the sentence in line 164. Furthermore we altered the following paragraph based on your suggestions.

*For each of the four variants and 250 factors $C_v$, we tested 21 values of the surface density between $\rho_0 = 250 \, \mathrm{kg \, m^{-3}}$ and $\rho_0 = 450 \, \mathrm{kg \, m^{-3}}$, using steps of $\Delta \rho_0 = 10 \, \mathrm{kg \, m^{-3}}$.*

L165: The sentence starting: "Applying the method to all..." – I am not sure what you are getting at with this; either remove or rewrite/clarify.

We added this part in response to a comment we received from you after the first review which raised the question why we haven't used measured near surface density values when available.

*L269: In the cases where you have cores that have measurements from the near surface, why don't you just use the measured surface density from the cores? Are there cases where the surface density from the optimization scheme is significantly different than the observations from SUMup? It seems possible that your scheme could allow a surface density that is different than the observation in order to get a better overall RMSD fit. Do you use the same density at all time steps for a given site/simulation?*

With the addressed sentence we wanted to justify why we haven't used measured near surface values directly. By applying the testing method to all profiles we establish comparability. However, density profiles reaching to the surface are well represented. Furthermore the use of measured values always requires some kind of adjustment of the measured density values. One can not simply pick the first measured value because the variability of the density is high at low depths.

We have adjusted the addressed sentence to clarify our point.

*We used the method of testing 21 surface density values for all the analyzed firn profiles. We included profiles including measurements of the density at small depths. In this way, we established that the results are comparable. Profiles including near-surface density values are, however, well-represented.*

L179 – point 4 – should be: "must not exceed"

The additional "to" was deleted.

Figure 1 caption: specify that the colored dashed lines are model results, e.g. "Colored dashed horizontal lines show modeled horizons of firn deposited in the indicated years"

We altered the addressed part within the caption according to the suggestion for clarification.

*Colored dashed horizontal lines show horizons obtained in the simulations. Horizons plotted in gray (to the right of the vertical dashed line) represent the same surfaces as those determined by Miller and Schwager (2004) during analysis of the core.*

L180: Specify that it is mean surface mass balance

We have adjusted the bullet point as suggested.

*The annual mean surface mass balance at the profile locations must be strictly positive.*

L189-L190: Does this mean you should restrict your analyses to firn younger than 1979 for Antarctic cores (as described in Section 2.2)?

Yes, this is true and we considered it in our optimisation and analysis. As this wasn't formulated clear enough before, we added some context in Section 2.2.

*For firn profiles retrieved in Antarctica, climate forcing from RACMO2.3 begins in 1979. Thus, only those results located above the simulated horizon of 1979 are considered for comparison with the Antarctic firn profiles.*

L212: I realize here that you are responding to the other referee's comment asking for a statement to this extent, but as it is written it sounds as if you are doing the spatial interpolation, where in reality is RACMO that is doing a spatial interpolation of ERA data – perhaps, "The spatially-interpolated RACMO fields have the potential to include systematic errors" or something like that – and you could cite (in which they specifically mention RACMO bias) e.g.

van Wessem, J. M., van de Berg, W. J., Noël, B. P. Y., van Meijgaard, E., Amory, C., Birnbaum, G., Jakobs, C. L., Krüger, K., Lenaerts, J. T. M., Lhermitte, S., Ligtenberg, S. R. M., Medley, B., Reijmer, C. H., van Tricht, K., Trusel, L. D., van Ulft, L. H., Wouters, B., Wuite, J., and van den Broeke, M. R.: Modelling the climate and surface mass balance of polar ice sheets using RACMO2 – Part 2: Antarctica (1979–2016), The Cryosphere, 12, 1479–1498, https://doi.org/10.5194/tc-12-1479-2018, 2018.

Indeed we have performed a spatial interpolation as the profile sites do not accidently match the grid of RACMO output. We thought for completeness this should be mentioned within the manuscript. As it is an ongoing topic of confusion we deleted the sentence regarding this point in the revised version of the manuscript. It is kind of self-explanatory.

To serve the demands of Referee #1 we still mention that RACMO data may contain systematic errors. We will cite the paper suggested by you. Thank you for this useful suggestion.

L213: Can you be more specific about these systematic errors? I don't think for this paper you want to go down the road of investigating systematic errors in RCM outputs; perhaps it would be more appropriate for you to simple state something like, "any error in the RCM forcing data will manifest itself as error in the modeled firn profiles; these error analyses are outside the scope of this paper".

Please see our response to your comment on line 212.

L220: change to: "We use second example to illustrate how..."

We changed the sentence as suggested by the referee. The sentence now reads:

*We present a second example to illustrate the effect of the temporal resolution of the forcing on the optimization results and why we used yearly averaged data provided by RACMO2.3.*

Figure 3 caption: The second sentence ("Colour coded...") does not make sense – reword/rewrite for clarity

We rephrased the addressed sentence.

*Colored lines show the optimal simulation results for four tested variants of the constitutive relation.*

L257: consider removing "easily" – it would indeed be easy to add the Freitag impurity model, but I don't think getting the model to realistically simulate layering is easy.

As suggested we removed "easily" from the addressed sentence.

L259: I don't follow what you are saying that the activation energy is temporally averaged? The Freitag activation energy, or the Arrhenius one? Didn't you just say that you are not including the Freitag equation?

We reformulated the addressed paragraph as it was not clearly understable before. Please see our answer to your comment on line 262 below.

L262: I don't follow here what you are trying to say about running mean – I agree that it neglects information, but you are simulating the firn at annual resolution, while the layered firn you are comparing to is much higher resolution than that. Are you assuming that deviations due to layering will be equally distributed positive and negative? I am not saying your method is wrong, but I think you need to explain your thinking more clearly. I would omit the part about the Freitag equations (there are some number of people in the firn community who don't agree that impurities are the source of layering), and stick to a simpler story: layering exists in real firn, your model does not simulate that (most firn models do not), you are still comparing to the raw data, and here is why.

We fully agree with you regarding this point. However, both referees stressed the point that small scale layering is not covered within the model before. We therefore added this paragraph. However, we reformulated it to address the problem of simulating small scale layering in firn in a more general form.

*Ice core ngt03C93.2 (Fig. 1a) is an example of a high-resolution density measurement showing extensive small-scale layering. Only a few of the 159 firn profiles are of such high quality and include this type of layering. Although our proposed model works at high temporal and spatial resolution, it does not cover layering, as shown in Fig. 1a. The density profile retrieved at site 3 of the iSTAR traverse (Morris et al., 2017) (Fig. 3) illustrates that the model, if it is forced with data of higher temporal resolution, still does not cover the measured density variability. Small-scale layering of firn appears to be driven by a number of different processes (Hörhold et al., 2011). An extension of the model to cover such processes may be introduced in the future. We would prefer the approach of Freitag et al. (2013) who introduced the concept of impurity-controlled densification. Forcing data for this model are not globally available. However, the model in its current form does reproduce the mean density well, as demonstrated in Fig. 1a.*

L266: either state here that the Neumann condition is set to zero, or reference section A4.

We added the information that the Neumann condition prescribes the temperature gradient to be zero at the profile base.

*This choice raises the question of whether the use of a Neumann boundary condition set to zero at the profile base to solve for the temperature is justified for this particular model setup (Sect. A4).*

L302: Dependency on what?

We reformulated the addressed sentence.

*To check for possible mean surface temperature dependence of the 159 factors found by optimization, these values are plotted against each other in Fig. 7. The values of the mean surface temperature were calculated from the forcing data for each firn profile site.*